

# ESD Reviews: mechanisms, evidence, and impacts of climate tipping elements

Seaver Wang[1], Zeke Hausfather[1]

[1]The Breakthrough Institute, Oakland, 94612, United States

*Correspondence to*: Seaver Wang (seaver@thebreakthrough.org)

**Abstract.** Increasing attention is focusing upon "climate tipping elements" – large-scale earth systems anticipated to respond through positive feedbacks to anthropogenic climate change by shifting towards new long-term states. In some but not all cases, such changes could produce additional greenhouse gas emissions or radiative forcing that could compound global warming. Developing greater understanding of tipping elements is important for predicting future climate risks. Here we

review mechanisms, predictions, impacts, and knowledge gaps associated with ten notable climate tipping elements. We also evaluate which tipping elements are more imminent and whether shifts will likely manifest rapidly or over longer timescales. Some tipping elements are significant to future global climate and will likely affect major ecosystems, climate patterns, and/or carbon cycling within the current century. However, assessments under different emissions scenarios indicate a strong potential to reduce or avoid impacts associated with many tipping elements through climate change mitigation. Most tipping elements

do not possess the potential for abrupt future change within years, and some tipping elements are perhaps more accurately termed climate feedbacks. Nevertheless, significant uncertainties remain associated with many tipping elements, highlighting an acute need for further research and modeling to better constrain risks.

## 1 Introduction

Global climate change is projected to continue over the 21st century in response to ongoing human emissions of greenhouse

gases and land-use changes (Peters et al., 2020; Raftery et al., 2017). In assessments of impacts associated with climate change, increasing focus is centering around "climate tipping elements" – large-scale mechanisms or systems associated with positive feedbacks that could potentially be triggered in response to modest levels of additional warming (Lenton et al., 2008). These tipping elements are often comprised of major carbon cycle feedbacks, climate shifts, and/or ecosystem changes and include mechanisms like Arctic permafrost thaw (Turetsky et al., 2020), destabilization of methane hydrates in marine sediments

(Archer et al., 2009), and warming-induced mass loss from major ice sheets (Golledge et al., 2019).

Such climate tipping elements are thought to have the potential to drive significant biodiversity loss, alter the boundaries of major ecological biomes, and produce important consequences for human society via effects like sea level rise. Some researchers have also proposed that such tipping elements could dynamically interact with one another, producing

compounding effects that could ultimately commit the climate system to several degrees of additional warming beyond the level implied by anthropogenic factors alone (Steffen et al., 2018). Such warming thresholds have been proposed to have the potential to alter the long-term climate trajectory of the earth on geologic timescales (Lenton et al., 2019). Even apart from the risk of compounding greenhouse gas emissions and radiative forcing, however, tipping elements carry important implications



for the future well-being of human communities and natural ecosystems at local, regional, and global scales. Many tipping
mechanisms may also be difficult to halt, reverse, or mitigate once they have begun shifting between states in response to
climate perturbations.

Evaluating the projected impacts associated with climate tipping elements and their potential interactions has proven
challenging. Most of the most frequently-cited climate tipping elements, including but not limited to permafrost carbon release
(Turetsky et al., 2019), disruption of the Atlantic Meridional Overturning Circulation (AMOC) (Lynch-Stieglitz, 2017),
degradation of the Amazon rainforest (Nobre et al., 2016), and large-scale ecosystem shifts within the northern circumpolar
boreal forest (Scheffer et al., 2012), involve highly complex systems, competing feedbacks in response to climate change, and
large uncertainties. Many of these systems are poorly resolved by the current generation of climate models or omitted
altogether, making such mechanisms – alongside carbon cycle feedbacks - an important source of larger uncertainty regarding
climate predictions and global climate sensitivity.

Discussion of "tipping elements" and "tipping points" in the context of earth and climate science research increased sharply
in the mid-2000s (Kopp et al., 2016) with work (Lenton et al., 2008; Lenton and Schellnhuber, 2007) establishing definitions
for climate tipping elements and proposing a number of earth system mechanisms and feedbacks as examples. Climate tipping
elements have since become an increasing focus of research and discussion e.g. (Cai et al., 2016; Kriegler et al., 2009; Lenton,
2011, 2012). The prominence of tipping elements was recently further amplified by a 2018 research paper which assigned
elevated risks of triggering a "tipping point cascade" leading to irreversible and significant additional global warming and a
"hothouse" climate for warming of >2C above pre-industrial temperatures (Steffen et al., 2018), and researchers have since
continued to develop tipping point cascade theory (Lenton et al., 2019). At the same time, the concept of climate tipping points
has also become a popular topic of discussion within media and political discourse (Extinction Rebellion, 2019; McKibben,
2019).

Increasing frequency of use has demonstrated that the definition of what constitutes a "tipping point" can itself vary greatly
when comparing how different commentators, reporters, and scientists employ the term. Whether tipping mechanisms are
necessarily irreversible, fast-acting once activated, or triggered under a precise versus a broad range of climate conditions
depends strongly on the writer. Inconsistency in how the term has been employed can lead to considerable confusion,
particularly regarding the timescales over which climate mechanisms act.

As outlined by (Kopp et al., 2016), a "tipping point" as invoked in socioeconomic contexts prior to the term's adoption by
climate researchers referred to a small change beyond a key threshold triggering networked positive feedbacks and a rapid
shift between dramatically different system states. Subsequently, (Lenton et al., 2008) interpreted "climate tipping points" as
referring more generally to dramatic shifts in components of the climate system resulting from relatively small forcings,



including examples not involving network-associated feedbacks and/or anticipated to take place over longer time scales of centuries or more. This latter, broader definition has become standard within contemporary discussions of climate tipping

points in the earth science community (Lenton et al., 2019; Steffen et al., 2018). Although (Kopp et al., 2016) advocated for restricting the use of "tipping points" to refer only to systems capable of rapid change and instead employing the more general term "tipping elements" for mechanisms fitting the broader definition, such distinctions have not generally been made within the       literature.

In this report, we have adopted the convention proposed by (Kopp et al., 2016) to maximize clarity, characterizing rapidly-acting (within a decade) systems as "tipping points" and otherwise utilizing the term "tipping elements". We describe systems with a more linear, direct, predictable response to climate forcing simply as "climate feedbacks". Further, we use the term "irreversible" to refer only to changes that cannot be stopped or reversed on non-geologic timescales once initiated, while a "reversible" system refers to mechanisms that can be either halted or reverted to the original state under the right conditions.

Note that our chosen terminology for tipping elements, tipping points, and climate feedbacks does not necessarily imply reversible or irreversible changes for any category.

Given the high importance of climate tipping elements for informing future risk assessments and determining optimal societal actions, a firmer understanding of potentially abrupt (acting within a couple decades) climate mechanisms and how they may

interact holds the potential to greatly benefit climate discourse. This review seeks to help fulfill this need by synthesizing the latest research on a number of the most frequently-discussed climate tipping elements. Evaluation of the risks posed by climate tipping elements requires considering the timescales over which a climate mechanism would act, the climate impact of the tipping element, and important uncertainties surrounding factors required to trigger such mechanisms.

We explore each of these considerations for ten global and regional climate tipping elements and feedbacks (Table 1) that have featured prominently in broader climate discourse and are commonly thought to present significant global or regional dangers. We also evaluate scientific discussion of the likelihood of a "tipping point cascade" with illustrative aid from a simple climate model (FAIR) (Millar et al., 2017; Smith et al., 2017) and find the additional change in global mean surface temperature over the 21st century from tipping element impacts to be 0.54C in the high-end RCP8.5 warming scenario. Finally, we provide an

overall assessment of the climate tipping elements that we consider the highest-risk mechanisms in terms of driving more significant, imminent changes to major ecosystems, global carbon cycling, or climate patterns and circulation systems.


## 2 Global tipping elements

### 2.1 Slowdown or collapse of the Atlantic Meridional Overturning Circulation (AMOC)

#### 2.1.1 Background

The Atlantic Meridional Overturning Circulation (AMOC), also referred to as the thermohaline circulation (THC), is a large-scale pattern of ocean circulation taking place in the North Atlantic Ocean. A key feature driving heat transport in the Atlantic basin from low to high latitudes, this circulation pathway is also responsible for generating the formation of North Atlantic Deep Water, which represents a major flux of heat, salt, and biogeochemical tracers into the deep ocean. On a global scale, the AMOC is responsible for redistributing heat between the Northern and Southern Hemispheres, thereby playing a significant

role in the climate system (Buckley and Marshall, 2016).

Ongoing climate change has sparked concerns that this important feature of ocean circulation could be disrupted by rising water temperatures and melting ice sheets. A slowdown or shutdown of the AMOC system would significantly affect regional and global climate patterns. Paleoclimate evidence has identified weakening of the AMOC as an important potential driver of

multiple large, abrupt shifts in past climate, including changes occurring on timescales as short as a few decades (Alley et al., 2001; Clark et al., 2001). The impacts of past AMOC shifts affected climate globally, significantly altering tropical rainfall patterns and causing heat redistribution between the northern and southern hemispheres (Masson-Delmotte et al., 2013). Changes to the overturning circulation could also affect the ocean's strength as a heat and carbon sink.

On interannual timescales, the AMOC is driven by large-scale wind patterns across the North Atlantic that move warm, saline waters northward in the upper 1000 m and cold, fresh waters southwards at depth. Over decadal timescales and beyond, temperature and salinity influences also exert a strong influence over this overturning circulation (Biastoch et al., 2008). The poleward-moving waters of the warm Gulf Stream unite with the colder Labrador Current east of Newfoundland's Grand Banks, forming the North Atlantic Current (NAC), which in turn travels north before dividing into a number of branches

feeding into the subpolar gyre (Buckley and Marshall, 2016). Cooling as they travel north along these pathways, these waters increase in density and lose their buoyancy, sinking once they reach a critical density threshold and becoming North Atlantic Deep Water (NADW) (Broecker, 1991).

Increasing temperatures in combination with heightened meltwater fluxes from the Greenland Ice Sheet (GIS) could warm and

freshen surface waters to the point of slowing or even halting the formation of NADW (Bakker et al., 2016). As the twin factors of temperature and salinity play key roles in the buoyancy loss process responsible for driving the overturning circulation, warming and freshening of surface waters may prevent the sinking process that drives NADW formation from taking place. Consequently, the climate and oceanographic research community is devoting considerable attention towards determining



whether the AMOC is currently weakening or at risk of weakening or even collapse in response to future warming beyond a

certain threshold.

### 2.1.2 Evidence for a weakening AMOC remains uncertain but warrants concern

A recent proxy-based study has presented evidence that the AMOC's strength has fallen over time since the onset of the

industrial age, in a pattern distinct from AMOC strength calculated for the past 1,500 years (Thornalley et al., 2018). However,

this analysis relied upon silt grain size in sediment cores taken offshore of North Carolina as a proxy for bottom current flow

speed—a relatively indirect measure of AMOC. Uncertainty associated with precisely dating the age of silt grains as well as

potential sediment mixing from physical or biological activity represent potential sources of error, although the good agreement

of sediment core-derived AMOC strength with temperature-based reconstructions lend some credibility to the technique.

Furthermore, the study suggests that AMOC weakening may have begun early in – if not prior to – the industrial era, and the

authors note that whether such weakening is associated with anthropogenic forcing remains an open question.


Using historical oceanographic data to constrain past AMOC variability, some researchers have uncovered evidence that

AMOC has declined in magnitude over the 20th century, with a possible acceleration after 1970 followed by a partial recovery

after 1990 (Caesar et al., 2018; Rahmstorf et al., 2015). Other studies seeking to reconstruct AMOC strength using proxies and

reanalysis products since the mid-to-late 20th century have however produced conflicting results showing small to incoherent

trends with time (Frajka-Williams, 2015; Karspeck et al., 2017).

The question of whether the AMOC is weakening is further complicated by a relative paucity of direct measurements of AMOC

strength. The first continuous, basin-wide observations of the AMOC only became available with the installation of the

RAPID-MOCHA instrument array in 2004 (Cunningham et al., 2007). A new monitoring array, OSNAP, was installed in the

subpolar North Atlantic at approximately 52-60°N in 2014 (Lozier et al., 2017). Consequently, direct measurements of the

overturning circulation have only been available for 15 years.

Insights from the RAPID array have also established that AMOC strength varies strongly on interannual, interannual, and

decadal time scales (range of ~4-35 Sverdrups, or million cubic meters per second over one year) (Srokosz and Bryden, 2015).

Such variability complicates efforts to determine a trend over time in AMOC strength from such a limited span of observations.

An apparent AMOC decline over time has been observed since the start of RAPID measurements (Smeed et al., 2018; Srokosz

and Bryden, 2015), but the time series is not yet long enough to rule out the potential impact of natural multidecadal variability.

The first few years of observations from OSNAP have also produced findings that have dramatically changed the current

understanding of the AMOC system. These measurements have revealed that in contrast to the dominant paradigm that most

NADW originates from the Labrador Sea between the Canadian archipelago and the southwest coast of Greenland, the majority

of NADW may in fact be formed in subpolar waters to east of Greenland (Lozier et al., 2019). Current models may be overpredicting the magnitude of Labrador Sea Water (LSW) by up to a factor of five. Glaciers on Greenland's eastern side are currently less active than the island's southwestern ice sheets (Bevis et al., 2019), so this result suggests that AMOC may be
more insulated from meltwater fluxes for the time being than previously thought.

Such findings have called into question aspects of climate models attempting to represent the AMOC, which often do not accurately represent the locations where deep water originates, the magnitude of deep water fluxes, or links between deep water formation and export (Lozier et al., 2019). Yet at the same time, the strong agreement of such models that AMOC
strength is expected to decline under continuing emissions scenarios (Collins et al., 2019; Menary and Wood, 2018) emphasizes that continued efforts to refine existing models and re-assess risks would be prudent.

In the future, as the span of measurements from RAPID and OSNAP lengthens and as additional instrument arrays such as the planned SAMOC monitoring system are commissioned, direct observational evidence for a long-term trend in AMOC strength
may emerge. Furthermore, such data may spur improvements to ocean and climate models and reconstruction techniques that may strengthen their ability to accurately represent the AMOC and predict or reconstruct its future or past strength.

Whereas assessments of whether the AMOC is currently weakening remain subject to some uncertainty, the research community stands largely in agreement that a complete collapse of the AMOC is a low-probability event. The IPCC's Special
Report on Oceans and the Cryosphere in a Changing Climate concluded that an AMOC collapse in the 21st century was "very unlikely," with only one of 27 models producing an AMOC collapse by 2100, and even then only under the worst-case RCP8.5 high-emissions scenario (Collins et al., 2019). Modeling results from a 2016 study concluded that the likelihood of an AMOC collapse under RCP8.5 was 44% by 2290-2300, while AMOC collapse was averted altogether under the RCP4.5 pathway (~2.5C warming) (Bakker et al., 2016).

Furthermore, while the basis for a weakening AMOC may be limited, evidence that the AMOC is increasing in strength is virtually absent, while the null hypothesis that AMOC strength is stable is also uncertain. Intuitively, the expectation that sea surface warming and increased meltwater fluxes from Greenland should weaken the overturning circulation is also consistent with our understanding of the AMOC's underlying mechanisms. Overall, the possibility that the overturning circulation is
currently weakening and may weaken further with continuing warming is sufficiently backed by recent research to justify the degree of past and ongoing attention devoted to this potential climate tipping element.

### 2.1.3 Regional impacts of a weakened AMOC

The consequences of an AMOC collapse would undoubtably be of substantial magnitude and global in scale. A shutdown of the overturning circulation has been modeled to cause significant North Atlantic cooling (3-8C), South Atlantic warming (0-





3C), reduced North Atlantic rainfall (-0.5 to -2 mm/d), and a southward shift of the tropical rain belt (~10 degrees latitude)

(Jackson                  et                  al.,                  2015).

In contrast to total collapse, however, weakening of the AMOC results in more moderate impacts. Modeling a freshwater-induced reduction in AMOC strength from 16 to 9 Sverdrups ($1 \text{ Sv} = 10^6 \text{ m}^3 \text{ s}^{-1}$) yielded North Atlantic subpolar gyre cooling

of 2C (Haarsma et al., 2015). It is worth noting that the results from (Haarsma et al., 2015) stem from modeled weakening of the AMOC somewhat in excess of what IPCC CMIP5 models predict: a decline on the order of $-5.5 \pm 2.7$ Sv by 2100 under the high-end RCP8.5 scenario (Figure 1) (Collins et al., 2019). Considerable paleoclimate evidence supports the existence of linkages between AMOC weakening and Northern Hemisphere cooling that would be accompanied by Southern Hemisphere warming, as reviewed by (Lynch-Stieglitz, 2017).


Changes in the AMOC may also alter storm tracks and intensities in the N. Atlantic (Gastineau et al., 2016) as well as Arctic sea ice patterns (Delworth and Zeng, 2016; Yeager et al., 2015). Evidence of sea level changes along parts of the eastern coast of North America linked to AMOC variability (Ezer et al., 2013; Goddard et al., 2015) also suggests that a weakening AMOC may have implications for adaptation to sea level rise in this region. Climate modelling supports a potential for AMOC

weakening to drive accelerated sea level rise in the North Atlantic basin (Levermann et al., 2005), with a follow-up study predicting meaningful regional sea level impacts even under lower emissions scenarios (Schleussner et al., 2011). Some of the most acute AMOC-induced impacts for human communities include substantial potential reductions in rainfall in the Sahel region, negatively impacting crop yields (Defrance et al., 2017). Other global-scale interactions with the Pacific Ocean and the Antarctic remain uncertain but are the subject of continuing research (Collins et al., 2019).


Slowdown of the AMOC is also theorized to represent a positive feedback on the carbon cycle due to reduced carbon uptake and possible wetland methane-related feedbacks (Parsons et al., 2014). The possibility also exists that an AMOC slowdown may interact with other potential tipping elements within the climate system, such as the potential for Amazon forest dieback or the acceleration of ice loss from the West Antarctic Ice Sheet (Steffen et al., 2018).


Finally, defining particular temperature thresholds expected to contribute to weakening of the overturning circulation also represents a challenge. (Hoegh-Guldberg et al., 2018) determines a higher likelihood of more intense weakening for >2C of warming based on model predictions. Given that substantial loss of the Greenland Ice Sheet is also more likely to occur than not beyond a 2C warming threshold (Pattyn et al., 2018), with important implications for buoyancy dynamics in deep water

formation regions, the IPCC's assessment of a 2C threshold seems a plausible lower bound above which the risks of significant weakening of the AMOC increase. However, we stress that the current ability of models to accurately represent the AMOC and predict its response to climate change remains low. Further improvements to our understanding of the AMOC will prove crucial for better constraining potential outcomes.



## 2.2 Large-scale release of methane from destabilization of marine methane hydrate deposits

**2.2.1 Background**

Methane ($CH_4$) is a potent greenhouse gas whose global warming potential has been revised slightly upwards over the last few decades to a current figure of 34 $CO_2$e ($CO_2$ equivalents) over a 100-year time horizon, and 86 $CO_2$e based on a 20-year time horizon (IPCC, 2013). The majority of atmospheric methane is biological in origin, often the product of methane-emitting bacteria undergoing respiration under low or no-oxygen conditions (Dean et al., 2018). Wetlands, inundated rice fields, and
gases released from the gastrointestinal tract of ruminant herbivores like cows, for instance, all represent significant fluxes of methane globally. The remainder of atmospheric methane is anthropogenic in origin, released by activities such as oil, gas, and coal extraction and production (Table 2).

Significant deposits of frozen methane are currently sequestered in ocean sediments as methane ice, also termed methane
clathrates or methane hydrates. Hydrate deposits accumulated over long geologic timescales as methane released from microbial decomposition of organic matter in ocean sediments (Claypool and Kvenvolden, 1983) or generated from thermogenic processes (Sassen et al., 1999) froze under the low temperatures and high pressures of the deep ocean. Typically, water depths of greater than 500m are necessary for methane hydrates to form in sediment; yet in colder Arctic waters, only around 300m in depth is required (Ruppel and Kessler, 2017).

Whether abrupt or gradual, a large-scale release of methane from hydrate deposits ocean sediments would have implications for global climate, possibly representing a significant positive climate feedback that would exacerbate warming. As a result of being positioned at shallower depths, Arctic methane hydrate deposits are thought to be the most vulnerable pool of marine methane hydrates to warming-induced thaw. With the Arctic Ocean containing a large percentage of the planet's total
continental shelf area, a significant inventory of marine hydrate (~116 Gt C) exists across this basin at relatively modest depths (Kretschmer et al., 2015). Consequently, the scientific community is actively assessing the potential for a large-scale thaw and release of methane hydrate from the Arctic seafloor should increasing sediment temperatures destabilize hydrate formations.

Methane hydrate release will likely manifest as a gradual flux of methane to the atmosphere over long timeframes, due to the
dynamics of sediment column warming (Archer et al., 2009; Ruppel, 2011; Ruppel and Kessler, 2017). Slow release carries important implications for warming, as methane release from hydrate deposits would take place on long timescales relative to atmospheric methane's rapid decay (~10 years) to carbon dioxide, a far less-powerful greenhouse gas on a molecule-for-molecule basis. Furthermore, several mechanisms in both sediment and overlying waters will act as powerful sinks that consume rising methane from thawing deposits, preventing much of the destabilized methane from reaching the atmosphere
(Figure 2). Consequently, the methane hydrate feedback represents a lower threat than popularly imagined, although it remains a climate feedback well worth further attention and study.



### 2.2.2 Methane hydrate dissociation would be a "slow tipping point", with correspondingly gradual impacts

Estimates of total carbon contained within marine methane hydrates globally vary widely, but have largely fallen since the 1990s as important assumptions used in early estimates were overturned by field sampling (Table 3) (Ruppel and Kessler, 2017). Most recent calculations range between ~1100 Gt C (Kretschmer et al., 2015) and ~12,400 Gt C (Dickens, 2011), with a number of studies converging on values towards the lower end of this range (<2000 Gt C) (Archer et al., 2009; Boswell and Collett, 2011; Milkov, 2004; Piñero et al., 2013). As much of the methane hydrate carbon pool globally is located at deeper depths in bottom ocean sediments with higher pressures, large amounts of this carbon (95%) are not considered to be vulnerable to dissociation or release due to projected ocean warming (Kretschmer et al., 2015; Ruppel, 2011). At the same time, however, methane clathrates co-occur with free methane in marine sediments, and within vulnerable regions the quantity of free methane at risk of release may itself approach two-thirds of the local methane hydrate pool (Hornbach et al., 2004).

First, a significant time lag separates atmospheric warming due to climate change and the much longer timescales required for transport and diffusion of heat anomalies into the ocean and sediment. As sediment warming is required for methane hydrate instability, dissociation may not be initiated until centuries to millennia after the requisite warming spike (Archer et al., 2009; Kretschmer et al., 2015; Ruppel, 2011). For deep ocean sediments, tens of millennia might be required for the methane hydrate zone to begin appreciably warming, let alone for hydrate to begin dissociating (Archer et al., 2009; Ruppel, 2011). This factor does not preclude eventual significant release of carbon from methane hydrate, but does mean that this climate feedback occurs with a far more substantial delay than is often acknowledged in mainstream reporting.

A number of further factors likely ensure that future methane release from hydrate deposits will occur as a gradual flux to the atmosphere rather than an abrupt spike. In particular, a number of physical mechanisms will act to greatly slow down the rate of methane escaping from sediments into the water column. Hydrate dissociation is an endothermic process (heat is consumed by the thaw process), limiting the rate at which methane hydrates respond to increased heat flux (Circone et al., 2005; Gupta et al., 2008). Hydrate deposits are also generally expected to thaw first at deeper depths in the sediment column. This is because the stability of frozen methane hydrates is greatest at the seafloor, with hydrates becoming less stable as sediment temperature increases with depth due to geothermal heating from below (Zatsepina and Buffett, 1998). Consequently, a "cold trap" exists for destabilized methane bubbling upwards to shallower, colder depths in sediment, encouraging refreezing of the escaping methane (National Research Council, 2013).

Physical barriers also exist that can significantly slow or halt the release of methane from sediments. These include factors such as low permeability and barriers to upward migration. Researchers have proposed a critical bubble volume threshold, where thawed gaseous methane can only escape the sediment layer after methane has coalesced into bubbles that take up a sufficient percentage of sediment pore space (Archer et al., 2009). This strongly affects the total amount of methane hydrate





released in models in response to warming, with only 35 Gt of C released after 3°C of warming for a critical bubble volume
        of 10%, while 940 Gt of C was released for the same amount of warming under a critical bubble volume of 2.5% (Archer et
        al., 2009).

        At the same time, other physical features such as faults may facilitate the delivery of released methane to the water column.
Abrupt physical events such as submarine landslides could also destabilize hydrate deposits, although even the large Storegga
        landslide (8150 years ago, 5 Gt C released) did not release sufficient quantities of methane to measurably affect climate
        (Archer, 2007). Such considerations as well as significant limitations of the bubble volume threshold and other important
        parameters considerably limit the ability of models to confidently predict methane release in response to warming.

Current models also do not explicitly account for a number of chemical, physical, and biological methane sinks within both
        sediments and ocean waters. Microbial activity within the sediment column can consume a significant percentage of escaping
        methane gas (variable by site but up to 80-90%) (Reeburgh, 2007). This biological sink would be expected to intensify slightly
        with increasing sediment temperatures thanks to the favorable effect of temperature upon bacterial metabolic processes, acting
        as a negative feedback. Finally, significant quantities of released methane would likely dissolve and be oxidized to $CO_2$ within
ocean waters (Kessler et al., 2011; McGinnis et al., 2006; National Research Council, 2013). Transformation of methane to
        $CO_2$ represents a tradeoff, where methane results in larger immediate climate impacts, whereas $CO_2$ lasts in the atmosphere
        for a much longer residence period but exerts a far lower molecule-for-molecule greenhouse forcing. Nevertheless, a slow
        release of $CO_2$ to the atmosphere represents both the more likely and the more preferable outcome as opposed to a large, abrupt
        release of methane from thawed gas hydrates. For 50% of methane within a bubble escaping from the seafloor to reach the
atmosphere from a depth of 100m, the bubble must be two centimeters in diameter, with even larger bubbles required to drive
        substantial methane fluxes from deeper depths (McGinnis et al., 2006). On top of this physical sink, microbial activity within
        the water column will also act to oxidize escaping methane (Elliott et al., 2011; Mau et al., 2013; Valentine et al., 2001).
        Overall, the combined effect of such sinks will likely greatly limit the potential magnitude and rate of any atmospheric release
        of methane from hydrate deposits.


        A number of older modeling analyses had produced much larger climate impacts resulting from methane hydrate release under
        warming scenarios. However, much of this earlier work (Gornitz and Fung, 1994; Harvey and Huang, 1995) significantly
        overestimated the available methane hydrate in marine sediments and/or oversimplified the propagation of heat through the
        ocean and sediment, while omitting important methane sinks in sediment and the water column, as reviewed by (National
Research Council, 2013; Ruppel and Kessler, 2017).

        More recent modeling analyses suggest that the climate impacts of methane hydrate release would be gradual and moderate if
        not minimal. For a 3°C level of warming, one study calculated an additional 0.4-0.5°C temperature increase over multiple





millennia as a result of methane hydrate destabilization of 940 Gt C, if assuming a low critical bubble volume threshold of
2.5% (Archer et al., 2009). However, the quantity of carbon released in response to warming is overwhelmingly driven by the
choice of bubble threshold, making modeled climate impacts highly uncertain. A higher bubble threshold led to insignificant
quantities of released carbon for negligible climate impact. Further, for a total anthropogenic $CO_2$ forcing of 1,000 Gt C
resulting in 2°C of warming, methane hydrate dissolution was modeled to contribute less than an additional 0.5°C warming
even assuming that all dissociated methane reached the atmosphere. In contrast, an assumption of total conversion of escaping
methane to $CO_2$ only extended the duration of 2°C warming, without producing an additional temperature increase (Archer et
al., 2009). Due to the slow thermodynamics of sediment heating, overall methane hydrate-derived fluxes thus represent gradual
long-term additions of methane and carbon dioxide to the atmosphere as opposed to threatening an abrupt methane release.

Present-day and near-future methane hydrate fluxes are not thought to represent climatically significant greenhouse forcings.
A near-term modeling study (Kretschmer et al., 2015) determined that relative to current *annual* anthropogenic methane
emissions of 335 Mt C per year, warming caused by doubling $CO_2$ concentrations over the 21st century would release a *total*
quantity of just 473 Mt of methane from destabilized hydrates over a 100-year period. A more recent modeling analysis
similarly concluded that the quantity of methane within shallower, more vulnerable hydrate deposits and the likely rates of
dissociation were too low to impart a significant climate impact within the next few centuries under current climate change
(Mestdagh et al., 2017). Although the last deglaciation does not represent a perfect analogue to contemporary warming,
analysis of atmospheric concentrations from 18,000 – 8,000 years ago suggest that release of ancient methane from permafrost
and seafloor hydrates in response to past warming may have been small (<19 Mt $CH_4$/yr) (Dyonisius et al., 2020; Petrenko et
al., 2017).

In conclusion, while levels of warming exist beyond which large quantities of methane in hydrate deposits may eventually
become destabilized, numerous physical, thermodynamic, chemical, and biological factors combine to substantially limit the
rate at which methane might escape to the atmosphere. For more moderate warming of ~2°C, methane hydrates might well
exert a negligible overall impact. Methane hydrate dissociation would additionally take place on extremely long timescales of
millennia, rather than over abrupt timescales that would produce an acute warming spike. Consequently, in relation to other
tipping elements covered within this report, marine methane hydrates represent a relatively minor climate threat.

## 2.3 Multi-meter sea level rise from loss of major Greenland and Antarctic ice sheets

### 2.3.1 Background

Currently, the majority of sea level rise (~3.5 mm/yr) (Global Sea Level Budget Group, 2018) is driven by thermal expansion
of ocean waters in response to rising temperatures (~1.40 mm/yr) as well as water inputs from melting glaciers (~0.61 mm/yr)
outside of the major Greenland and Antarctic ice sheets. The Greenland Ice Sheet (GIS) and the Antarctic Ice Sheet (AIS) have





been historically responsible for relatively little sea level rise to date, but their contribution to sea level rise has recently accelerated (currently ~1.20 mm/yr - (Bamber et al., 2018; Oppenheimer et al., 2019; The IMBIE team, 2018)), increasing by 700% relative to the 1992-2001 period (Bamber et al., 2018; Meredith et al., 2019; The IMBIE team, 2018). With continued warming, however, the potential for rapid acceleration of sea level rise due to ice loss from these two regions poses a large

threat to coastal regions globally.

The GIS and the AIS would drive multi-meter increases in sea level over centuries in the event of significant ice sheet loss (Oppenheimer et al., 2019). At the same time, an increasing understanding of ice sheet dynamics has heightened worries that air and sea temperature increases may initiate feedback mechanisms leading to uncontrolled, irreversible ice sheet collapse

over centuries (Pattyn et al., 2018). Climate change is expected to precipitate loss of ice sheets in sequence, with collapse of the GIS and large-scale losses from the West Antarctic Ice Sheet (WAIS) occurring at lower levels of climate forcing, followed by a further acceleration in rising sea level as vulnerable portions of the East Antarctic Ice Sheet (EAIS) increase their currently small contribution to sea levels under continued warming (Meredith et al., 2019). Given the high concentration of vulnerable human populations and infrastructure in coastal regions worldwide, the risk of crossing tipping thresholds for major polar ice

sheets represents a particularly high-priority topic in terms of research, planning, and climate mitigation.

**2.3.2 Powerful feedback mechanisms suggest a real possibility for irreversible ice sheet collapse**

Major ice sheet processes have dominatingly influenced sea level patterns over the geologic past, driving many meters of sea level change in response to shifting temperatures (Cook et al., 2013; Dutton et al., 2015). The Greenland (GIS), West Antarctic (WAIS), and East Antarctic (EAIS) ice sheets contain tremendous quantities of ice. Full loss of the GIS would raise global

mean sea level by 7 m (Letreguilly et al., 1991); collapse of marine-terminating glaciers of the WAIS would yield an increase of 3.4 m (Fretwell et al., 2013). While the EAIS may only begin to lose significant ice mass for higher thresholds of warming, vulnerable marine-terminating basins of the EAIS contain sufficient mass to potentially raise sea levels globally by up to 19.2 m (Fretwell et al., 2013). However, the stability of large portions of the EAIS may depend on the integrity of relatively small volumes of ice at the margins of features such as the Wilkes marine ice sheet, which could impart tipping behaviour leading

to large-scale ice loss over centuries (Mengel and Levermann, 2014).

The magnitude of future sea level rise resulting from ice sheet loss depends on climatic and topographic factors as well as differences in glacier characteristics that may trigger ice sheet destabilization at different temperature thresholds for different ice sheet regions. The GIS and WAIS are less insulated from current and projected climate change and will likely reach key

thresholds first, while the EAIS only responds to higher intensities of warming. Significant ice loss is already occurring for both the GIS and WAIS in the present day, with an ongoing sea level rise contribution of 1.20 mm/yr (Bamber et al., 2018; Oppenheimer et al., 2019; The IMBIE team, 2018). The EAIS is currently at a mass balance thanks to increased snowfall and



precipitation, although subject to considerable temporal variability and uncertainty (Bamber et al., 2018; Boening et al., 2012; Martin-Español et al., 2017; Velicogna et al., 2014).


Ice sheets on the AIS are typically in direct contact with the ocean at their edges. Consequently, such marine-terminating glaciers are at risk of mass loss from processes that result from both oceanic warming (Shepherd et al., 2004) as well as atmospheric warming (Figure 3) (DeConto and Pollard, 2016).

First, marine-terminating ice on reverse slopes, where height decreases with further distance inland, is subject to a positive feedback in which the rate at which the glacier flows out to sea increases as the ice retreats (Schoof, 2007). On reverse slopes, the elevation of the bed falls with increasing distance inland, increasing the height of the glacier above the grounding line - the point beneath the glacier where the ice sheet ceases contact with the bedrock below. As the height of the glacier above the grounding line increases, its seaward movement accelerates. This process of runaway ice loss is called Marine Ice Shelf

Instability (MISI) (Thomas and Bentley, 1978; Weertman, 1974).

Second, warmer waters and atmospheric heating also cause overlying ice shelves to shrink. In the event that ice shelves disappear completely, a feedback mechanism known as Marine Ice Cliff Instability (MICI) may trigger at locations where the height of the cliffs at an exposed ice sheet's edge exceeds 90 meters. At such locations, the shear strength of ice is insufficient

to maintain cliff structure against the longitudinal stress at the cliff face, causing progressive collapse via iceberg calving (Bassis and Jacobs, 2013; Bassis and Walker, 2012). This phenomenon is anticipated to represent a severe risk when the total height of the ice shelf above the underwater grounding line exceeds 800 meters. Since the thickness of ice shelves often increases further inland, MICI would represent a strong positive feedback for ice sheet loss. Furthermore, unlike MISI, MICI may be capable of occurring on flat or forward slopes so long as the ice cliff threshold is exceeded (Pollard et al., 2015). As

ice shelves also provide a supportive "buttressing" effect that opposes and slows the rate of ice flux to sea, loss of ice shelf mass itself also helps speed the glacier's progress towards the ocean (Schoof, 2007).

While not all vulnerable Antarctic glaciers terminate on reverse slopes and are currently thought to be undergoing MISI today, several major basins are currently retreating thanks to processes that may indicate MISI dynamics (Favier et al., 2014; Joughin

et al., 2014; Rignot et al., 2014). In this process, warm subsurface waters cause melt beneath ice shelves, resulting in inland retreat of the grounding line (Shepherd et al., 2004). Should grounding lines retreat beyond forward slopes and onto reverse-sloping topography, as is the potential case for many major Antarctic basins (Ross et al., 2012), the process of MISI would begin rapidly accelerating ice loss from many major glaciers. As for MICI, no evidence yet suggests that MICI is occurring currently on Antarctic ice sheets, and a knowledge gap remains concerning the potential importance of MICI for projections

of future ice loss (Meredith et al., 2019). Overall, projecting ice sheet losses from MICI remains a contentious topic given its



significant impact on near-term and long-term sea level rise estimates, and so studies including MICI arguably represent high-end scenarios based on ice cliff collapse mechanisms not yet fully validated by field observations (Edwards et al., 2019).

In contrast to the Antarctic sheets, Greenland glaciers typically terminate on land prior to reaching the sea. Consequently, surface melt in response to rising air temperatures that greatly outpaces new ice sheet accumulation represents the primary cause of ongoing and projected ice loss for the GIS (Bevis et al., 2019). Surface ablation and snow melt can result in the formation of meltwater pools or exposure of darker ice that reduce surface reflectivity, further intensifying melt through a melt-albedo feedback (MacFerrin et al., 2019; Ryan et al., 2019). Refreezing of meltwater also warms adjacent ice. At the same time, meltwater can percolate through holes and crevices through to the foot of the glacier, lubricating the glacier where

it meets the bedrock and accelerating the speed of ice flow towards the sea (Bell, 2008). Warming glaciers also exhibit lower viscosity, further increasing their rate of flow. As glaciers move seawards, their altitude falls, bringing them into lower, warmer layers of air - another positive feedback known as the melt-elevation feedback or height-mass-balance feedback (Huybrechts and De Wolde, 1999).

Surface mass loss via atmospheric heating represents a major threat to Antarctic glaciers as well. Atmospheric warming may threaten smaller ice shelves and tongues that currently help stabilize glaciers (Schoof, 2007), thus increasing the threat MISI and MICI pose to predominantly marine-terminating Antarctic ice sheets over the coming centuries. Some recent modeling work has also predicted that the majority of future sea level rise resulting from Antarctic ice mass loss may occur thanks to atmospheric warming rather than ocean heating, due to amplification of MISI/MICI dynamics in addition to surface melt and

associated meltwater feedbacks (DeConto and Pollard, 2016). However, considerable uncertainty continues to surround future projections of MISI and MICI feedbacks, particularly the latter (Edwards et al., 2019).

At the same time, a handful of negative feedbacks may somewhat slow rates of ice sheet loss. For the AIS, for instance, increasing ice congestion in restricted bays may provide added resistance, buttressing some glaciers and inhibiting ice

movement out to sea (Seneca Lindsey and Dupont, 2012). Increased snowfall over portions of Antarctic ice have also provided mass inputs that offset a portion of ice loss (Boening et al., 2012). These factors only represent relatively minor influences upon the ice sheets' mass balance as a whole. More significantly, uplift of the land surface beneath retreating ice sheets as the weight of the overlying ice is reduced has been suggested to act much faster than previously assumed, potentially proving a strong negative feedback (Barletta et al., 2018).


Overall, the balance of current trends acts to drive the GIS and WAIS increasingly towards feedbacks leading to sustained, irreversible ice sheet loss. Evidence from field observations, surveys of glaciers and bed topography, and modeling analyses strongly support the conclusion that warming beyond key thresholds will initiate ice sheet collapse. Sea level rise in response





to loss of the Greenland ice sheet and major portions of the Antarctic ice sheet therefore represents an extreme and credible
threat to coastal regions worldwide.

### 2.3.3 Commitment to loss of the Greenland and West Antarctic ice sheets yields rapid, multi-meter sea level rise over centuries and millennia

Ongoing field and satellite studies have confirmed or even revised upward earlier findings that vulnerable ice sheets are already
feeling the effects of climate change. Mass loss from Greenland has continued, with the pace of its contribution to sea level
having accelerated over time in recent decades (McMillan et al., 2016). Individual ice shelves on the West Antarctic Peninsula,
the Southern Antarctic Peninsula, and in the Bellingshausen and Amundsen seas are experiencing losses via behavior
potentially consistent with irreversible onset of MISI (Paolo et al., 2015; Rignot et al., 2014; Wouters et al., 2015). The IPCC,
however, has assessed the current record of field work as too limited to conclusively determine whether irreversible retreat
was underway for these areas (Meredith et al., 2019). Recent observational evidence also suggests that the EAIS may be
worryingly more vulnerable to ocean heating than assumed previously (Silvano et al., 2016). New research has also pointed to
a potentially longer-term natural trend of ice sheet thinning for parts of the EAIS over the past 300 years, potentially increasing
the region's vulnerability to future ice loss (Dickens et al., 2019).

Initiation of irreversible ice sheet collapse would bring about serious long-term consequences for global sea level. For summer
warming of more than 2°C, the GIS has been assessed as more likely to reach an irreversible tipping point than not (Pattyn et
al., 2018). Local annual mean warming of 3°C could be attained this century and would eliminate nearly all Greenland glaciers,
raising seas by 7 m over approximately 1,000 years (Gregory et al., 2004; Huybrechts et al., 1991). Under a worst-case RCP8.5
emissions scenario incorporating a strong MICI feedback, modelers have calculated a potential for Antarctic ice loss to
contribute over a meter of sea level rise by 2100, with an eventual sea level increase of 12.3 m by 2500 (DeConto and Pollard,
2016). The moderate RCP4.5 emissions pathway results in nearly complete WAIS collapse within 500 years, driven strongly
by intense retreat of the Thwaites Glacier, yielding 32 cm of sea level rise by 2100 and an eventual total Antarctic contribution
of 5 m higher seas by 2500. As noted previously, however, such estimates represent high-end worst-case figures reliant on
heavy contributions of MICI to ice sheet loss, a yet-uncertain scenario (Edwards et al., 2019).

Modeling results not incorporating MICI dynamics, in contrast, produce considerably more conservative sea level changes,
with an Antarctic contribution to sea level rise of 30-40 cm of sea level rise under the high-emissions RCP8.5 pathway
(Edwards et al., 2019) and 30 cm under the A1B emissions scenario, which is approximately comparable to RCP6.0 (Ritz et
al., 2015). Overall, total predicted sea level rise by 2100 generally ranges between 0.61 and 1.10 meters for RCP8.5
(Oppenheimer et al., 2019). Under the high-emissions RCP8.5 forcing, modelers assessed a cumulative contribution from
Greenland and Antarctica of 25 cm of sea level rise by 2100 (Golledge et al., 2019).



Antarctic ice sheet collapse would be irreversible absent substantial cooling that would permit reformation of buttressing ice shelves. Otherwise, retreat would continue until topography that halts MISI/MICI is reached. Greenland melt could theoretically be reversible in the event of immediate cooling. To fully counteract heat buildup in Greenland glaciers, however,
a greater magnitude of cooling than total warming to date would be necessary to stabilize the surface mass balance.

Timescales of collapse depend strongly on bed topography of individual basins and can require centuries to millennia. The Thwaites glacier, for instance, may already have reached the early stages of irreversible mass loss, although its rate of reduction will likely remain moderate over the 21st century, with collapse potentially occurring over a period of 200-1000 years (Joughin
et al., 2014). Worst-case models incorporating MICI, however, have yielded considerably faster rates of collapse, with simulations producing complete deglaciation of the WAIS in a single century after reaching critical thresholds (DeConto and Pollard, 2016). Other estimates of WAIS collapse have tended towards longer timeframes of up to 400 years (Cornford et al., 2015; Golledge et al., 2015). EAIS basins may take more than a century under warming scenarios to begin shedding significant mass, with multi-meter contributions to sea level rise over the course of several millennia (Golledge et al., 2015). As mentioned
above, reduction of the Greenland ice sheet will likely require a millennium. Consequently, ice sheet collapse cannot generally be considered an abrupt phenomenon, although sustained high rates of sea level rise (>1 cm/yr by 2200, with further acceleration to up to a couple centimeters per year beyond) may seriously strain coastal adaptation efforts (Oppenheimer et al., 2019).

At the same time, models indicate that strong climate mitigation may avert significant fractions of potential sea level rise and prevent ice sheet collapse across large regions. Under RCP2.6, DeConto and Pollard's modeling experiment produced virtually no net Antarctic contribution to global mean sea level by 2100, with only a 20 cm rise by 2500. Another study found that the RCP2.6 scenario prevented collapse of the WAIS regardless of uncertainties in model parameters (Bulthuis et al., 2019). Overall, the IPCC's synthesis of scientific findings suggests that total sea level rise by 2100 under RCP2.6 could be contained
to a range of just 0.29 to 0.59 m (Oppenheimer et al., 2019). Although significant uncertainties remain regarding the precise temperature thresholds that could trigger ice sheet collapse, research to date strongly suggests that climate mitigation at a level approaching the RCP2.6 emissions pathway significantly lowers risks of initiating ice sheet instability.

## 2.4 Climate feedbacks from Arctic permafrost thaw and decomposition

### 2.4.1 Background

Permafrost is soil that remains at freezing temperatures for two or more consecutive years, with many permafrost regions worldwide having remained predominantly frozen over the past several thousand years. Due to minimal rates of decomposition at such low temperatures, considerable quantities of organic matter accumulate and become incorporated in these frozen soils. Consequently, sizable stocks of carbon lie contained within permafrost soils globally. Total organic carbon content of all





permafrost soils in the Northern Hemisphere are assessed to range between 1460 - 1600 Gt C, nearly twice the amount of carbon currently in the atmosphere (Schuur et al., 2018). On a worldwide scale, permafrost carbon represents about one-third of all global soil carbon within the upper 3m (Jobbágy and Jackson, 2000; Schuur et al., 2015).

Naturally, given the expectation that rising global temperatures will thaw an increasing volume of permafrost over time, the climate community has devoted serious efforts towards assessing whether or not widespread permafrost melt could liberate
climatically-significant amounts of methane and carbon dioxide as previously-frozen organic matter begins to decompose.

In popular climate discourse, commentators have sometimes characterized this climate-permafrost carbon feedback as a tipping point which, once crossed, would trigger abrupt and severe warming (Doucleff, 2018; Yoder, 2019). However, such characterizations exaggerate the speed of the permafrost carbon feedback. Even abrupt permafrost thaw processes
(thermokarst), while occurring on fast timescales locally (days to decades), will cumulatively contribute towards climate change over a century or more rather than in a single event (Turetsky et al., 2020, 2019). Overall, the majority of permafrost carbon release will only take place after 2100 (McGuire et al., 2018; Turetsky et al., 2020), as gradual permafrost thaw in particular occurs relatively slowly on a time scale of centuries (McGuire et al., 2018; Schneider Von Deimling et al., 2015). Even beyond 2100, methane fluxes from thermokarst will likely induce only a modest to moderate climate forcing (average of
65 Mt C /yr under a worst-case RCP8.5 scenario (Turetsky et al., 2020)) relative to anthropogenic methane emissions today (335                                                                    Mt                                                                    C/yr).

Nevertheless, this added input of atmospheric carbon (worst-case estimates equivalent to up to ~400 Gt C by 2300 at an average rate of ~1.3 Gt C / yr) (Turetsky et al., 2019) would be climatically significant, further incentivizing climate change mitigation
to a level that would minimize the impact from this feedback. Remaining uncertainty concerning feedbacks between temperature, permafrost thaw, landscape changes, and the response of microbial communities and larger arctic ecosystems (Turetsky et al., 2019) also leaves considerable error associated with such $CO_2$ and methane emissions estimates (Dean et al., 2018).

At any rate, the sheer size of the permafrost carbon pool itself warrants continued attention and concern to this feedback mechanism. Permafrost thaw also carries important consequences for local communities, infrastructure, and ecosystems ranging from structural damage and loss of life caused by unstable ground to an elevated risk of infectious disease outbreaks at the hands of live pathogens liberated from thawed permafrost (Meredith et al., 2019; Walsh et al., 2018). Models suggest that ambitious climate mitigation efforts would appreciably reduce the extent of global permafrost loss (Chadburn et al., 2017;
McGuire et al., 2018). Consequently, limiting the extent of climate change represents a powerful approach for reducing both the climate risks associated with permafrost thaw as well as impacts upon northern communities.



### 2.4.2 Cumulative impacts of methane or CO2 release from abrupt and gradual permafrost thaw manifest over a century or more

The potential for thawed permafrost to drive widespread release of methane or $CO_2$ to the atmosphere depends upon two
mechanisms - the response of gradual and abrupt permafrost thaw processes to rising temperatures, and microbial decomposition of organic matter that releases carbon dioxide and generates methane (methanogenesis). Extensive documentation points to an ongoing rise in permafrost temperatures globally, Record high temperatures at depths 10-20m within permafrost are now 2-3°C higher than those observed 30 years before (Figure 4) (Romanovsky et al., 2017a). Permafrost within the colder continuous permafrost zones of the far north warmed by 0.29 +/- 0.15°C between 2007 and 2016
(Biskaborn et al., 2019).

Overall, a strong possibility exists that a sizable fraction of permafrost area globally may thaw by end-of-century, with estimates of the extent of loss ranging very widely from 5% to 70% (Koven et al., 2015; Lawrence et al., 2012; McGuire et al., 2018; Schaefer et al., 2011; Schuur et al., 2015; Schuur and Abbott, 2011; Wisser et al., 2011). Ultimately, timescales for
gradual deep permafrost thaw via thickening of the active permafrost layer are measured in centuries, meaning that some organic      carbon      stored      at      deeper      depths      may      remain      stable      for      some      time.

However, processes driving abrupt permafrost thaw are increasingly anticipated to contribute meaningfully to global climate on timescales more immediate than gradual thaw (Turetsky et al., 2020). These thermokarst processes - the collective term for
slumping, subsidence, rapid erosion, and similar phenomena - lead to more abrupt exposure and thaw of permafrost on time scales of days to years (Abbott and Jones, 2015). Associated with ~20% of the permafrost region where permafrost is discontinuous and subsurface ground ice is more prevalent (Olefeldt et al., 2016), thermokarst is often associated with melting ground ice that triggers sudden landslides, crevasses, or subsidence. Such events can destabilize and expose permafrost soils to depths of dozens of meters, allowing for more rapid decomposition of belowground organic matter (Abbott et al., 2015).
Carbon mobilized by thermokarst events in particularly old Yedoma permafrost soils (>21000 years old) has demonstrated rapid rates of biodegradation, underlining the potential for significant carbon release from thermokarst features (Vonk et al., 2013).

Natural methane stores trapped under currently-frozen soil layers may also be released thanks to permafrost thaw, although
this source is not currently anticipated to be climatically significant (Petrenko et al., 2017). Higher temperatures in permafrost regions could boost the frequency of wildfires, both releasing significant carbon and transferring heat to deeper permafrost layers (Goetz et al., 2007; Mack et al., 2011; Randerson et al., 2006; Walker et al., 2019). Strong evidence points towards an increasing frequency and severity of wildfires throughout the arctic and boreal north (Flannigan et al., 2009; Hanes et al., 2019; Kasischke                         and                    Turetsky,               2006).



Summed up over the entirety of northern permafrost regions, the potential of such factors to significantly influence overall methane release remains relatively poorly constrained. Analysis of historic methane concentrations during periods of past warming have suggested that release of methane from permafrost and marine gas hydrates during the last deglaciation was likely small (<19 Mt CH$_4$/yr), although the present-day Arctic is warming at a much faster rate than that previously experienced

in the region (Dyonisius et al., 2020; Petrenko et al., 2017). Other mechanisms may also potentially mitigate or compensate for permafrost carbon dioxide release in the far north, such as increased plant growth and new accumulation of soil carbon in response to warming temperatures and thawed surface permafrost (Koven et al., 2015; Turetsky et al., 2007, 2019).

Once thawed, rates of CO$_2$ and methane release from permafrost are strongly controlled by a complex network of different

microbes with varying metabolic strategies and responses to increasing temperatures. Microbial methane production is primarily active under low-oxygen conditions deep within soil layers or underwater (Abbott et al., 2015), as more efficient metabolic pathways are strongly favored in the presence of oxygen. Consequently, higher soil water content and inundated conditions for thawed permafrost would favor increased methane generation while drier environments would result in reduced methane production (Blanc-Betes et al., 2016; Walter et al., 2001; Zhuang et al., 2004). Additionally, methanogenic bacteria

also exhibit a strong temperature response, with experiments demonstrating a greater than hundred-fold increase in methane production for a temperature rise of 10°C (Metje and Frenzel, 2007; Rivkina et al., 2004; Tveit et al., 2015). Increasing evidence is also coming to light that emissions of nitrous oxide - another potent greenhouse gas - from permafrost may be non-negligible, contrary to previous assumptions (Wilkerson et al., 2019).

Paralleling the potentially strong biological sinks for methane liberated from seafloor gas hydrates, methane oxidating bacteria may exhibit an ability to consume 58 to 90% of methane produced in northern bogs (Popp et al., 2000). More recent measurements of hydrological flow from thermokarst landscape features have shown some evidence for strong methane sinks with the potential to reduce the methane impact from abrupt thaw processes (Abbott et al., 2015). Such biological methane consumption is also likely to increase in response to temperature, but may not be able to fully keep up with the increase in

methane production, allowing a relatively greater percentage of methane to escape to the atmosphere (van Winden et al., 2012).

Ultimately however, research to date on potential permafrost carbon release suggest that no mechanisms exist at sufficient scale to drive enough methane emissions to dramatically influence global climate within an abrupt time scale of years (Dean

et al., 2018; National Research Council, 2013). Longer-term carbon dioxide release projections for emissions from thawed permafrost by 2100 are quantitatively large albeit uncertain (range of 37-174 Gt C, with two recent estimates of 57 Gt C and 87 Gt C under a worst-case RCP8.5 emissions scenario) (Koven et al., 2015; Meredith et al., 2019; Schneider Von Deimling et                                        al.,                                        2015).



Models for carbon release from gradual permafrost thaw indicate that this climate feedback acts on time scales of several centuries, with more substantial net carbon emissions not occurring until after 2100 (McGuire et al., 2018). Current multi-century model projections continue to demonstrate high uncertainty, with estimates for carbon release by 2300 under a worst-case emissions pathway ranging between 167 Gt C absorbed to 641 Gt C released (mean of 208 Gt C released) (McGuire et al., 2018). Research to date suggests that thermokarst processes could emit an additional 60-100 Gt C by 2300 under the

RCP8.5 scenario, although the corresponding climate impact could be larger depending on the proportion of methane emissions (Turetsky et al., 2020, 2019). Climate mitigation can powerfully moderate the permafrost carbon climate feedback, with the modest-mitigation RCP4.5 scenario potentially minimizing permafrost carbon release to the point where the northern permafrost ecosystem could remain a net carbon sink (McGuire et al., 2018).

Considering all of these factors, permafrost carbon release might be more accurately characterized as a tipping element as opposed to an abruptly-acting tipping point. Research findings indicate that overall, carbon release from permafrost takes place over timescales of a century to several centuries. Modeling under different emissions scenarios also suggests a high ability of ambitious emissions reductions to significantly preserve existing permafrost carbon. Efforts to quantify the impact of permafrost carbon upon global climate have yielded worst-case (RCP8.5) additional temperature increases of ~0.27C by 2100

and ~0.42C by 2300, although these studies do not account for abrupt thaw processes (Burke et al., 2017; Schaefer et al., 2014). These effects pose a serious challenge to ambitious mitigation targets but suggest that permafrost feedbacks are not independently capable of warming global climate by a degree or more within this century (also see Section 2.6).

### 2.4.3 Strength of potential permafrost carbon release impacts and remaining knowledge gaps warrant caution

Permafrost thaw-derived greenhouse gas fluxes do represent climatically-significant quantities of greenhouse gases. A multi-

model mean of 92 Gt C released from gradual thaw by 2100 under the high-emissions RCP 8.5 scenario (Meredith et al., 2019) would represent about a decade's worth of present-day annual carbon emissions (~10 Gt C/yr) (Peters et al., 2019). Thus, even if the permafrost-climate feedback does not represent an extreme warming threat, it nevertheless carries appreciable importance in the context of the wider climate challenge. The permafrost feedback will increasingly contribute to climate change over the course of the coming centuries, while climate mitigation within the next decades will powerfully affect the magnitude of future

climate impacts associated with permafrost processes.

Impacts of permafrost thaw itself also directly affect human communities and infrastructure throughout the northern permafrost region, threatening the integrity of homes, roads, rail lines, and runways as the terrestrial surface deforms and warps in response to thaw of ground ice below. A possibility even exists for permafrost warming to cause pipeline damage resulting in increased

leakage of fugitive methane to the atmosphere, a novel if relatively minor climate feedback (Zhou, 2006). Pathogen release from currently-dormant bacteria and viruses trapped within frozen soils also represents a significant potential human health



hazard (Meredith et al., 2019). Anthrax possibly released by permafrost thaw has already been implicated in disease outbreaks among reindeer of Russia's Yamal Peninsula in 2016 (Walsh et al., 2018).

Widespread permafrost thaw will also result in considerable ecological impacts across the circumpolar north. These will likely include significant changes to vegetation patterns, including possible borealization of tundra as the northern extent of boreal forest begins to encroach into newly-warmer, thawed poleward areas (Gauthier et al., 2015). Such habitat shifts will carry significant implications for flora and fauna across biomes currently characterized by permafrost soils.

Furthermore, significant uncertainties remain in terms of accurately assessing methane sources and sinks at a global scale. Top-down and bottom-up estimates of methane emissions have often arrived at differing values (Saunois et al., 2016). An abrupt halt in increasing global methane concentrations from 2000-2007, for instance, has remained unsatisfactorily explained to the present day (Fletcher and Schaefer, 2019). Such challenges, alongside methodological and technological limitations, have inhibited attempts to confidently scale up local processes such as permafrost thermokarst, lake formation, and microbial
methane production to planetary scales (Nitze et al., 2018; Turetsky et al., 2019).

Finally, the very magnitude of the vast quantities of carbon contained within the global permafrost pool is sufficient to justify continued monitoring and attention of the soils of the arctic north. Permafrost soil carbon loss is irreversible, as permafrost carbon cannot be replenished on time scales of less than thousands to tens of thousands of years. The permafrost carbon stock
is large to the point that even small deviations in its expected behavior in response to climate change may affect long-term total greenhouse forcing. Ongoing assessment to further constrain estimates of future carbon release from thermokarst as well as gradual permafrost thaw will remain an important research priority with key implications for tomorrow's climate.

## 2.5 Large-scale boreal forest ecosystem shifts

### 2.5.1 Background

The northern boreal biome has felt the effects of climate change to a dramatic degree, with decades of observations highlighting rapid regional temperature increases, more frequent occurrence of wildfires, and intensification of pest-driven tree mortality. Near-surface fall and winter air temperatures within the band of latitudes from 70-90°N rose by 1.6°C per decade over the 1989-2008 period, with summer temperatures rising at a rate of 0.5°C per decade (Screen and Simmonds, 2010). The observational record has seen a marked increase in the extent, frequency, and severity of boreal wildfires, with this increasing
trend predicted to continue with further warming (Flannigan et al., 2009; De Groot et al., 2013; Veraverbeke et al., 2017). Warmer temperatures have additionally been linked to acute outbreaks of insects leading to large-scale tree mortality events in Western Canada (Kurz et al., 2008), sparking concern that similar pest invasions could occur more often in the future.



Between the biomass in soil, permafrost, and living and dead vegetation, boreal forests represent a significant pool of terrestrial
organic carbon (30% of global soil carbon) (Mcguire et al., 2009; Turetsky et al., 2019), and constitute 30% of global forest area (Kasischke, 2000). Of this fraction, two-thirds of boreal forest are found within Russia, with Russia's boreal forests estimated to contribute around half (0.6 G C/yr) of the total global terrestrial carbon sink (Dolman et al., 2012; Schaphoff et al., 2013). Recent research has proposed that boreal forest carbon stocks could be underestimated, with updated calculations suggesting that boreal regions hold more terrestrial carbon (Bradshaw and Warkentin, 2015) than tropical areas, which have
been previously suggested to harbor the most carbon globally among all biomes (Pan et al., 2011).

It is important to note that 95% of boreal zone carbon is stored in peatlands and soils (Bradshaw and Warkentin, 2015), and an uncertain proportion of this underground carbon stock overlaps with this report's discussion of permafrost carbon (Section 2.4). Boreal vegetation and permafrost soils also interact with one another (Carpino et al., 2018), complicating attempts to
disentangle potential carbon fluxes associated with changes in the vegetation and soil systems. Regardless, boreal forests make up an important component of the terrestrial and global carbon cycles, with changes in this biome potentially acting as large climate feedbacks.

A growing number of research studies over the past 25 years have provided increasing evidence for the possibility of region-
specific shifts within the circumpolar boreal biome with potentially significant climate implications (Chapin et al., 2005; Foley et al., 1994; Lenton et al., 2008; Scheffer et al., 2012). This hypothesis theorizes that a region-dependent mix of temperature, moisture, and precipitation changes, shifts in wildfire regimes, greater vulnerability to pest insect outbreaks, and a lengthening of the growing season may all combine to drive acute mortality of boreal forest vegetation at the southern margins of boreal zones and its replacement by more open deciduous woodlands and grasslands, as reviewed by (Gauthier et al., 2015). Such a
phenomenon could represent a climatically significant tipping element with the potential to change land surface albedo over large areas and release a considerable pool of carbon to the atmosphere, in addition to altering the strength of an important terrestrial organic carbon sink.

Considerable uncertainties regarding the resilience of boreal forests to climate change and the mechanisms driving potential
ecosystem shifts remain, however, making it difficult to determine the likelihood that this system could act as a tipping element (Schaphoff et al., 2016). For instance, reductions in boreal forest area in southern boreal regions may be accompanied by expansion of boreal ecosystems northwards in response to warmer temperatures, with multiple competing, complex climate impacts (Beck et al., 2011; Foster et al., 2019; Ju and Masek, 2016; Pastick et al., 2019; Pearson et al., 2013). Nevertheless, the vast size of the boreal biome, the near-certainty that the region will undergo imminent changes this century that are large
relative to other parts of the world, and the strong observational evidence for intensification of disturbances to the boreal forest suggest that future trends in boreal ecosystems could play an important role in determining the trajectory of global climate.



### 2.5.2 Boreal forests worldwide are experiencing rapid changes with the potential for northward expansion and southern margin shifts to deciduous forest or grasslands

Observations of ongoing changes in boreal ecosystems worldwide provide abundant evidence of increasing climate impacts to

these regions. As mentioned above, temperatures for the Arctic north are rising at two times or more the global rate of warming (Screen and Simmonds, 2010). Tree mortality across the Russian boreal forest has increased over the late 20th and early 21st centuries (Allen et al., 2010). The same region has also seen a dramatic intensification in fire occurrence, with the fire return interval falling from 101 years in the 19th century to 65 years in the 20th century for larch-dominant forest stands (Kharuk et al., 2008). Forest area burned has correspondingly increased across Siberia based on data from multiple sources (Soja et al.,

2007). Satellite data also highlight decreases in forest productivity across Alaska since 1982 (Beck et al., 2011). With regional warming, Alaska has also begun to see large-scale forest mortality events driven by previously cold-limited spruce beetles, as reviewed in (Soja et al., 2007). The North American boreal forest is also exhibiting an increase in the proportion of deciduous tree cover (Wang et al., 2019).

Yet while boreal forest productivity and tree cover are on the decline at the southern edge of the boreal zone and within interior regions, thirty-year datasets of satellite and observational evidence also point towards ongoing expansion of boreal forests northwards into area previously occupied by tundra thanks to warmer temperatures (Beck et al., 2011; Ju and Masek, 2016; Pastick et al., 2019). Since 1960, the growing season across the boreal zone has lengthened by 3 days/decade (Euskirchen et al., 2006).


The accelerated pace of boreal climatic shifts relative to the rest of the world is likely to continue over the 21st century. Warming of 3-5°C globally by end-of-century would imply mean temperature increases of 7-10°C for large parts of Russia, with regional warming of up to 12°C (Schaphoff et al., 2016). Such warming will likely considerably exacerbate the abovementioned climate impacts upon the boreal environment. For example, one study predicts that Canadian boreal forest

fires may double in annual burned area and increase in frequency by 50% by 2100 (Wotton et al., 2003), while a more recent analysis modeled potential increases in burned area of 29-35% for the Northwest Territories and 46-55% for interior Alaska by 2050-2074 (Veraverbeke et al., 2017).

The rapid pace of such observed and predicted patterns, which in some cases exceed older predictions, raises the possibility

that future change and warming-induced feedbacks within the boreal biome may proceed non-linearly rather than linearly (Foster et al., 2019; Johnstone et al., 2010; Soja et al., 2007). An extensive survey of forest cover across the boreal environment has indicated that intermediate states of landscape tree cover are rare and potentially unstable, suggesting that forested areas may transition to systems with sparse tree cover more abruptly than thought (Scheffer et al., 2012). Shifts towards more prevalent fires potentially play a major role in driving a transition towards more deciduous tree cover (Johnstone et al., 2010).

Paleoclimate evidence of boreal forests responding strongly to temperature changes from natural orbital patterns, coupled with



a strong modeled importance of variability in boreal forest area for overall climate, suggest that large shifts in boreal biome extent could act as a large-scale climate feedback and have done so in the geologic past (Foley et al., 1994).

Shifts in boreal ecosystems, however, are anticipated to be variable for different regions of the boreal zone. Replacement of
boreal forest from its southernmost extents by grasslands and deciduous forest is expected to continue based on the response of vegetation models to climate forcing (Foster et al., 2019; Wang et al., 2019). At the northern boundary of the boreal zone, models conversely predict further expansion of boreal forest into the present-day tundra biome (Foster et al., 2019; Pearson et al., 2013). The extent of wildfires in boreal environments is widely anticipated to increase in the future (Balshi et al., 2009; Kloster et al., 2012; Shuman et al., 2017; Wotton et al., 2017). Together with climate-driven changes to precipitation and soil
moisture, more frequent and intense wildfires may drive declines in boreal forest productivity within the interior of boreal regions (Foster et al., 2019; Wang et al., 2019). Region-specific modeling studies focusing on Alaska (Foster et al., 2019; Mann et al., 2012), Siberia (Shuman et al., 2015), and China's boreal forests (Wu et al., 2017) support the potential for future climate-induced        shifts        in        vegetation        consistent        with        such        patterns.

Ecosystem changes for the boreal forest may take place within decades to a century. A multi-region model analysis found that for large portions of all major boreal regions with the exception of Eastern North America, projected temperature changes under worst-case emissions would alter local climatic conditions to resemble current woodland/shrubland climates by 2090 (Gauthier et al., 2015). (Mann et al., 2012) in fact suggest that a transition to mixed woodlands has already begun for the Alaskan boreal region and is anticipated to reach completion by mid-century. Projections for Siberia also indicate a relatively
rapid shift, with dark and light-needled forest declining from ~60% of the modeled area to ~40% and ~24% for a 720ppm and >800 ppm end-of-century emissions scenario, respectively (Shuman et al., 2015).

However, a notable degree of uncertainty continues to surround the triggering thresholds, extent of change, and climatic impacts of large-scale boreal ecosystem shifts. Polling of expert researchers produces a wide range of global mean temperature
increases relative to pre-industrial temperature (3-4°C) for intensifying boreal forest transitions (Kriegler et al., 2009). Many modeling studies to date have focused heavily on worst-case emissions scenarios like RCP8.5 (Gauthier et al., 2015; Shuman et al., 2015), which may no longer represent most-likely climate trajectories (Hausfather and Peters, 2020). Comparisons of vegetation shifts under multiple climate scenarios suggest that boreal forest changes may be strongly mitigation-dependent, with more limited impacts for more moderate emissions pathways (Lucht et al., 2006). Loss of boreal forest in southern regions
may also be partially compensated by afforestation along the northern boundaries of the boreal biome as warmer, less-frozen tundra gives way to forested landscapes (Scheffer et al., 2012). Existing vegetation models also remain limited in their ability to replicate complex terrestrial ecosystem dynamics and produce confident estimates for changes in biomass carbon (Schaphoff et al., 2016). Finally, the future of boreal forests will also be strongly dependent on human management practices.



### 2.5.3 Although tipping element behavior remains uncertain for boreal ecosystems, climate importance of this system is large even without abrupt shifts

Overall, the picture of boreal forests as a tipping element remains uncertain, but troubling (Table 4). The recent pace of change witnessed in the boreal north clearly points to the potential for large-scale transitions. Projections for such a transition, however, remain based on models that must make simplifying assumptions about complex mechanisms such as precipitation, fire, and soil moisture availability and their effects on needle-leaved trees. These processes as well as competitive interactions between individual trees and species are important factors for modeling the response of boreal regions to climate forcing (Foster et al., 2019). Nevertheless, parallel findings from a growing number of analyses predicting major vegetation shifts across the biome region underline the real possibility for a boreal transition beginning later this century.

One of the most worrying aspects of a potential boreal tipping element involves the difficulty of quantifying the climatic impact of replacement of boreal forests by deciduous woodlands or shrublands. Boreal forests contain 30% or more of global soil carbon, and 95% of organic carbon within boreal forest ecosystems is stored belowground (Bradshaw and Warkentin, 2015; Flannigan et al., 2009). Small-scale processes triggered by warming and permafrost thaw may precipitate the release of some of this large carbon pool via decomposition (Turetsky et al., 2019), while fire and post-fire mortality have the potential to emit considerable carbon from both forest biomass as well as upper soil organic layers (De Groot et al., 2013; Shvidenko et al., 2011). At the same time, projections for potential carbon release from boreal soils are complicated by a high degree of overlap with estimates for permafrost carbon release (Bradshaw and Warkentin, 2015), as the northern boreal and permafrost zones partially coincide. A potential also exists for interactions between permafrost thaw and vegetation shifts. For instance, observations in Canada suggest that permafrost thaw may push boreal landscapes towards wetland-like conditions, resulting in declines in forest cover (Carpino et al., 2018).

Some research from an Alaskan study suggests that after wildfires, a shift in vegetation from needle-leaved trees to deciduous forest would not substantially change the total pool of organic carbon, and would potentially increase the storage lifetime of carbon due to the greater resistance of deciduous biomass to fire and decay (Alexander and Mack, 2016). Apart from carbon, a shift in vegetation from darker coniferous species to lighter deciduous trees would increase regional reflectivity of the southern and interior areas of the boreal zone, acting as a negative feedback on warming (Betts, 2000; Liu et al., 2019; Mykleby et al., 2017; Piao et al., 2020).

As boreal forests expand northwards into current tundra biomes, however, the same research suggests that the darker surface of boreal vegetation – particularly as it overtakes treeless ground previously snow-covered in winter – will act as a positive feedback reinforcing climate change (Betts, 2000; Liu et al., 2019; Mykleby et al., 2017; Piao et al., 2020). This albedo-driven effect will likely overcome any positive benefits from added carbon sequestration. Ultimately, after accounting for all factors, changes across the boreal zone due to a warming climate could act as a net positive climate feedback overall, thanks to the role





of permafrost thaw and wildfires in liberating soil carbon that makes up the majority of stored carbon across this ecosystem.

Overall, calculations of changes to carbon stocks, regional albedo, carbon sinks, and the timescales involved even at local or regional scales remain imprecise and depend upon multiple complex processes and feedbacks (Foster et al., 2019; Shuman et al., 2015). Consequently, estimating the climatic impact of worldwide changes to the boreal biome under expected future emissions remains challenging. Given the sheer magnitude of carbon sequestered within boreal soils and forest biomass as well as the areal extent of boreal regions, however, boreal forest dieback and shifts represent one of the more potentially

immediate              and              significant              climate              tipping              elements.

## 2.6 Tipping point cascade leading to irreversible "Hothouse Earth" warming

### 2.6.1 Background

In 2018, an article by (Steffen et al., 2018) proposed a "Hothouse Earth" scenario, in which modern anthropogenic warming

could trigger strong positive climate feedbacks leading to significant temperature increases well beyond those expected from human greenhouse gas forcing alone. Such an event, the authors propose, could alter the long-term climate trajectory of the Earth itself, removing planetary temperature variations from the glacial-interglacial cycle and placing the biosphere in a newly-created    warmer    equilibrium    that    could    persist    for    up    to    hundreds    of    thousands    of    years.

Throughout its long history, the Earth has transitioned between numerous climate states, from the very warm Paleocene-Eocene Thermal Maximum 56 million years ago (McInerney and Wing, 2011) to the glacial and interglacial cycles of the Quaternary Period (2.6 million years ago to present). Within the latter period, which encompasses the entirety of recorded human history, shifts between glacial and interglacial periods have been primarily driven by slow changes in the Earth's orbit on timescales of approximately 100,000 years. Anthropogenic climate change, however, is driving changes in the earth system at a rapid

pace that the planet has rarely experienced apart from cataclysmic events such as asteroid impacts (Zeebe et al., 2016).

Indeed, a number of studies argue that current human greenhouse gas emissions and associated warming will likely prove sufficient to delay the onset of the next ice age by 50,000 years or more (Berger and Loutre, 2002; Ganopolski et al., 2016). Cumulative emissions of over 1000 Gt C might be sufficient to produce a prolonged postponement of glaciation, which would

include all climate pathways with emissions exceeding an RCP2.6 scenario (Ganopolski et al., 2016). Given the high importance of atmospheric greenhouse gas levels for driving glacial-interglacial cycles alongside cycles in the Earth's long-term orbit, climate change will also likely alter the dynamics of the transitions between glacial and interglacial states, potentially amplifying or accelerating the response of global climate to orbital forcings, as reviewed in (Masson-Delmotte et



al., 2013).


In describing the path towards a "Hothouse Earth" equilibrium, Steffen et al. suggest that failure to meet a 2°C warming goal may risk triggering one or more climate tipping elements with relatively low warming thresholds. Additional warming brought on by these more sensitive tipping elements would in turn activate a number of other tipping elements in a self-reinforcing cascade, potentially pushing the climate system several degrees hotter in response to even a modest failure to meet a 2°C

warming threshold. Due to the extremely long response time of natural carbon sinks that would eventually draw down atmospheric $CO_2$ released through human activity and the tipping point cascade, the authors express concern that such a Hothouse Earth climate could last for hundreds of millennia - a timescale comparable to the length of glacial - interglacial cycles.

Paleoclimate evidence suggests that past climate states of the Earth may have shared similarities with a warmer future world impacted by anthropogenic climate change, including the Early Eocene (50 million years ago) and the Mid-Pliocene (3.3 - 3 million years ago) (Burke et al., 2018). Such periods were marked by notably warmer mean surface temperatures (Early Eocene: 13±2.6°C warmer than present day; Mid-Pliocene 1.8-3.6°C warmer than present day), reduced (Mid-Pliocene) or absent (Early Eocene) ice sheets, elevated sea levels, and higher $CO_2$ concentrations (Early Eocene: 1400 ppm; Mid-Pliocene:

400 ppm). Such paleoclimate analogs for a warmer earth such as the Eocene may also have been characterized by a greatly-reduced extent of subtropical clouds (Schneider et al., 2019). The Mid-Miocene (15.5 to 17 million years ago, 2-4°C warmer, 300-500 ppm $CO_2$) (Greenop et al., 2014; Kominz et al., 2008) also holds some potential parallels with a warmer Earth. The range of conditions denoted by these past eras support idea that a similar future, hotter climate could remain stable over geologic time.


Recently, several of the authors of the original paper published a new Nature Comment, updating their risk assessment of a tipping point cascade with the argument that warming of 1-2°C globally already represents an unacceptably high risk of triggering major tipping elements (Lenton et al., 2019). Both articles nevertheless emphasize substantial disagreement and uncertainties surrounding key assumptions regarding tipping elements and their interactions. Both the 2018 and 2019 articles

themselves stress the need for more quantitative and modeling analysis to substantiate the risks of significant additional climate change as well as the long timescales required for some of the climate mechanisms involved to act.

In the following subsections, we critically review both publications and discuss the likelihood of significant additional climate change driven by tipping elements with the aid of a simple climate model (FAIR) (Millar et al., 2017; Smith et al., 2017).

Drawing upon our literature synthesis throughout this report, we suggest that is unclear if near-term tipping elements possess the potential to drive large (>1°C) additional warming this century, while long-term tipping elements act sufficiently slowly that their impacts lower the risk of cascading effects and can be potentially mitigated by societal action. Other tipping elements,





such as coral reef loss or ice sheet retreat will have minimal effects on global mean surface temperatures. Such considerations are important to include when assessing and discussing tipping point cascade theory. Nevertheless, knowledge gaps associated
with many tipping elements and their potential interactions with one another and global carbon cycle feedbacks highlight a priority need for additional research.

### 2.6.2 Imminent tipping elements

Evaluation of the Hothouse Earth hypothesis involves considerable complexity, as rather than representing its own independent mechanism, the Hothouse Earth pathway effectively consists of numerous individual tipping elements and teleconnections
between them. Consequently, assessment of this hypothesis ultimately boils down to an analysis of its individual parts.

Discussion of the Hothouse Earth hypothesis thus involves three questions: 1) what warming thresholds are tipping elements activated at, 2) are tipping elements with lower thresholds sufficient to drive warming that activates tipping elements with higher thresholds, and 3) what is the cumulative effect of all activated tipping elements and climate feedbacks?
Tipping elements identified by Steffen et al. that this synthesis indicates are at high risk of manifesting in response to warming this century include loss of Arctic summer sea ice, loss of portions of the Greenland Ice Sheet, loss of portions of the West Antarctic Ice Sheet, Amazon rainforest dieback, boreal forest dieback, some permafrost carbon release, potential destabilization of the AMOC, and coral reef loss (Figure 5). Significant ice mass loss from mountain glaciers this century is
also virtually certain (IPCC, 2019).

While irreversible ice mass loss from the Greenland Ice Sheet and West Antarctic Ice Sheet (Section 2.3) and weakening of the Atlantic Meridional Overturning Circulation AMOC (Section 2.1) are likely to be initiated by current rates of warming over the next 100 years, prevailing scientific opinion holds that even under worst-case warming scenarios these mechanisms
will likely take multiple centuries before strong climate impacts manifest (Bakker et al., 2016; Clark et al., 2016; Golledge et al., 2015; Huybrechts and De Wolde, 1999). Consequently, these three longer-term tipping elements are unlikely candidates to play important roles in activating components of a tipping point cascade by driving significant climate change in the near-term. We will instead discuss these mechanisms in the next headed section covering long-term tipping elements.

Of the major tipping elements likely to be triggered this century under current trends, Amazon rainforest dieback (see Section 3.2) represents a large uncertain factor, with high potential to contribute significantly towards climate change on a relatively abrupt timeframe. This stems from the considerable range in estimates of the size of the Amazon forest carbon pool, the strength of the current Amazon carbon sink (0.4 - 0.6 Gt C/yr) (Malhi et al., 2006; Phillips et al., 2009), and the potential magnitude and timescales of conversion of rainforest to a different ecosystem (50-70%, 50-100 years) (Cook and Vizy, 2008;
Lyra et al., 2016). A study simulating the climate impact of Amazon rainforest dieback under a moderate emissions scenario





(IS92a, with $CO_2$ levels reaching 713ppm in 2100) estimated that rainforest loss causes around an additional 0.3°C of warming (Betts et al., 2008). Some researchers have proposed that the Amazon forest could reach a tipping threshold within coming decades due to the combined stresses of deforestation and warming (Lovejoy and Nobre, 2019).

Boreal forest dieback (Section 2.5) also represents a potentially imminent and impactful climate feedback or tipping element. The circumpolar boreal forests store large quantities of carbon and are currently being subjected to increasing stress from fires (Flannigan et al., 2009; De Groot et al., 2013; Shvidenko et al., 2011), pests (Kurz et al., 2008), and rising temperatures (Screen and Simmonds, 2010), exhibiting elevated rates of tree mortality over the observational record (Allen et al., 2010). Modeling studies suggest a credible potential for large-scale biome shifts across major boreal regions in response to high-emissions
climate scenarios over the next century (Foster et al., 2019; Mann et al., 2012; Shuman et al., 2015; Wu et al., 2017). The sign, speed, and magnitude of the climate impact of boreal forest changes remain uncertain, although a strong potential exists for a sizable net warming contribution to global climate.

While gradual Arctic permafrost (Section 2.4) predominantly represents a slow process occurring over centuries, gradual thaw
could also contribute significant additional carbon emissions over the near-term (92 Gt C by 2100 under worst-case scenarios) (Meredith et al., 2019). Abrupt permafrost thaw processes acting over much more immediate timescales could emit up to ~18 Gt C by 2100 including a significant quantity of methane (Turetsky et al., 2020, 2019). Over this century, mean emissions rates from abrupt thaw could contribute approximately ~80 Mt C/yr of methane and 0.14 Gt C/yr of $CO_2$ under the worst-case RCP8.5 scenario (Turetsky et al., 2020), compared to current annual human emissions of 335 Mt C/yr of methane and 10 Gt
C/yr of $CO_2$. Combined, gradual and abrupt permafrost carbon emissions by 2100 will likely meaningfully reduce remaining carbon budgets for current climate targets, but are far from sufficient to independently drive substantial additional warming.

For Arctic sea ice (Section 3.4), a realistic modeling scenario for future Arctic sea ice loss, in which sea ice extent is considerably reduced with a completely ice-free period recurring for approximately one month every summer yields a change
in radiative forcing of 0.3 W/m$^2$ – climatically significant, but only on a level similar to that of current anthropogenic forcing from atmospheric halocarbons (Hudson, 2011).

The remaining other imminent impacts – coral reef collapse (Section 3.1) and loss of mountain glaciers – will likely yield minimal to negligible additional warming or greenhouse forcing. From a climate perspective, it stands to reason that only
tipping elements resulting in significant changes to the Earth's energy balance - through greenhouse gas emissions or albedo changes - increase the risk of a tipping point cascade. Neither coral reef collapse nor mountain glacier melt are anticipated to meaningfully alter greenhouse gas fluxes or planetary-scale albedo. Meanwhile, the likelihood of monsoon regime shifts remains                    highly                    disputed                    (Section                    3.3).



To conduct a first-order approximation of the cumulative impact of these near-term tipping elements, we leveraged the FAIR simple climate model (Millar et al., 2017; Smith et al., 2017), adding projected $CO_2$ and methane natural emissions as well as predicted sea ice radiative forcing changes under the RCP4.5 and RCP8.5 scenarios to the emissions and forcings already incorporated under both pathways, as detailed in (Table 5).

Permafrost $CO_2$ and methane emission rates are derived from mean estimates of cumulative release by 2100 by (Schneider Von Deimling et al., 2015; Turetsky et al., 2020) under RCP4.5 and RCP8.5, assuming a linear rate of release between the present day and 2100. As these estimates do not include the strong compensating carbon-sequestering effect of increased Arctic vegetation growth rates, these figures are potentially aggressive. Methane hydrate emissions utilize the estimate for cumulative release by 2100 of (Kretschmer et al., 2015) for RCP4.5 and double this for RCP8.5, assuming total liberation of methane to

the atmosphere. This is assuredly a high overestimate given strong water column methane sinks as discussed in Section 2.2. For the impact of reduced Arctic sea ice, we base global radiative forcing changes on the estimate of (Hudson, 2011), applying additional radiative inputs reaching 0.3 $W/m^2$ by 2050 and 0.4 $W/m^2$ by 2100 for RCP4.5 and 0.3 $W/m^2$ by 2030 and 0.6 $W/m^2$ by 2100 for RCP8.5. For tipping elements for which a definitive recent estimate of projected carbon release in response to climate change does not yet exist (boreal forest shifts, Amazon forest dieback), we apply the extreme upper-end value selected

by (Steffen et al., 2018).

In total, these climate tipping elements with a high likelihood of occurring this century likely will not produce large additional warming of several degrees C. Rather, the total end-of-century additional warming resulting from the collective contribution of these tipping elements totals ~0.50 C under RCP4.5 and ~0.54 C under RCP8.5 relative to the original scenarios alone

(Figure 6). Even employing aggressive estimates for carbon release and radiative forcing changes, the sum effect of these near-term tipping elements does not meaningfully depart from the envelope of current climate projections. Overall, anthropogenic emissions and uncertainties in both climate sensitivity and carbon cycle feedbacks are likely to remain the dominant factor relative to these tipping elements in determining total warming over the next century. Arctic sea ice, boreal forest dieback, permafrost thaw, and Amazon loss carry implications for carbon budgets and accelerate warming, but this exercise suggests

that even combined, the potential for these mechanisms to drive the transition to a hothouse climate state remains unclear at present. We do however emphasize that a strong need exists for more rigorous modeling work to further explore possible scenarios, particularly in conjunction with general carbon cycle feedbacks and their associated uncertainties.

### 2.6.3 Longer-term tipping elements

For tipping elements that potentially act over centuries to millennia, long timescales of action introduce the possibility that in

the distant future atmospheric greenhouse gas concentrations may have begun to decline thanks to natural carbon sinks, mitigating overall climate effects. Multi-millennial modeling of several high-forcing scenarios indicates that $CO_2$ levels ultimately decline by hundreds of parts per million over the course of one to two millennia, driven by natural carbon sinks


(Clark et al., 2016). Over such timeframes, near-term tipping elements will already have run to completion as well.

An extended time horizon also leaves a significant opportunity for human society to further drive greenhouse gas removal from the atmosphere through carbon capture over future centuries. Many of the future emission scenarios that limit warming to 2C or below by 2100 already include large-scale use of negative emissions technologies, and the use of these could logically be extended and expanded in following centuries (Riahi et al., 2017). Such considerations ameliorate the potential impact of tipping elements with higher triggering thresholds that act over extremely long timescales of centuries or more, such as

permafrost carbon release (see Section 2.4) and methane hydrate release (Section 2.2). The cumulative effects of permafrost thaw and methane hydrate destabilization manifest over centuries and millennia, respectively (Archer et al., 2009; McGuire et al., 2018; Schuur et al., 2015; Turetsky et al., 2020). More recent research indicates that methane hydrate deposits in particular will likely play a much more minimal role in global climate over coming centuries and millennia than previously believed, as reviewed                by              (Ruppel              and              Kessler,              2017).


Large-scale collapse of the Greenland and West Antarctic Ice Sheets (Section 2.3) could strengthen climate change by increasing temperatures through the ice-albedo feedback. However, the planetary albedo change resulting from ice loss takes place on very slow timescales and is additionally balanced by seasonal snow and sea ice and the substantial marine ice generated from ice sheet collapse itself. Some have even suggested that meltwater inputs from Antarctic ice sheet loss could

even impart a near-term cooling effect on global climate, although understanding of meltwater feedbacks on climate remains incomplete and in need of further study (Bronselaer et al., 2018). Complete loss of land-based ice sheets in Greenland is anticipated to require multiple millennia (Pattyn et al., 2018), while loss of the Western Antarctic Ice Sheet similarly takes place over a time scale of centuries at minimum even under extreme scenarios (DeConto and Pollard, 2016; Golledge et al., 2015, 2019).


The likelihood of an AMOC shutdown (Section 2.1) is currently assessed as very low (Collins et al., 2019), with projections still inconclusive as to how much the AMOC may weaken in response to climate change. Current climate models utilize a relatively simple representation of the AMOC, and new oceanographic measurements continue to highlight limitations in our understanding of this system (Lozier et al., 2019) that impact our confidence in model results. Apart from such uncertainties,

it also remains unclear whether the larger climate impacts from a weakening AMOC would produce additional net global warming.

(Lenton et al., 2019) additionally invoke the possibility of triggering stratocumulus cloud deck evaporation (Section 2.7) as a potentially important tipping point. However, the level of greenhouse gas forcing required to cause marine stratocumulus cloud

formations to break down lies at around 1200 ppm $CO_2$ (Schneider et al., 2019), at the upper end of even aggressive, worst-



case          emissions          pathways          (Hausfather          and          Peters,          2020).

Overall, many of the long-term tipping elements discussed above exert climate impacts gradually over lengthy timeframes, while other factors such as an AMOC collapse, stratocumulus cloud deck evaporation, or destabilization of methane clathrate deposits may represent lower-probability events than earlier theorized. The hypothesis that multiple tipping elements can cascade resulting in several degrees of additional warming remains in need of additional substantiation through quantitative modeling. Nevertheless, we underline that the risks of a tipping point cascade and long-term alteration of the Earth's climate trajectory are sufficiently large that any future updates in our understanding of individual tipping elements should motivate reassessment of the dangers of significant additional climate change.

## 2.7 Catastrophic warming due to breakup of stratocumulus cloud decks

### 2.7.1 Background

Stratocumulus cloud decks are a distinct feature over subtropical ocean. They cover some 20% of tropical oceans (Eastman et al., 2011) and play an important role in the global energy balance. These wide swaths of cloud cover reflect 30-60% of incoming shortwave solar radiation (Wood, 2012), in contrast to the high degree of shortwave absorption exhibited by exposed open ocean waters. Accurate representation of cloud dynamics have remained a weakness of large-scale climate models, resulting in uncertainties arising from parameterizations of cloud processes (Lin et al., 2014; Nam et al., 2012). Such challenges have persisted despite the considerable importance of clouds to Earth's radiative balance.

A notable recent scientific paper has proposed that under very high $CO_2$ emissions scenarios (1200 ppm $CO_2$ equivalent), an extreme climate feedback may occur in which stratocumulus cloud decks disintegrate, triggering rapid and substantial warming of 8°C globally (Schneider et al., 2019). Stratocumulus cloud decks are maintained by a key set of temperature and moisture-driven mechanisms that are in turn affected by atmospheric radiative forcing and surface warming. Stratocumulus cloud deck breakup has been proposed to occur when greenhouse gas-induced weakening of radiative cooling at the top of cloud layers in combination with elevated fluxes of warm, dry air across the temperature inversion in the upper troposphere cuts off the cloud decks from their surface moisture supply.

This extreme climate feedback remains a novel, emerging theory, having only been just presented in a single research publication to date. However, the underlying principles of the mechanisms involved, in which weakened radiative cooling decouples clouds from moisture inputs, are well-established (Bretherton and Wyant, 1997), providing theoretical grounding for this hypothesis. This mechanism is suggested to only occur at concentrations of $CO_2$ that are not anticipated to occur this century even under worst-case emissions pathways. At any rate, the significant and abrupt temperature spike that would



accompany stratocumulus cloud deck evaporation represents a catastrophic outcome that strongly justifies greatly increased research attention to this potential tipping point.

### 2.7.2 Mechanisms of stratocumulus cloud deck evaporation

Stratocumulus cloud decks over the subtropical ocean are maintained thanks to a self-sustaining feedback in which emission of longwave radiation from cloud tops plays an important role (Figure 7). Clouds have a high ability to absorb longwave radiation relative to free air. Consequently, re-emission of longwave radiation from cloud tops cools stratocumulus cloud banks from above, fueling convective circulation that supplies these cloud formations with moisture from near-surface air over the ocean (Wood, 2012).


Using a high-resolution large-eddy simulation of a representative summertime subtropical ocean region, Caltech researchers Tapio Schneider, Colleen Kaul, and Kyle Pressel determined that a sufficiently high concentration of atmospheric greenhouse gases could induce rapid disintegration of stratocumulus cloud decks by upsetting this self-sustaining feedback, further suggesting that this process may occur over short timescales at a global scale.


A high greenhouse gas concentration increases the opacity of the air above stratocumulus cloud decks to longwave radiation. This effect increases the downwelling longwave radiative flux from the overlying atmosphere, reducing the difference between this flux and the stronger upwelling radiative flux from the cloud heights. As a result, radiative cooling at the cloud tops weakens with increasing $CO_2$ concentrations, eventually losing the strength required to drive sufficiently strong convection of

air parcels all the way to the ocean's surface.

At the same time, increased surface evaporation due to GHG-induced warming strengthens atmospheric turbulence generated by latent heat release from condensation in clouds. Increased turbulence in turn drives greater entrainment of warm, dry air across the tropospheric temperature inversion (the altitude at which temperature ceases to decrease with increasing height in

the lower atmosphere, increasing again with altitude beyond that height). This second effect also serves as a feedback that works against the existence of stratocumulus decks at higher $CO_2$ concentrations.

Together, these two factors are responsible for breakup of stratocumulus cloud decks in nature (Bretherton and Wyant, 1997), and both mechanisms are expected to intensify with continued $CO_2$ emissions. Schneider et al. found that total cloud deck

collapse occurred at modeled $CO_2$ equivalent concentrations of more than 1,200 ppm, with continuous cloud banks becoming replaced by scattered cumulus clouds. Breakup of stratocumulus cloud decks removes the majority of cloud cover over these ocean regions, allowing greatly increased absorption of solar radiation at the water's surface and triggering dramatic, rapid surface warming. The authors calculated that the resulting change to the Earth's overall energy balance would be sufficient to increase surface temperatures worldwide by 8°C, with particularly intense warming of up to 10°C in subtropical regions.




Model runs suggested that cloud deck collapse occurs suddenly rather than gradually. At lower $CO_2$-eq concentrations of 400-800 ppm, cloud cover remained dense at present levels, albeit with reduced water content. Once collapse was initiated, however, the process proceeds aggressively and rapidly as the abovementioned feedbacks rapidly self-amplify with the loss of cloud cover, likely starting from the edges of stratocumulus regions where stratocumulus cloud decks exist closer to the critical

stability threshold. However, because cloud decks in different regions do not share the same proximity to the stability threshold, it remains unclear how abrupt of a process cloud deck breakup would be at a global scale.

The authors do note that the precise $CO_2$ concentration at which stratocumulus cloud deck evaporation is triggered also depends on large-scale atmospheric dynamics that may themselves shift with climate change. They further highlight a potential

mediating mechanism reducing the likelihood of collapse in which predicted weakening of large-scale subsidence in the troposphere with continued warming may somewhat buttress the effect of increasing cloud deck instability (Blossey et al., 2013; Bretherton, 2015; Held and Soden, 2006; Tan et al., 2017).

### 2.7.3 Potentially extreme impacts of this cloud feedback underline need to fill knowledge gaps

The consequences of stratocumulus cloud deck disintegration triggering a highly abrupt warming of 8°C or more would have

devastating implications for human society and natural ecosystems worldwide. Such a magnitude of warming would severely challenge humanity's adaptive ability by massively disrupting global agriculture, altering weather patterns, and significantly accelerating sea level rise, among other effects—all within an extremely short timeframe. A global mean temperature increase of this level would also carry a high risk of activating or exacerbating many of the other tipping elements covered in this report, resulting in further amplified climate impacts.


The Schneider et al. paper further modeled a lagging, hysteresis-like recovery pattern for stratocumulus decks in which these cloud banks only re-form after $CO_2$ levels drop significantly below the original triggering threshold. Reformation of stratocumulus cloud decks only occurred after $CO_2$ concentrations fell to nearly pre-industrial levels (<300 ppm $CO_2$-eq). This suggests that once stratocumulus cloud deck collapse occurs, the process of restoring them through carbon sequestration, solar

geoengineering, or other means may involve an extremely delayed recovery.

A substantial mitigating consideration, however, is that the greenhouse gas concentrations required to initiate stratocumulus cloud deck evaporation (~1200 ppm $CO_2$ equivalent) are high, at the upper limit of potential greenhouse gas concentrations in 2100 under the very-high RCP8.5 emissions scenario. With the RCP8.5 pathway itself representing a "worst-case" emission

scenario that is unlikely to be realized (Hausfather and Peters, 2020), it appears uncertain that the concentration thresholds required to trigger the stratocumulus cloud deck phenomenon will be reached this century. Thresholds for other tipping



elements such as ice sheet collapse or Amazon forest dieback could be reached first at much lower levels of forcing, suggesting that the bulk of scientific and policy attention should remain on more proximate climate feedbacks.

Furthermore, given the magnitude of the requisite critical $CO_2$ threshold, it remains unclear whether stratocumulus cloud deck breakup could be triggered in the case of a tipping point cascade that includes other tipping elements discussed in this report. As discussed above, current consensus holds that many of the tipping elements that are popularly thought to carry a risk of abrupt carbon release (permafrost thaw, marine methane clathrate release, boreal forest dieback, Amazon forest dieback) would most likely cumulatively release greenhouse gases on timescales that are more gradual (many decades to millennia). More
long-term, steady release of carbon dioxide or methane from, say, decomposing permafrost would lead to a moderate increase in the steady-state concentration of atmospheric $CO_2$, as opposed to a spike in greenhouse gas forcing of sufficient severity to initiate cloud deck disappearance. Were human emissions to accelerate along a worst-case emissions pathway, however, the risks of climate feedbacks pushing us above the threshold for cloud deck breakup would become elevated.

Overall, stratocumulus cloud deck evaporation should be assessed as a high-impact tipping point that emerges as a concern only for greenhouse gas forcing at the upper limits of worst-case emissions scenarios for this century. Furthermore, this hypothesized phenomenon remains a novel, under-studied problem that has only recently been brought to the attention of the scientific community. Substantial further research efforts are required in order to investigate this climate feedback in detail, assess risks and potential impacts, and better clarify the conditions that may trigger cloud deck disintegration.

## 3 Regional tipping elements

### 3.1 Continued die-off of tropical, shallow-dwelling coral reefs

#### 3.1.1 Background

Coral reefs are among the most productive and biodiverse marine ecosystems worldwide, with their biodiversity collapse
posing the possibility of uncertain, unanticipated shifts to regional marine ecology. Strong scientific evidence points towards tipping point in rising temperature beyond which tropical coral reefs will undergo severe ecosystem shocks. Tropical corals demonstrate high sensitivity to temperatures outside of their accustomed range, and are additionally under stress due to increasing trends in ocean acidification and deoxygenation (Bindoff et al., 2019). Temperature anomalies are a very strong predictor of coral bleaching (Sully et al., 2019), a phenomenon in which corals expel the photosynthetic algae with which they
share a symbiotic relationship. As these photosynthetic dinoflagellates of the *Symbiodiniaceae* family provide corals with up to 90% of their energy requirements, the loss of these symbionts for a prolonged period can lead to starvation, susceptibility to                              diseases,                              and                              death.



While tropical corals undergo daily environmental fluctuations and have survived substantial evolutionary pressures over the
course of geologic history, the current rate of ecosystem change presents a serious threat. As marine heatwaves affect tropical
coral reefs worldwide (Baker et al., 2008; Heron et al., 2016; Spalding and Brown, 2015) coupled with the increasing rate of
mean ocean temperature rise (Bindoff et al., 2019), the outlook for corals appears grim. Since the 1980s, researchers have
additionally documented three large global-scale coral bleaching events (1998, 2002, and 2015-2016) in response to
particularly high sea temperature anomalies, with a fourth bleaching event likely to occur within the next decade or two
(Hughes                                                et                                                al.,                                                2017).

Even for a relatively moderate warming scenario, reef organisms would need to demonstrate the ability to adapt to a
temperature change of more than 2°C by the late 21st century, a pace of adaptation likely too extreme for these ecosystems to
match (Donner et al., 2005). Coral researchers consequently show strong agreement that the majority of tropical coral habitats
will see sharp declines in biodiversity within the coming decades (Bindoff et al., 2019).

The economic and societal impacts of coral reef loss to Indo-Pacific, Caribbean, and other communities can be expected to be
substantial. Coral reefs serve as important factors for fishery productivity, represent important cultural resources, and provide
shoreline fortification from storm damage (Ferrario et al., 2014). Many small island nations are also highly dependent upon
economic activity provided by coastal tourism, which existing reef ecosystems play a strong role in attracting (Spalding et al.,
2017; Weatherdon et al., 2016). Degradation of warm-water coral ecosystems thus represents an acute threat to island nations
and coastal communities across the Asia-Pacific region, endangering their economic security, access to food, and resilience to
sea level rise and extreme weather.

### 170 3.1.2 Ongoing and projected temperature stresses and amplified ocean acidification will drive major loss of coral reefs within this century

Coral bleaching events in response to elevated temperatures has been repeatedly documented in observational studies (Hughes
et al., 2017; Wernberg et al., 2015). Four to six week-long periods where water temperature remains +1°C warmer than the
average long-term summer maximum can cause significant bleaching and high mortality (Jokiel and Coles, 1990). Significant
long-term declines in coral abundance have already occurred in tropical coral regions, including the Caribbean (Gardner et al.,
2003; Jackson et al., 2014), the Indo-Pacific region (Bruno and Selig, 2007), and the Great Barrier Reef (De'Ath et al., 2012)
due to overfishing, hurricanes, and severe bleaching events. The severity of bleaching events is further observed to be
increasing over time and becoming the top threat to coral reefs (De'Ath et al., 2012; Hughes et al., 2018).


Coral reef ecosystems have demonstrated a limited ability to adapt to temperature stress and rebound once conditions improve (Brown et al., 2002; Coles and Brown, 2003; Jury and Toonen, 2019), with observations of recent coral bleaching events being triggered at ~0.5°C higher temperatures than previous bleaching episodes (Sully et al., 2019). Coral reef ecosystems may recover following mass mortality events, replacing the loss of coral cover and diversity in a decade or more (Spalding and Brown, 2015). However, further warming is strongly anticipated to outpace the adaptive capacity of corals and time required

for recovery (Frieler et al., 2013; Hoegh-Guldberg et al., 2007; Veron et al., 2009). Furthermore, the capacity to adapt may be limited to just a small subset of coral species and associated marine organisms (Heinze et al., 2015).

The pace of current and future ocean warming will intensify pressures on corals. Over the past 30 years, global mean sea surface temperatures have risen by 0.015°C per year, with this rate projected to accelerate to an average of 0.027°C per year

between 1990 and 2090 (Bopp et al., 2013). Under predicted temperature increases, modelers anticipate bleaching occurring with an annual or biannual frequency for most reef ecosystems within 30-50 years (Donner et al., 2005).

Ongoing ocean acidification represents another important stressor, with mean surface ocean pH declining at a rate of 0.018 units per decade across 70% of ocean biomes (Lauvset et al., 2015). The IPCC assigns near-certainty to the likelihood of

continued surface ocean acidification, with a decline in pH of 0.287-0.29 pH units by 2081-2100 relative to 2006-2015 under the RCP8.5 emissions pathway (Bindoff et al., 2019). Even under the optimistic RCP2.6 scenario, marine pH declines over the same period by 0.036-0.042 pH (Bindoff et al., 2019). Lower pH requires higher energetic costs for coral to build their calcium carbonate skeleton, while also subjecting dead corals that serve as the foundation for living reef structure to accelerated dissolution and erosion (DeCarlo et al., 2015; Silbiger et al., 2014). Numerous observational studies have confirmed the

ongoing negative impacts of ocean acidification upon reef habitats and project continued degradation of corals with further increases in atmospheric $CO_2$ concentration (Bove et al., 2019; Jiang et al., 2018; Kroeker et al., 2010; Mollica et al., 2018; Orr et al., 2005).

Simultaneously, sea level rise also presents another additional threat to shallow-dwelling reefs through inundation, as waters

rise faster than the reefs can grow upwards towards light (Perry et al., 2018). Many coral reefs are also directly subject to human-induced negative impacts such as increased sediment runoff, chemical pollution, nutrient-fueled harmful algal blooms, and damage from fishing activities including dynamite fishing (Burke et al., 2011; Halpern et al., 2015; Hodgson, 1999). Nutrient pollution, competition with algal overgrowth, and marine de-oxygenation also pose potential threats to reef systems in close proximity to human activities (Bindoff et al., 2019).


Research results demonstrate strong agreement that severe coral reef degradation will likely continue throughout the current century even under optimistic climate mitigation scenarios, with coral abundance declining to 10-30% of today's levels even with warming limited to 1.5°C (Hoegh-Guldberg et al., 2019). Warming of 2-2.5°C, well within projections for the 21st century





under current emissions rates, would cross a tipping point, eliminating >99% of tropical corals, completely and irreversibly

transforming coral reef ecosystems (Frieler et al., 2013; Hoegh-Guldberg, 2014b, 2014a; Hoegh-Guldberg et al., 2019; Hughes

et al., 2017; Schleussner et al., 2016).

Currently, a future in which tropical corals undergo significant ecosystem transitions therefore remains the most likely outcome

in the absence of extremely aggressive climate mitigation efforts. Although tropical corals may persist in some form, chances

are high that future coral ecosystems may appear completely unrecognizable compared to their state today. Given the

considerable sensitivity of reef habitats to even modest temperature increases under strong climate mitigation scenarios,

combined with the impact of multiple stressors like pollution and ocean acidification, avoiding a future in which coral reefs

degrade may no longer be possible.

### 3.2 Loss of Amazon rainforest and conversion of significant rainforest area to savanna

**3.2.1 Background**

The Amazon region of South America contains the world's largest expanses of old-growth tropical rainforest. This large

ecosystem possesses global importance, alone producing an estimated 15% of total terrestrial photosynthesis worldwide (Field

et al., 1998). The rainforest region has also long been recognized to possess rich biodiversity, and also represents a major

terrestrial biological carbon sink of some 0.4 – 0.6 Gt C per year (Malhi et al., 2006; Pan et al., 2011), a flux of similar

magnitude to the ~0.5-0.6 Pg C per year sequestered by all northern boreal forests globally (Pan et al., 2011; Schaphoff et al.,

2013). The Amazon rainforest additionally contains a substantial amount of stored carbon in biomass and soil organic matter,

tentatively estimated to range between 150-200 Gt C (Cerri et al., 2007; Gibbs et al., 2007; Malhi et al., 2006; Saatchi et al.,

2011). Between all of these factors, the Amazon rainforest represents a critical component of the global carbon cycle.

As with many of the world's major rainforests, a significant fraction of precipitation over the Amazon rainforest is recycled

water originating from the rainforest vegetation itself. While on average recycled water accounts for some 25-35% of total

precipitation in the Amazon region (Eltahir and Bras, 1994; Da Rocha et al., 2009; Salati et al., 1979; Zeng et al., 1996), the

contribution of recycled water to total rainfall increases during the dry season. In the dry season, recycling is the majority

source of regional water vapor, and also plays a role in initiating the onset of the wet season (Li and Fu, 2004; Wright et al.,

2017). Consequently, dry season precipitation and length for the Amazon region are directly linked to the total size of the

rainforest system itself.

Currently, the Amazon rainforest is experiencing a high rate of deforestation, having lost close to 20% of its pre-1970 areal

extent to date. At its peak around 2005, deforestation rates approached 30,000 sq km of cut or burned forest per year, and while

this fell to an average of 6,000 sq km lost per year between 2011-2015 and has remained low relative to early 2000s rates, such



a rate of loss nevertheless remains highly unsustainable (Nobre et al., 2016). Recent policy shifts in Brazil have once again intensified deforestation, with 9,762 sq km of forest loss recorded during the period between July 2018-2019 (INPE, 2019).

Some Amazon researchers have warned that vegetation loss could ultimately reduce recycled precipitation inputs across large
portions of the Amazon basin to the point where large rainforest trees become subject to increased mortality due to the dry season lengthening (Nobre and Borma, 2009). As rising numbers of trees die due to water stress (Phillips et al., 2009), further reducing the strength of the regional water cycle, dieback of rainforest vegetation becomes a self-sustaining positive feedback, ultimately causing significant sections of the Amazon rainforest to collapse and potentially shift towards a savanna-type ecosystem (Lyra et al., 2016; Nobre and Borma, 2009).

Other authors have outlined alternative pathways for this region-wide tipping point theory, suggesting that increased wildfire represents a more immediate threat to rainforest health (Brando et al., 2019), pointing out that rainforests may shift towards seasonal forest as opposed to savanna grasslands (Malhi et al., 2009), emphasizing uncertainties in the ability of models to predict seasonal precipitation (Good et al., 2013), and highlighting the potential for $CO_2$ fertilization to compensate for biomass
losses (Cox et al., 2013; Huntingford et al., 2013). Such factors could mean that changes for the Amazon forest will be driven by local climate-forced conditions, rather than a biome-wide climate threshold.

Large-scale Amazon ecosystem shifts are nevertheless likely in the absence of climate mitigation and would severely impact local populations by radically altering regional ecology and climate, and could influence climate patterns around the world.
Such a change has been hypothesized to potentially occur within a relatively short time frame of around 50 years (Cook and Vizy, 2008; Lyra et al., 2016), significantly reducing the strength of the Amazon system as a terrestrial natural carbon sink and also releasing considerable quantities of stored carbon to the atmosphere via decomposition, wildfire, and land-use changes.

**3.2.2 Substantial evidence indicates that significant regions of the Amazon are at risk of dieback caused by fires and**
**drought**

Both observations as well as modeling studies have proposed a threshold for Amazon forest dieback, although ongoing debate continues to discuss flaws in this hypothesis and contest whether a biome-wide tipping point exists. Uncertainties also complicate efforts to precisely determine the boundaries of this regional tipping point. Nevertheless, the general threat of regional ecosystem shifts driven by wildfires, drought, and deforestation is well agreed upon within the research community.

Water represents a key variable for assessing the Amazon's future. The Amazon rainforest typically receives annual precipitation of approximately 2200mm; the western reaches of the region receive the highest amount of rainfall (3000mm) thanks to the topographic influence of the Andes mountains (Salati and Vose, 1984), and these wettest portions of the rainforest





are consequently likely to survive even under drier conditions triggered by dieback. The lower limit of precipitation necessary

to maintain a closed canopy tropical rainforest sits at around 1600 mm/yr (Hirota et al., 2011). During the dry season, the

Amazon region actually experiences a water deficit that is partially replenished by recharge during the wet season.

Consequently, the twin factors of dry season severity and overall rainfall play central roles in maintaining existing rainforest

cover (Malhi et al., 2009).

Observations are already indicating that the regional water cycle is shifting towards conditions of increasing water stress. The

southeastern Amazon basin currently experiences reduced rainfall relative to the rest of the basin, around 1700 mm/yr - an

effect that is attributed to higher intensity of land use (Salati and Vose, 1984). The frequency of drought events in the southern

Amazon has also increased according to a study examining dry events from 1970-1999 (Li et al., 2008). Significant evidence

suggests that the dry season has lengthened, particularly over the southern and southeastern Amazon (Dubreuil et al., 2012; Fu

et al., 2013; Marengo et al., 2011).

At the same time, multiple factors exert important influences over Amazon ecosystem health (Figure 8). Increasing regional

temperatures due to both local and global drivers (Jones, 1994; Victoria et al., 1998) elevate potential evaporation and reduce

humidity, increasing quantities of dry fuel and heightening fire risk (Cochrane and Barber, 2009; Cochrane and Schulze, 1999;

Pueyo et al., 2010). The seasonality and magnitude of rainfall may change due to shifts in the El-Nino Southern Oscillation,

although predictions remain highly uncertain. Some research has even suggested that elevated $CO_2$ concentrations may reduce

the rate of stomatal leaf opening in trees, lowering evapotranspiration and thereby further weakening recycled precipitation

(Lammertsma et al., 2011; Walker et al., 2014).

The influence of these diverse factors complicates efforts to more precisely define the critical boundaries to Amazon rainforest

stability. A temperature increase of 3-4°C (Nobre et al., 2016), deforestation of >40% (Nobre et al., 2016), or precipitation

decrease of 30-40% (Lenton et al., 2008; Salazar and Nobre, 2010) each have been proposed as thresholds independently

capable of initiating Amazon dieback according to models, but all three factors are currently occurring simultaneously. Lovejoy

and Nobre have proposed a deforestation extent of 20-25% as the tipping point of savannization, suggesting that the current

Amazon system already stands perilously close to a critical threshold (Lovejoy and Nobre, 2018), beyond which a strong self-

sustaining water stress feedback will act to convert 50-70% (Cook and Vizy, 2008; Lyra et al., 2016) of the region to a savanna-

type                                                                                                                              ecosystem.

Such a threshold may be aggressive, however. Substantial uncertainty characterizes modeled future precipitation changes for

the region (Good et al., 2013; Li et al., 2006) and models' ability to represent subsurface hydrological processes remains

limited (National Research Council, 2013). In particular, some researchers have asserted that increases in wildfires as a result

of drying will play a more immediate and serious role in driving forest loss (Brando et al., 2019). Field studies have indicated





that while prolonged drought does eventually raise mortality rates for rainforest trees, this only occurred after the third consecutive year of a particularly intense simulated drought (Nepstad et al., 2007). In contrast, a forest fire during a drought

period can kill half the adult trees in the affected area within a much shorter period (Balch et al., 2015; Brando et al., 2014; Nepstad et al., 1999). Tree mortality and reduced canopy cover caused by fire can in turn permit invasive, more fire-prone grasses to invade the forest floor, increasing the region's susceptibility to fire and driving conversion to grassland that primarily results    from    dryness    and    fire,    not    drought-induced    water    stress    (Balch    et    al.,    2015).

A number of other studies have also identified potential caveats to the precipitation-driven regional dieback hypothesis. For instance, a climatological study proposes that Amazon forest vegetation might generally give way to seasonal forest rather than savanna grasslands (Malhi et al., 2009). Other research suggests that promotion of biomass accumulation thanks to $CO_2$ fertilization may offset tropical forest losses due to drought and rising temperature (Cox et al., 2013; Huntingford et al., 2013). Consequently, the IPCC has only assigned medium likelihood and confidence to their assessment of the probability of

precipitation-driven regional dieback (Settele et al., 2014).

### 3.2.3 Loss of Amazon rainforest could occur over decades to a century and cause important climate, human, and ecosystem impacts

According to the rainfall-driven dieback theory, after crossing the critical threshold, affected portions of the Amazon would undergo savannization over a timeframe of likely somewhere between several decades to a century (Lenton et al., 2008;

Lovejoy and Nobre, 2018; Nepstad et al., 2008; Nobre et al., 1991; Nobre and Borma, 2009). Collapse begins with die-off of the largest trees as soil moisture becomes increasingly depleted (Ivanov et al., 2012; Nepstad et al., 2008), progressively thinning the forest canopy. This transition would likely be accompanied by intensification of wildfires as dead vegetation and drier conditions contribute towards higher fire risk (Brando et al., 2014; Cochrane and Barber, 2009; Cochrane and Schulze, 1999). This drought-fire feedback might in fact independently drive ecosystem shifts, dominating over impacts from water

stress and resulting in a pattern of forest loss driven by more local factors as opposed to a biome-wide threshold (Balch et al., 2015;    Brando    et    al.,    2019,    2014).

Degradation of the Amazon rainforest might even still occur under strong climate mitigation scenarios, with significant biomass losses possible at 1.5-2°C of warming (Hoegh-Guldberg et al., 2018). However, forest management policy could act

as a strong lever upon the Amazon's ecosystem health, with a halt to new forestation potentially reducing burned area by 2050 by 30% and accompanying greenhouse gas emissions by 56% (Brando et al., 2019). Fire suppression also represents a potentially powerful tool to reduce rainforest loss by preventing positive feedbacks between fire and savannization. As the Amazon rainforest is naturally warm, forest litter decomposes rapidly and does not accumulate as it does in some temperate forests,    meaning    that    aggressive    fire    suppression    carries    no    risk    of    intensifying    future    fires.





Once savannization or conversion to seasonal forest has occurred, significant hysteresis may inhibit the return of rainforest vegetation even over very long timescales, thanks to potentially more frequent fires in a savanna ecosystem and to savanna
vegetation's ability to prevent recruitment of larger tree species. Paleoclimate evidence confirms that regrowth of rainforest cover often lagged significantly behind the return of wet conditions to South America during past periods of the Holocene (Ledru et al., 1998).

Loss of large portions of the Amazon rainforest would carry significant implications for the strength of the Amazon carbon
sink as well as the fate of the organic carbon within Amazon vegetative and soil biomass. Rainforest degradation could likely shift the Amazon region from a carbon sink of up to 0.6 Pg/yr (Malhi et al., 2006) to a potentially strong carbon source (Nobre and Borma, 2009). Modeling of an Amazon rainforest dieback scenario under moderate emissions (713ppm in 2100) produced an additional 0.3°C of warming globally as a result of rainforest loss (Betts et al., 2008).

Precise estimates of the carbon impact of Amazon dieback are however subject to variability due to uncertainty regarding the potential extent of forest loss, the current size of the Amazon forest carbon pool, and potential timescales of change. Under a worst-case RCP8.5 emissions pathway with continued deforestation, carbon dioxide emissions from intensifying fires were estimated to sum to 4.6 Gt by 2050 (Brando et al., 2019). However, carbon release as a result of fire can eventually be partially offset by recovery of vegetation within the burned areas. Carbon emissions from losses in forest area may also be potentially
compensated for if the positive response of rainforest vegetation to increased carbon dioxide levels is large (strong $CO_2$ fertilization). As the effect of $CO_2$ fertilization upon plant growth remains highly uncertain, this complicates efforts to estimate the overall carbon impact of the world's tropical forests as they respond to climate change (Cox et al., 2013; Huntingford et al., 2013). However, recent observational evidence strongly suggests that the $CO_2$ fertilization effect is nearing saturation in African tropical forests as well as the Amazon on faster-than-predicted timescales (Hubau et al., 2020).

Amazon rainforest loss possesses acute regional ramifications. Degradation of the rainforest represents a serious threat to the livelihoods and lifestyles of indigenous Amazon peoples. Total rainfall throughout the Amazon basin may generally fall following dieback and savannization, placing regional agriculture at risk (Arvor et al., 2014; Oliveira et al., 2013). A study has found that for 40% deforestation, hydroelectric power generation from major dams on the Xingu River is dramatically reduced
thanks to falling river runoff (Stickler et al., 2013). That said, the complexity of rainfall dynamics means that impacts depend strongly on the spatial pattern of forest loss (Lawrence and Vandecar, 2015). Increased wildfire frequency and severity will additionally put regional communities at risk and create air pollution crises. Finally, dieback of the Amazon rainforest will represent a major threat to the biodiversity of this region (Esquivel-Muelbert et al., 2017).



Among the tipping elements covered throughout this report, the Amazon forest dieback scenario stands among the more imminent, likely, and fast-acting of the climate feedbacks popularly discussed within the climate community. Substantial scientific evidence backs the assessment that the Amazon rainforest faces a serious threat within the 21st century, with deforestation and wildfire playing a particularly critical role in driving the ecosystem towards a sudden transition. Forest management policy will prove key to determining the future of the Amazon, as evidenced by the success of the Brazilian

government in at least temporarily reducing the rate of deforestation relative to the high pace of loss seen in the early 2000s (Nobre et al., 2016). Under present pressure, however, the Amazon rainforest will indeed face serious threats this century, with significant regional and global consequences accompanying its degradation.

### 3.3 Disruption to African or Indian seasonal monsoons

#### 3.3.1 Background

Seasonal monsoonal rains are a distinct feature of the regional climate of South America, Mexico, northern Australia, southern Africa, West Africa, and South and Southeast Asia, playing key roles in governing water availability and agricultural yields for large human populations (Christensen et al., 2013). Extreme precipitation events in these regions are often associated with the monsoonal wet season as well, with the potential to cause flooding and landslides that can inflict significant loss of life and property. Given the importance of regional monsoons to food security, water security, and natural disaster risks, researchers

worldwide have devoted sizable efforts towards predicting changes to monsoon seasonality and intensity that might be expected in response to climate change.

The seasonal cycle of monsoons results from annual latitudinal movements of the zone of maximum solar heating. Air over landmasses warms more rapidly than air over the ocean, due to the slower response of ocean waters to solar heating than

continental land. The difference in temperature between land and sea then proceeds to drive the monsoon circulation, bringing moist air over warmer land which then warms and rises, releasing moisture as rainfall (Christensen et al., 2013).

Efforts to project likely climate-driven changes for the South Asian and West African monsoons have been complicated by the difficulty of confidently modeling patterns of precipitation with coupled climate models. Rainfall records do suggest that

large-scale shifts may be underway across South and Southeast Asia and West Africa. The South Asian monsoon appears to have shifted in strength, with a 10% decrease in precipitation (Singh et al., 2019b). Analysis of historical observations from 1951-2011 reveals that while overall peak-season precipitation has fallen, day-to-day precipitation has increased in variability, with dry spells becoming less intense but more frequent and wet spells becoming more intense (Singh et al., 2014, 2019b). The historical trend of decreasing overall precipitation may also have reversed over the past few decades, with monsoon rainfall

having increased since 2002 (Jin and Wang, 2017). Meanwhile, rainfall in the Sahel region of Africa decreased markedly from



1950 to 1980 by 40%, a trend that proceeded to partially recover between 1980 and 2000 (Held et al., 2005). Overall however, CMIP5 models generally predict a strengthening on average of the seasonal monsoon worldwide in the future (Christensen et al., 2013; Hoegh-Guldberg et al., 2018). However, modelers caution that monsoon projections continue to be accompanied by large uncertainties, as only a few CMIP5 models are able to accurately reproduce spatial and temporal monsoon characteristics
(Singh et al., 2019b).

Some concern has been voiced that climate forcing could cause a sudden reduction in South and Southeast Asian monsoon strength and an abrupt increase in West African monsoon intensity, substantially altering total wet season rainfall for affected areas e.g. (Bathiany et al., 2018; Lenton et al., 2008; Levermann et al., 2009; Schewe et al., 2012; Zickfeld et al., 2005). This
theory remains controversial and disputed (Boos and Storelvmo, 2016a), with the majority of research to date suggesting a more linear increase of monsoon intensity in response to warming instead (Christensen et al., 2013).

Billions of people live in the regions of the world subject to seasonal monsoon rains. The annual timing of the monsoon has strongly influenced agricultural practices, culture, and daily life. The South Asian monsoon for instance represents a dominant
input of precipitation across much of the region, providing more than 75% of total annual mean rainfall in India over a four-month period (Kumar et al., 2011). Changes in rainfall further affect hydroelectric generation capacity and water availability for industrial purposes. Consequently, in addition to the role of monsoon precipitation in triggering natural disasters such as flooding or landslides, climate-induced changes in the monsoon's strength carry the potential to affect the lives of a significant fraction                                                                              of                                                                              humanity.

Although the likelihood of abrupt drying or wetting of existing monsoonal patterns in response to climate forcing remains contested and in opposition to the majority of the monsoon literature, this importance of monsoon systems for large populations globally provides ample motivation for continued study of the potential for abrupt change.

### 3.3.2 Potential instability in the South Asian and West African monsoon remains contested and uncertain

Strong disagreement surrounds the question of whether or not monsoon systems can exhibit tipping point behavior for monsoon systems (Boos and Storelvmo, 2016b, 2016a; Levermann et al., 2016). Initial proposals that the South Asian and West African monsoons could exhibit an abrupt, unstable transition between states in a warmer world invoked a breakdown of the regional moisture-advection feedback (Lenton et al., 2008; Levermann et al., 2009). In this feedback, when water vapor condenses to become rain, this phase change is accompanied by a release of latent heat that reinforces existing regional
gradients of heat and pressure (Levermann et al., 2009). The monsoonal circulation further strengthens in response to the more powerful gradient in a self-reinforcing dynamic.



Human-induced changes such as aerosol emissions or land use shifts over India could increase regional albedo, causing cooling over the South Asian landmass and weakening the temperature gradient responsible for the monsoon's onset (Zickfeld et al., 2005). The effect of cooling, moisture inputs, and cloud cover changes as a result of irrigation for agriculture can also have dramatic net cooling effects (Cook et al., 2015; Singh et al., 2018) which can influence patterns of monsoon precipitation (Devanand et al., 2019). The tipping points proposed for the South Asian monsoon suggest a potential for dramatically reduced monsoon precipitation in response to land mass albedo reaching or exceeding 0.5 in response to land use change and aerosol emissions over South Asia (Lenton et al., 2008; Zickfeld et al., 2005). Others have explored potential humidity thresholds for the monsoon circulation (Schewe et al., 2012).

Conversely, some have proposed that the West Africa region may experience a strengthening of the monsoon circulation, leading to a significant intensification of the Sahel region which could even result in greening of currently arid regions south of the Sahara (Lenton et al., 2008). The same article postulates a tipping threshold for the West African monsoon's intensification for global mean warming of 3°C or more. Relatively little subsequent research to date has explored this proposed threshold further.

These proposed "tipping points" for the South Asian and West African monsoons and their underlying assumptions remain extremely contentious and have been criticized (e.g. (Boos and Storelvmo, 2016b; Hill, 2019; Hoegh-Guldberg et al., 2018)), although proponents maintain that abrupt shifts in monsoon regimes are theoretically possible (Bathiany et al., 2018). Critics point out that modeling analyses producing abrupt monsoon regime shifts omitted the influence of static tropospheric stability, which results in linear as opposed to non-linear monsoon changes in response to warming once incorporated (Boos and Storelvmo, 2016b). In response, (Levermann et al., 2016) assert that tropospheric stability is insufficient to counteract the moisture-advection feedback, and point to paleoclimate evidence of past abrupt monsoonal transitions. Overall, it seems fair to say that a mechanism that could result in abrupt monsoon regime change in response to warming remains incompletely proven, and that agreement on the potential for abrupt monsoon shifts remains elusive. Literature surveys and IPCC projections for future monsoon shifts continue to favor more consistent increases in monsoon strength globally over the 21[st] century due to warming (Christensen et al., 2013; Turner and Annamalai, 2012), although rainfall variability may also increase (Menon et al., 2013).

Accurate simulation of monsoon patterns does generally remain a challenge, with considerable biases in models of historic and current monsoon dynamics. For instance, CMIP3 models hindcasting past rainfall produce precipitation results across India and Southeast Asia that differ considerably from historic rainfall observations, possibly due to poor ability of models to represent the effects of aerosols (Turner and Annamalai, 2012) or irrigation (Cook et al., 2015; Devanand et al., 2019; Singh et al., 2018). CMIP5 models have also generally failed to capture (Saha et al., 2014) the marked decline in total June-September mean Indian rainfall since 1950 (Ramesh and Goswami, 2014; Saha et al., 2014) and still do not incorporate effects from



irrigation (Singh et al., 2019b). Aerosols also remain a considerable source of variability in modeling results (Singh et al., 2019a), with more realistic aerosol-cloud interactions producing more accurate rainfall trends (Lin et al., 2018). Model hindcasts of Sahelian rainfall can capture the historic decline in regional precipitation over the 20th century, but poorly replicate multidecadal variability (Biasutti, 2013).

Conflicting results produced by future modeling studies overall point to a yet-incomplete understanding of monsoon dynamics and the limitations of current models (Hill, 2019). The IPCC Special Report on Warming of 1.5°C concluded that projections for monsoon trends under lower global temperature increases of 1.5°C to 2°C remain particularly uncertain (Hoegh-Guldberg et al., 2018). Most models produce increases in South Asian monsoon rainfall with additional warming, with projections of decline remaining relatively rare, although model predictions vary between steep and relatively flat gains in precipitation over the 21st century (Turner and Annamalai, 2012). Theoretical modeling has suggested that increases of monsoon strength in response to climate change are linear by nature (Boos and Storelvmo, 2016b). Increases in precipitation are strongly dependent on human-driven changes in aerosol forcings, however, with reductions in aerosol emissions over South Asia having the potential to accelerate rising trends in rainfall (Wilcox et al., 2020). As for the West African Monsoon, a similarly large range of uncertainty continues to accompany future rainfall and monsoon predictions for West Africa. A comparison of CMIP5 model trends found that 16 of 20 models agreed upon a drier start to the West African monsoon followed by a wetter period later in the wet season, but that the set of 20 models showed substantial spread overall in terms of rainfall outcomes (Biasutti, 2013).

Overall, current literature on the South Asian and West African monsoons largely disagrees with the assessment that these systems constitute tipping elements or tipping points, with most projections for the South Asian monsoon in particular suggesting rainfall changes in the opposite direction from the abrupt declines suggested by tipping threshold theories. Furthermore, the ability of investigators to accurately reproduce monsoon patterns through models remains limited due to complex but important interactions between aerosols, irrigation, cooling, and clouds (Devanand et al., 2019; Singh et al., 2019a). While room certainly remains for further improvement of future projections of monsoon strength and variability, the existing body of evidence generally does not find a meaningful likelihood of tipping behavior for these monsoon systems.

### 3.4 Loss of summer Arctic sea ice

### 3.4.1 Background

The shrinking of Arctic sea ice area over the late 20th and early 21st centuries has been clearly documented by observations of decreasing sea ice extent in all months, with particularly strong reductions in area during summer (Fetterer et al., 2017; Stroeve et al., 2012a). The characteristics of Arctic sea ice have also shifted, with multi-year sea ice at least five years old falling from 30% of Arctic ice to just 2% between 1984 and 2019 (Stroeve and Notz, 2018). All of these trends are driven by



a relatively rapid regional warming, as evidenced by rising surface solar heat inputs across 85% of the Arctic region between 1979-2007 (Perovich et al., 2007) and a fast-paced increase in regional ocean temperatures of around 0.5C per decade (Timmermans et al., 2017).

Looking ahead towards the future, past research has speculated whether the Arctic sea ice system might exhibit tipping behavior, abruptly transitioning to seasonally or year-round ice free conditions thanks to reinforcing positive feedback mechanisms (Lenton, 2012). However, the current research literature points towards a highly linear, predictable response of summer sea ice extent in response to greenhouse gas emissions (Niederdrenk and Notz, 2018; Ridley et al., 2007; Winton, 2011). Even in the absence of nonlinear feedbacks, however, the Arctic is likely already in the process of transitioning towards ice-free summer conditions, with the first ice-free summer potentially occurring before mid-century as assessed by a recent review (Notz and Stroeve, 2018). The complete loss of summer Arctic sea ice in some years would represent an important shift for regional climate and ecology, triggering significant consequences for regional warming and for Arctic biodiversity. For high levels of warming, a possibility also emerges of a more rapid transition towards an ice-free Arctic in winter, once ice-free summer conditions become more common (Bathiany et al., 2016; Eisenman and Wettlaufer, 2009).

### 3.4.2 Rapid decline of Arctic sea ice extent risks episodic ice-free summers before mid-century

In contrast to many of the longer-term climate feedbacks covered in this report, the loss of summer Arctic sea ice will likely occur within current lifetimes. Model projections predict a high likelihood of ice-free summers in the second half of the 21st century without strong climate mitigation (Massonnet et al., 2012; Stroeve et al., 2012b). Some uncertainty does exist regarding the exact timing when ice-free summers might be expected to manifest, with estimates for the approximate date of onset of occasional seasonally ice-free years varying by 20 years (Meredith et al., 2019). However, the strong relationship between ice loss and greenhouse forcing coupled with current emissions rates indicates a reasonable likelihood for the first ice-free summer to occur before mid-century (Notz and Stroeve, 2018). Modeling studies suggest that nothing short of extremely aggressive climate mitigation is likely to reduce the likelihood of future summer sea ice loss; only for warming of no more than 1.5°C is sea ice generally still present in summer by end-of-century (Jahn, 2018; Notz and Stroeve, 2016).

A substantial weight of evidence points towards dramatic ongoing degradation of Arctic sea ice. Mean winter multi-year ice thickness has declined from 3.6 m to 1.9 m over a thirty-year period from ~1980-2010 (Kwok and Rothrock, 2009; Wadhams, 2012). From 1997 to 2007, the Arctic ice cap has shrunk by an area of 1.5 million $km^2$ (Nghiem et al., 2007). Reduction in sea ice extent over the observational record has at times outpaced older IPCC model projections (Stroeve et al., 2007). The decline in Arctic sea ice is statistically attributable to a strong anthropogenic forcing (Notz and Marotzke, 2012).

Several positive feedbacks may be responsible for the rapid pace of shrinking sea ice extent. Sea ice loss is marked by an increasing transition from multi-year to seasonal sea ice, which is thinner and therefore more vulnerable to melt (Haine and





Martin, 2017). First-year sea ice has increased from 40% to 60-70% of Arctic sea ice area between 1984 And 2018 (Stroeve and Notz, 2018). Reduced cooling from lowered summertime cloud cover may also be exerting a potential effect (Kay et al., 2008). Finally, the well-known ice-albedo feedback, in which the melt-driven substitution of highly-reflective sea ice for
highly-absorptive dark open ocean waters results in increased surface inputs of solar energy, contributes to further ice loss via ocean and lower atmospheric warming.

Some negative or stabilizing feedbacks do exist however, moderating the pace of sea ice loss. Thinner first-year sea ice areas grow more rapidly during periods of freezing, while large areas that are ice-free or dominated by first-year ice lose heat to the
atmosphere faster during winter, allowing thin ice cover to regrow quickly over large areas (Eisenman, 2012; Notz, 2009; Sturm and Massom, 2016; Wagner and Eisenman, 2015). While the later onset of freezing is a contributing factor to reduced sea ice extent, this shift in timing also limits snow accumulation that would otherwise insulate the ice and inhibit growth (Sturm and Massom, 2016). These effects mean that years with very low summer ice coverage could enhance recovery of ice area the following winter, leading to a rebound in ice extent (Bitz and Roe, 2004).

While loss of summer Arctic sea ice represents a high-likelihood outcome under current rates of warming, a totally ice-free Arctic year-round remains a relatively much less probable scenario. Loss of winter Arctic sea ice is assessed to be possible, but only under worst-case emissions scenarios (RCP8.5) (Bathiany et al., 2016; Winton, 2006). One modeling analysis found that ice-free winter conditions required around 13°C mean annual warming at the north pole (Winton, 2006), and is thus
consequently unlikely to occur this century. Loss of winter sea ice may however proceed more abruptly than the long-term decline in summer sea ice extent to date, potentially occurring just a few years after the ocean becomes too warm to form ice in winter (Bathiany et al., 2016). The homogenously thin state of winter sea ice under conditions of frequent ice-free summers further contributes to its rapid loss (Bathiany et al., 2016). At the same time, loss of winter sea ice is thought to be reversible based on modeling results, merely requiring that winter sea temperatures fall back below freezing thresholds (Armour et al.,
2011; Bathiany et al., 2016; Li et al., 2013; Ridley et al., 2012).

Areas of uncertainty do remain when attempting to model future Arctic sea ice area. Cloud feedbacks that are important for determining summer sea ice dynamics still remain challenging even for state-of-art models (Tietsche et al., 2011). Poor representation of the varying distribution of sea ice thicknesses over the Arctic region is another weakness of current models
(Holland et al., 2011; Stroeve et al., 2014), likely due to varying ability to represent atmospheric circulation's influence on ice thickness (Bitz et al., 2002; Kwok and Rothrock, 2009; Stroeve et al., 2014). Nevertheless, such potential sources of error do not meaningfully affect the conclusion that ice-free summers across the Arctic Ocean will likely begin to occur starting before mid-century.



### 3.4.3 Loss of summer sea ice accelerates regional warming with global implications

With much of the region under permanent or extended daylight hours during summer months, ice-free conditions lead to a high potential for accelerated regional warming via the aforementioned ice-albedo feedback. Reductions in Arctic sea ice extent are implicated in enhanced Arctic warming that cannot be alternatively explained by cloud cover changes, internal climate variability, or variability in atmospheric and oceanic circulation (Screen et al., 2014; Screen and Simmonds, 2010; Serreze et al., 2009). Modeling analysis also indicates a strong sensitivity of Arctic temperatures to sea ice loss (Screen et al.,

2014). This effect has consequently played a strong role in driving rates of regional warming at more than double the global average rate, and is expected to continue to contribute to regional amplification of warming (Meredith et al., 2019).

This ice-albedo feedback additionally influences global climate at large. A modeling analysis examining a future Arctic with seasonally absent sea ice in summer found that an ice-free summer Arctic accelerated the rate of global warming by ~8%,

increasing radiative forcing by 0.3 W/m$^2$, a warming contribution equal to that of atmospheric halocarbons (Hudson, 2011).

The regional warming impact of significant reductions in Arctic sea ice extent also presents an indirect threat to global sea level rise by promoting additional melt from the Greenland Ice Sheet, an interaction identified as important over the geologic past in paleoclimate models (Koenig et al., 2014).

Finally, the loss of summer Arctic sea ice carries significant ecological implications. Wildlife dependent on sea ice for shelter or survival may be severely impacted, and the transition to ice-free summer conditions will also likely cause substantial shifts to phytoplankton community structure, driving transitions in regional marine ecology (Meredith et al., 2019). These impacts to wildlife and fishing will likely present economic, social, and cultural challenges for human communities across the Arctic.

We note that a year-round ice-free Arctic remains a scenario not currently believed to be plausible until 2100 even under worst-case warming scenarios (Bathiany et al., 2016; Lenton, 2012; Winton, 2006). Furthermore, loss of Arctic sea ice can be reversed on shorter timescales than most of the other climate mechanisms discussed in this review, as sea ice extent can recover within decades if initial warming is reversed (Notz, 2009; Ridley et al., 2012). At present however, the Arctic sea ice system

confronts an accelerated rate of warming, a rapid pace of sea ice decline, and an increasing likelihood of ice-free summer conditions occurring within a couple decades.

### 3 Conclusion

Considering the tipping elements that we have discussed above as an ensemble (Table 6), a number of useful commonalities emerge that highlight shared characteristics between some of these systems as well as important priorities for guiding future

research.



We note that relatively few of the tipping elements covered in this review are thought to demonstrate a potential for abrupt (within a couple decades) dramatic change that would categorize them as tipping points. A number of tipping elements, such as gradual permafrost thaw or Arctic sea ice decline, are arguably more accurately characterized as climate feedbacks due to
their more predictable response to increasing global temperature (Notz and Stroeve, 2016; Schneider Von Deimling et al., 2015).

Several of the most frequently-invoked climate tipping elements - permafrost thaw (McGuire et al., 2018; Turetsky et al., 2020), methane hydrate release (Archer et al., 2009; Kretschmer et al., 2015; Mestdagh et al., 2017), and ice sheet collapse
(DeConto and Pollard, 2016; Edwards et al., 2019) - cumulatively act on longer timescales of centuries to millennia, as opposed to driving rapid additional warming or sea level rise in a matter of years to decades. In the case of other tipping elements (AMOC shutdown/slowdown, methane hydrate release, monsoonal regime shifts), their likelihood of occurrence remains unclear and a topic of continued active research (Boos and Storelvmo, 2016a; Good et al., 2018; Ruppel and Kessler, 2017). For other systems such as the northern boreal forests, complex, competing feedbacks and ecological interactions complicate
efforts to assess overall climate impacts. Nevertheless, a number of tipping elements or climate feedbacks possess both a high certainty of occurrence this century and a relatively strong underlying scientific understanding (Arctic summer sea ice loss, coral reef collapse, Amazon forest dieback) (Frieler et al., 2013; Malhi et al., 2009; Notz and Stroeve, 2018; Salazar and Nobre, 2010).

The stratocumulus cloud deck evaporation hypothesis and the potential for AMOC slowdown/collapse sit in a unique space. Stratocumulus cloud deck evaporation remains relatively understudied and requires very high concentrations of greenhouse gases to initiate (Schneider et al., 2019). Given the large potential climate impacts of stratocumulus cloud deck breakup, however, constraining the likelihood of and triggering conditions for this powerful climate feedback deserves focused additional research. At the same time, our current state of understanding regarding the response of AMOC to climate change
remains incomplete (Collins et al., 2019; Good et al., 2018), and this feedback's potential importance to regional and global climate similarly demands further research attention.

Overall, our synthesis highlights that tipping elements will play a climatically significant role over the course of the 21st century, but that the possibility of a "tipping point cascade" driving a couple or more degrees of additional warming over the
next century remains unlikely. Simultaneously, this review does also point towards considerable remaining knowledge gaps associated with estimating the probability and impacts of many climate tipping elements. A continued need exists in particular for detailed modeling of the potential influence of tipping elements and carbon cycle feedbacks upon global climate. As scientific understanding of each tipping element or feedback improves further, the potential for abrupt climate change and for tipping element interactions will require ongoing re-assessment.



We do note that our report particularly highlights three tipping elements that pose a more imminent threat of climatically significant carbon release (Figure 9), namely boreal forest dieback, permafrost carbon release, and Amazon forest dieback. Both the tropical Amazon biome as well as the forests of the boreal north represent large repositories of carbon potentially vulnerable over the next century to large-scale ecosystem shifts (Cox et al., 2013; Mcguire et al., 2009). Meanwhile, although
the bulk of permafrost carbon release will take place over centuries, large uncertainties remain regarding how much carbon could be liberated from permafrost in the near term (Turetsky et al., 2020). Both the Amazon and the Arctic are ecosystems currently under significant stress that could shift quickly enough to release climate-altering quantities of greenhouse gases, yet substantial uncertainties remain. These yet-unresolved questions fundamentally control how much climate change the Amazon rainforest and northern Arctic could independently drive. With the fate of biomass carbon in both ecosystems also closely tied
to human land management practices, forest conservation policy in Brazil, Russia, Canada, Scandinavia, China and north central Asia may carry weighty climate implications. Both the Amazon as well as the circumpolar north consequently represent key priorities among the systems examined in this review.

The current literature also suggests that most of the tipping elements and climate feedbacks covered here are not yet committed
to fixed, dramatic changes in response to warming. This further emphasizes the significant potential for climate mitigation to moderate the impacts of tipping elements. Modeling of carbon release and other impacts associated with tipping elements such as ice sheet loss, Arctic sea ice decline, permafrost carbon release, and AMOC weakening point towards large differences between worst-case and strong mitigation emissions pathways. Limiting deforestation is also understood to play a key role in maximizing the long-term health of the Amazon tropical forest (Nobre et al., 2016). Decisive actions to cut greenhouse gas
emissions and reduce land-use impacts, potentially in combination with deployment of negative emissions technologies at scale, can thus dramatically reduce the future consequences associated with many of the tipping elements covered in this report.

Nevertheless, tipping elements presently remain one of the larger unknowns involved in predicting future warming and climate impacts. Continuing to better constrain the range of uncertainty surrounding likelihood and impacts of climate tipping elements
represents a particularly valuable contribution that the earth science community can offer to benefit policymakers and planners seeking to assess risks and craft effective climate policies moving forwards. In addition, given the acute interest and active discussion of tipping elements within the media and popular discourse, effective communication of scientific findings to a broad audience will continue to play an important role in promoting more informed general awareness of climate and earth science. For the foreseeable future, research into climate tipping elements will consequently remain of high value and in high
demand both within and beyond the research community.

**Code availability**



Python code and data files needed to replicate our simple FAIR modeling exercise are available hosted via GitHub at https://zenodo.org/badge/latestdoi/249821930.

**Author contributions**

SW was responsible for literature review, core writing, and figure generation for the article. ZH conceived of the article concept, contributed to specific sections, and provided editing and input on the article as a whole. Both SW and ZH worked together on the illustrative simple climate model exercise.

**Acknowledgements**

No dedicated grants funded this research. Funding sources for the Breakthrough Institute can be found here: https://thebreakthrough.org/about/who-we-are/funders. We thank Sijia Zou, Nicholas Foukal, David Archer, Olivier Gagliardini, Ted Schuur, Dirk Notz, William Boos, Laifang Li, Deepti Singh, Daniel Nepstad, Laura Borma, Beth Lenz, Heidi Hirsch, Tapio Schneider, Jaquelyn Shuman, Adrianna Foster, and Richard Betts for their valuable help in reading individual sections and providing comments and feedback. We also thank Merritt Turetsky, Paul Baker, Patrick Brown, and Brad Murray
for helpful conversations.

**Competing interests**

The authors declare no conflict of interest.

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

| Tipping element | Classification | Irreversible? |
|---|---|---|
| AMOC weakening/collapse | Weakening: tipping element<br>Collapse: tipping point | Irreversible. |
| Methane hydrate destabilization | Tipping element | Irreversible. |
| Greenland and Antarctic ice sheet loss | Tipping element | Greenland: Can be halted.<br><br>Antarctic: Irreversible. |
| Permafrost carbon release | Climate feedback | Irreversible. |
| Boreal forest ecosystem shifts | Tipping element | Uncertain |
| Stratocumulus cloud deck evaporation | Tipping point | Irreversible. |
| Coral reef habitat collapse | Tipping point | Irreversible. |
| Amazon rainforest dieback | Tipping element | Irreversible. |
| Abrupt transitions in S. Asian, African monsoon regime | Climate feedback | Reversible. |
| Loss of summer Arctic sea ice | Climate feedback | Reversible. |
| Tipping point cascade | Tipping element | Irreversible. |

**Table 1: List of tipping elements discussed in this review, classified as either tipping elements, tipping points, or climate feedbacks based on the terminology of (Kopp et al., 2016), and as reversible or irreversible system state changes.**





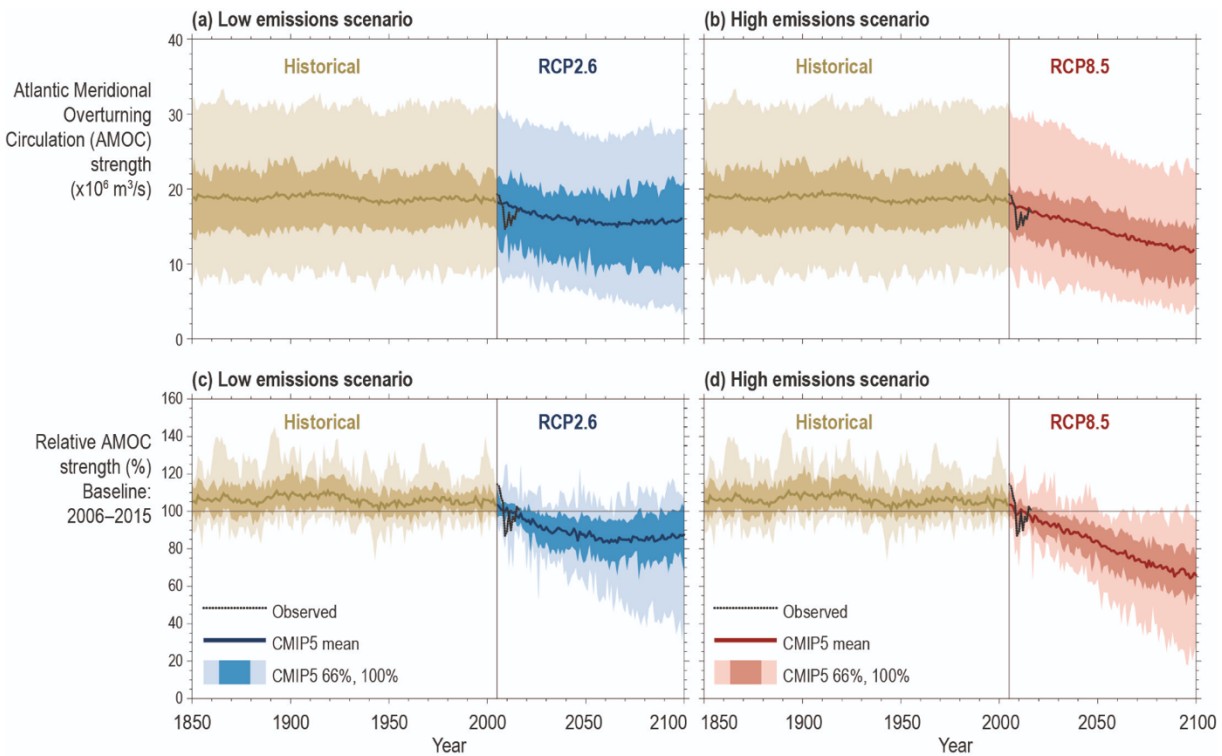

**Figure 1: Model simulations of AMOC strength at 26°N under (a, c) low-emissions (RCP2.6) and (b, d) worst-case emissions (RCP8.5) scenarios. Dark black line indicates observational estimates determined from the RAPID instrument array at 26°N. Gray, blue, and red lines show multi-model ensemble mean values (27 models in total). Darker and lighter shading show 66% and 100% ranges for model runs, respectively. Model results are expressed in estimated AMOC water mass transport rates (a, b) and in relative percentage change compared to the 2006-2015 baseline (c, d). Figure reproduced with permission, originally published as Figure 6.8 in the IPCC SROCC Ch 6 (Collins et al., 2019).**

| Methane source | Emission rate (Mt CH₄ per year) | | | |
|---|---|---|---|---|
| | Kirschke et al. 2013 (bottom-up estimates), IPCC AR5 estimate for methane hydrate release. | Schwietzke et al. 2016 | Saunois et al. 2016 (range of reported top-down and bottom-up estimates, 2003-2012) | Hmiel et al. 2020 |
| Wetlands | 217 (177-284) | | 127-235 | |
| Agriculture* | 200 (187-224) | | 115-243 | |
| Fossil fuels* | 96 (85-105) | 145±23 | 90-141 | 177±37 |
| Geological | 54 (17-97) | 51±20 | 35-76 | 1.6 |
| Freshwater | 40 (8-73) | | 60-180 | |







| Biomass burning* | 35 (32-39) | | 15-53 | |
| Wild animals | 15 (15-15) | | 5-15 | |
| Termites | 11 (2-22) | | 3-15 | |
| Methane hydrates | 6 | | 0-5 | |
| Wildfires* | 3 (1-5) | | 1-5 | |
| Permafrost | 1 (0-1) | | 0-1 | |

**Table 2: Current estimates for sources of atmospheric methane in Mt CH$_4$ yr$^{-1}$ based on (Kirschke et al., 2013), (Schwietzke et al., 2016), (Saunois et al., 2016), and (Hmiel et al., 2020). Asterisks denote anthropogenic methane sources, including wildfires. When combining a range of multiple estimates, uncertainties are not shown.**


**Figure 2: Schematic diagram of marine and sedimentary sources and sinks for methane and carbon dioxide derived from gas hydrate deposits, along with timescales required for warming to reach effective depth within methane hydrate stability zone.**






| Carbon stock | Total carbon (Gt C) |
|---|---|
| Methane hydrates (2017 estimate) | 1800 Gt C (Ruppel and Kessler, 2017) |
| Methane hydrates (1988 estimate) | 11000 Gt C (Kvenvolden, 1988) |
| Fossil fuels | 5000 Gt C |
| Soil carbon | 1400 Gt C |
| Dissolved marine carbon | 980 Gt C |
| Land biomass | 830 Gt C |
| Peat | 500 Gt C |
| Other | 67 Gt C |

**Table 3: Comparison between updated and initial estimates of carbon stored within methane hydrates, relative to other major stocks of mobile carbon on Earth. Adapted from charts published in (Ruppel and Kessler, 2017)**

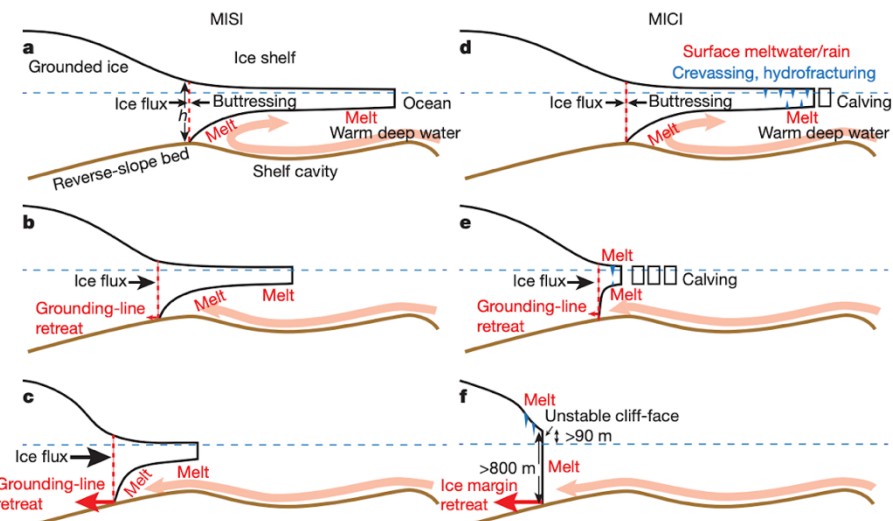

**Figure 3: Schematic diagram of the (a, b, c) marine ice shelf instability (MISI) and (d, e, f) marine ice cliff instability (MICI)**
**processes. In MISI, a) ocean and atmospheric heating induce melt proceeding from ice shelf edges, b) causing the ice sheet's grounding line to retreat onto reverse-sloping bedrock. c) As the height of the glacier above the grounding line grows, the ice sheet flows out to see faster in a positive feedback. According to the MICI hypothesis, a) loss of ice shelves due to iceberg calving and surface and subsurface heating leads to b) retreat of the ice sheet onto reverse-sloping bedrock, followed by c) exposure of progressively higher ice cliff faces that unstably collapse, driving rapid retreat. Figure originally published in (DeConto and Pollard,**
**2016).**


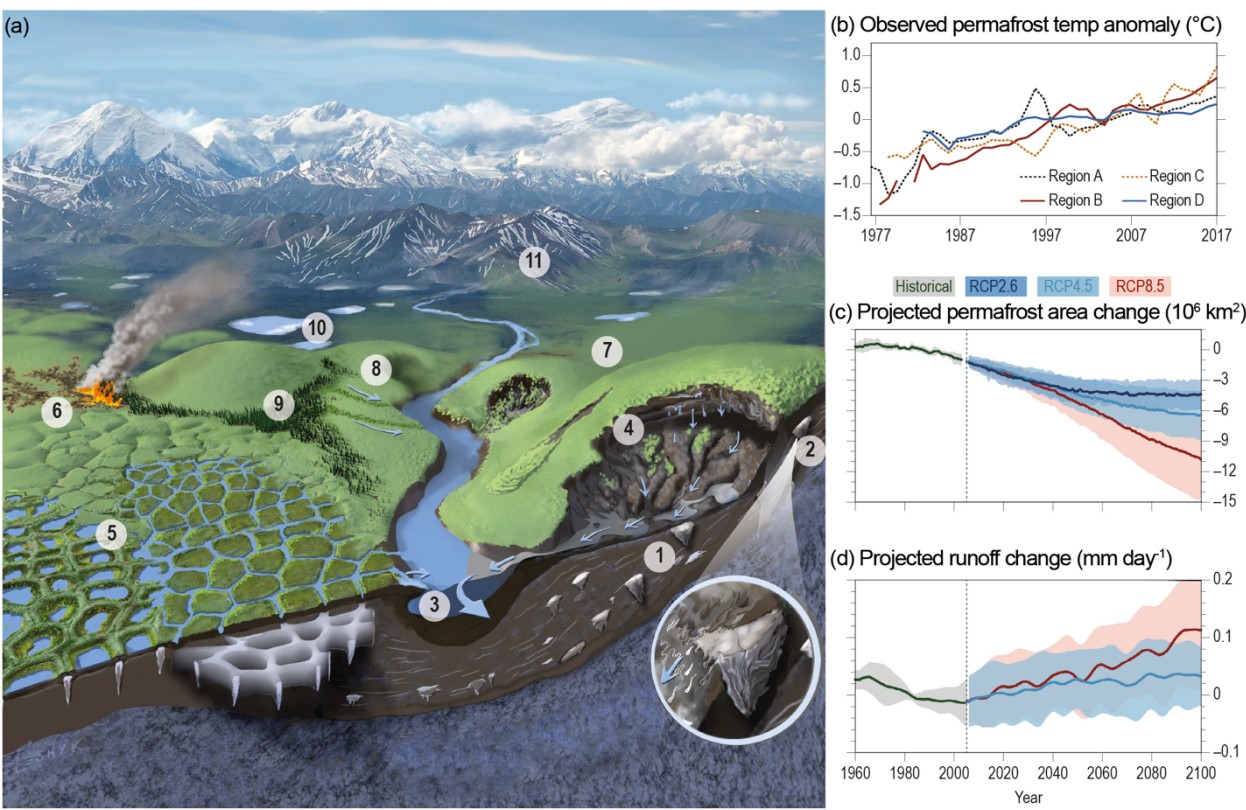

**Figure 4: a) Diagram illustrating components of the circumpolar Arctic terrestrial landscape potentially sensitive to climate change, including 1) permafrost, 2) ground ice, 3) river flow rates, 4) abrupt thaw via landslides and small-scale processes, 5) surface water accumulation, 6) wildfires, 7) tundra biome shifts, 8) shrub vegetation, 9) boreal forests, 10) lake ice, and 11) seasonal snow cover. b) Observed permafrost temperature anomalies (relative to the 2007-2009 International Polar Year baseline) over time** (Romanovsky et al., 2017b). **c) global area of near-surface permafrost soils and d) projected change in surface runoff to the Arctic Ocean under low (RCP2.6), middle-of-the-road (RCP4.5), and worst-case (RCP8.5) emissions scenarios. Figures adapted with permission from Figure 3.10 in the IPCC SROCC Ch 3 (Meredith et al., 2019)**

| | Boreal forest zone | | |
|---|---|---|---|
| | **Southern margins** | **Boreal zone interior** | **Northern margins** |
| **Warming feedbacks in response to warming** | Soil organic matter decomposes at higher rates<br><br>Increased wildfire impacts | Soil organic matter decomposes at higher rates<br><br>Increased wildfire impacts | Soil organic matter decomposes at higher rates<br><br>Increased wildfire impacts<br><br>Expansion of boreal conifers |



| | Increased boreal tree mortality from pests | Increased boreal tree mortality from pests | into previously bare ground darkens land surface |
|---|---|---|---|
| **Cooling feedbacks in response to warming** | Carbon storage eventually shifts aboveground into more robust deciduous vegetation<br><br>Replacement of dark-leaved boreal conifers increases land surface reflectivity | Carbon storage eventually shifts aboveground into more robust deciduous vegetation<br><br>Replacement of dark-leaved boreal conifers increases land surface reflectivity | Faster vegetation growth due to warmer temperatures stores more carbon |
| **Net effects** | Highly uncertain, but with a considerable possibility for net warming due to increased carbon release from decomposition and burning of organic matter in boreal forest soils.<br><br>Climate impacts may also differ by region (i.e. E. Siberia vs. W. Siberia vs. N. Canada vs. Alaska) | | |

**Table 4: Conceptual outline of changes within different regions of the boreal zone organized by their anticipated impacts upon climate.**

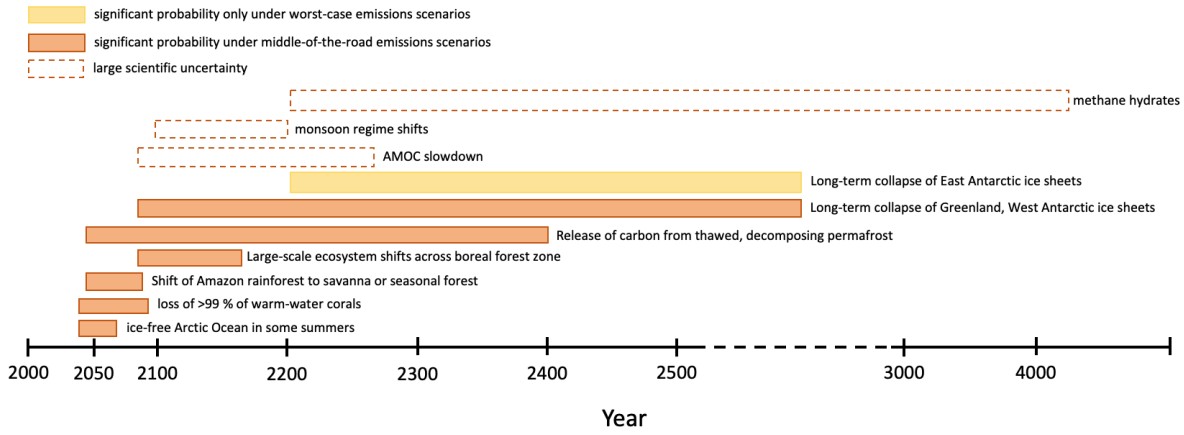

**Figure 5: Illustration of the approximate timeframes over which tipping elements and feedbacks might be expected to take place.**


| Tipping element | RCP4.5 TIPP | | RCP8.5 TIPP | | Reference |
|---|---|---|---|---|---|
| | $CO_2$ emissions/yr, Gt C from 2020-2100 | $CH_4$ emissions/yr, Mt C from 2020-2100 | $CO_2$ emissions/yr, Gt C from 2020-2100 | $CH_4$ emissions/yr, Mt C from 2020-2100 | |
| **Gradual permafrost thaw** | 0.667 | 13.58 | 1.074 | 13.58 | (Schneider Von Deimling et al., 2015) |




| | 0.039 | 70.74 | 0.141 | 82.93 | (Turetsky et al., 2020) |
|---|---|---|---|---|---|
| **Abrupt permafrost thaw** | 0.039 | 70.74 | 0.141 | 82.93 | (Turetsky et al., 2020) |
| **Methane hydrate destabilization** | 0 | 5.84 | 0 | 11.68 | (Kretschmer et al., 2015) |
| **Boreal forest ecosystem shifts** | 0.494 | 0 | 0.494 | 0 | (Steffen et al., 2018) |
| **Amazon forest dieback** | 1.078 | 0 | 1.078 | 0 | (Steffen et al., 2018) |

**Table 5: Additional CO₂ emissions, CH₄ emissions, and radiative forcing incorporated into RCP4.5 and RCP8.5 emissions pathways to simulate added climate impact of tipping elements based on available estimates from literature.**


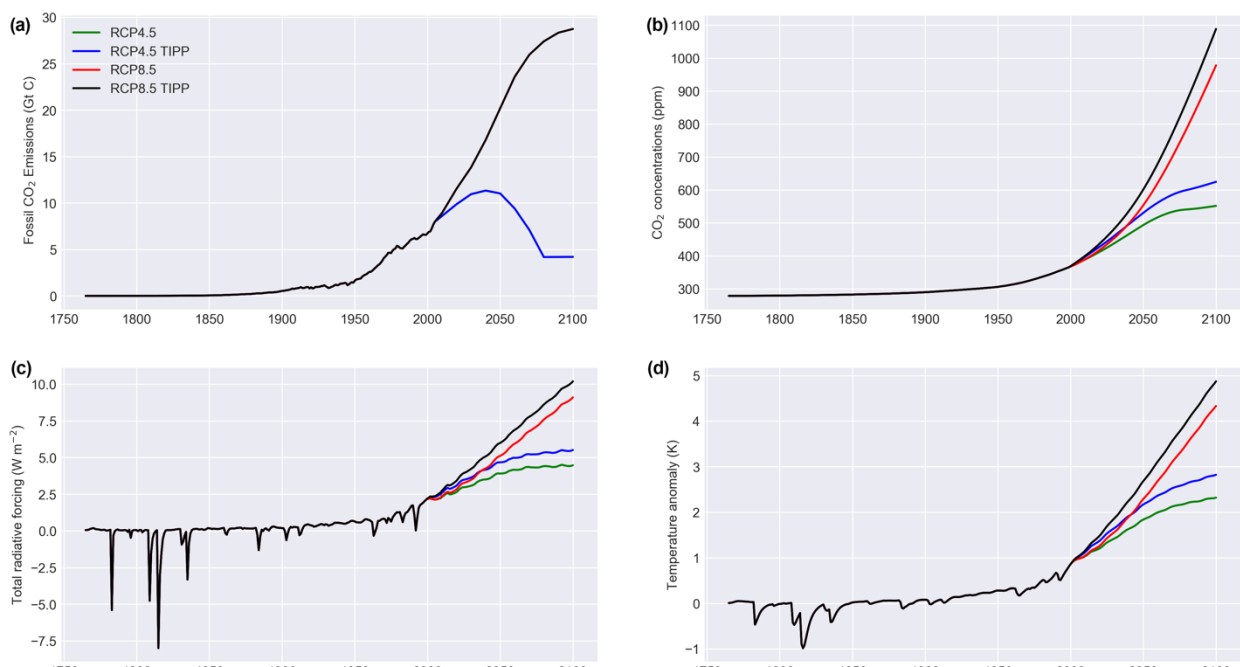

**Figure 6: a) fossil fuel CO₂ emissions, b) atmospheric CO₂ concentrations, c) total radiative forcing, and d) global mean surface temperature anomaly relative to pre-industrial (1861–1880 mean) temperatures for the RCP4.5 and RCP8.5 emissions scenarios as well as for modified RCP4.5 and RCP8.5 scenarios incorporating additional CO₂ emissions, CH₄ emissions, and radiative forcing as a result of tipping elements (see Table 4)**




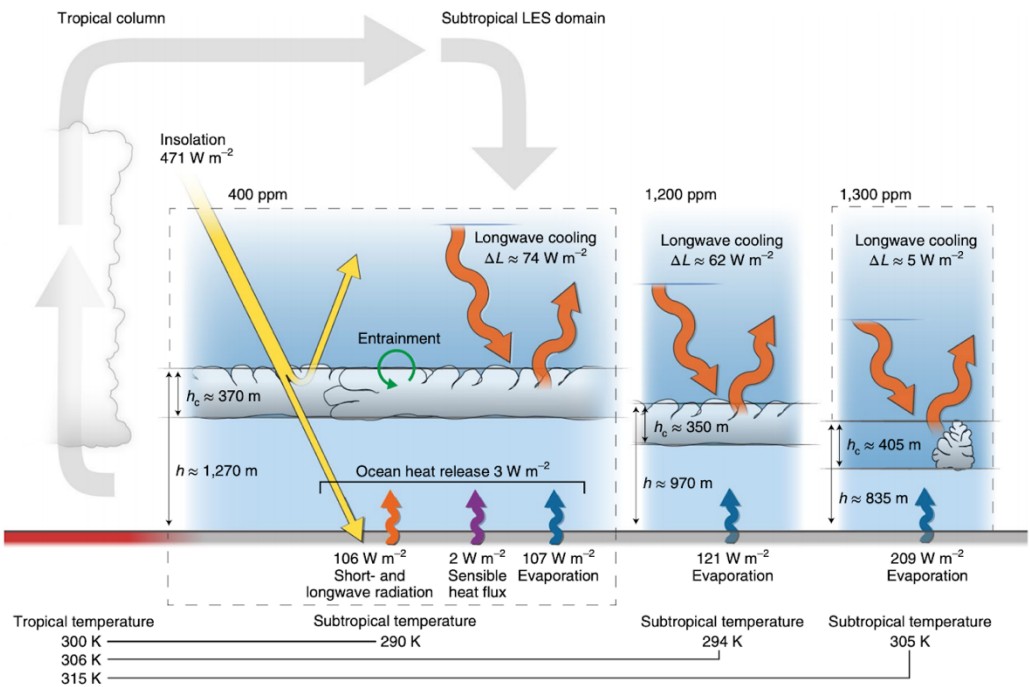

**Figure 7: Schematic diagram illustrating mechanisms of stratocumulus cloud deck collapse and resulting consequences upon net radiative energy balance. Under present-day conditions (400ppm), stratocumulus cloud decks are sustained by longwave radiative cooling at their topmost extents, which drives convective circulation that resupplies moisture to the cloud layer from the ocean's surface. Increasing greenhouse gas concentrations (1200ppm) increase the input of longwave radiation to stratocumulus clouds from above, reducing the strength of longwave cooling. Beyond a certain threshold of greenhouse gas levels (1300ppm), longwave cooling weakens to the point where cloud decks are cut off from surface moisture, leading to their disintegration and abrupt, acute surface warming from increased absorption of sunlight. Figure originally published in (Schneider et al., 2019)**







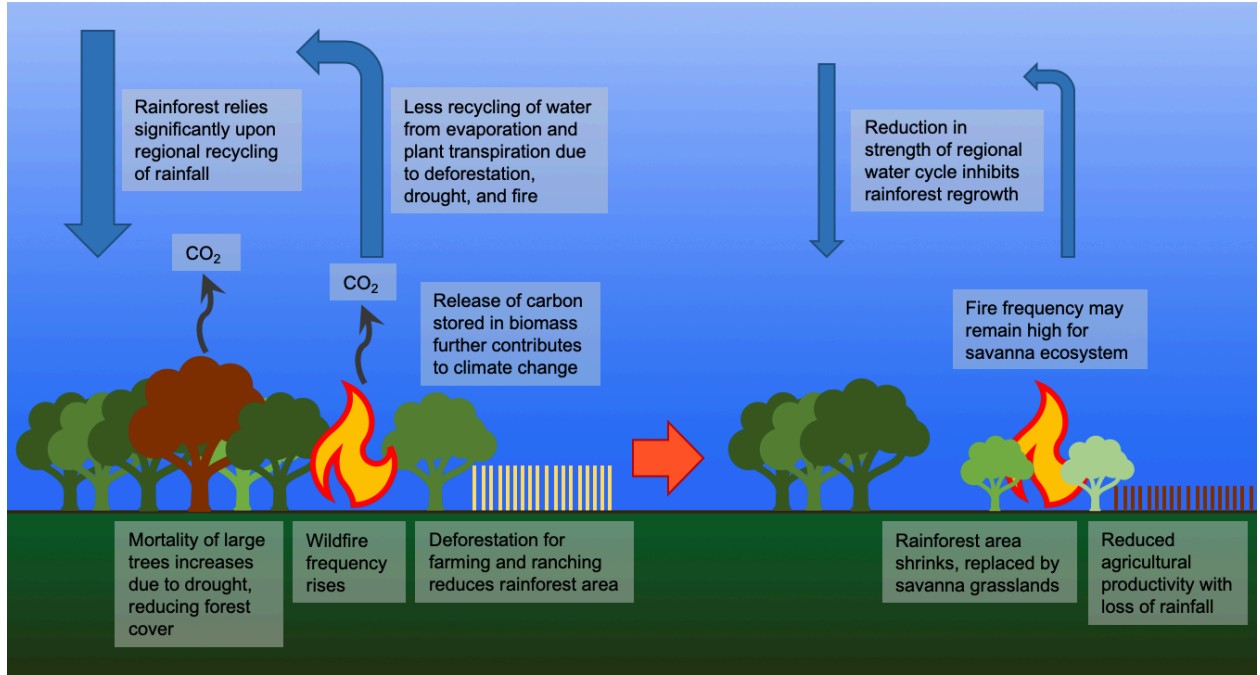

**Figure 8: Schematic diagram of causes, feedbacks, and impacts associated with Amazon rainforest mortality.**

| Tipping element | Level of scientific understanding | Predictability by models | Key thresholds | Climate impact | Timescale of impacts |
|---|---|---|---|---|---|
| AMOC weakening/ collapse | Moderate | Good agreement, significant model limitations | Uncertain | Collapse unlikely. Weakening causes regional cooling, wind, precipitation, sea level changes | Weakening occurs over centuries. Collapse would be abrupt. |
| Methane hydrate destabilization | Moderate | Low | Uncertain, long-term impacts higher beyond ~3C | Gradual long-term release of methane to atmosphere | Centuries to multiple millennia |
| Greenland and Antarctic ice sheet loss | Moderate to high | Moderate | Greenland: ~2C<br><br>West Antarctica: 2-3C | Multi-meter sea level rise over centuries to millennia, irreversible ice sheet loss | Centuries to millennia |



| | | | East Antarctica: Uncertain | | |
|---|---|---|---|---|---|
| **Permafrost carbon release** | **Moderate** | **Moderate to low** | **No firm threshold behavior** | **Added emissions of carbon dioxide and methane** | **Years to decades, continuing for centuries** |
| **Boreal forest ecosystem shifts** | **Low** | **Low** | **~3-4C** | **Increase in wildfires, significant changes in soil and biomass carbon storage, regional albedo changes, major ecosystem shifts** | **Decades to a century** |
| **Stratocumulus cloud deck evaporation** | **Low** | **Low** | **~1200 ppm CO₂e** | **Worldwide marine cloud deck breakup triggers global warming of up to 8C** | **Unclear but may be very abrupt, with impacts within a decade** |
| **Coral reef habitat collapse** | **Very high** | **High** | **Increasingly severe impacts beyond 1.5C, with critical threshold at 2C.** | **Degradation of ~99% of warm-water coral habitats worldwide, major socioeconomic impacts** | **Decades** |
| **Amazon rainforest dieback** | **Good** | **Moderate** | **40% deforestation, ~3-4C, 40% precipitation decrease, or some combination** | **Die-off of significant fractions of Amazon rainforest, large ecosystem shifts, significant carbon emissions** | **Decades to a century** |
| **Abrupt transitions in S. Asian, African monsoon regime** | **Low** | **Moderate to low, but majority of models predict increase in monsoon area and rainfall.** | **Thought to be non-tipping climate feedback. Increase in regional albedo over India to 0.5 proposed by some.** | **Abrupt decline in South Asian monsoon strength. Abrupt increase in West African monsoon strength,** | **Decades, potentially unlikely to occur this century.** |
| **Loss of summer Arctic sea ice** | **High** | **Moderate to high** | **Summer ice loss scales linearly with temperature. Ice-free summers likely** | **Increased occurrence of ice-free summer Arctic, global warming feedback** | **Decades** |





| | | | for warming of ~2C. | | |
|---|---|---|---|---|---|
| Tipping point cascade | Low | Low | Uncertain | Significant additional global warming due to interacting climate tipping points | Centuries to millennia |

**Table 6: Summary of scientific understanding, key thresholds, and impacts associated with the tipping elements and climate feedbacks covered in this review.**

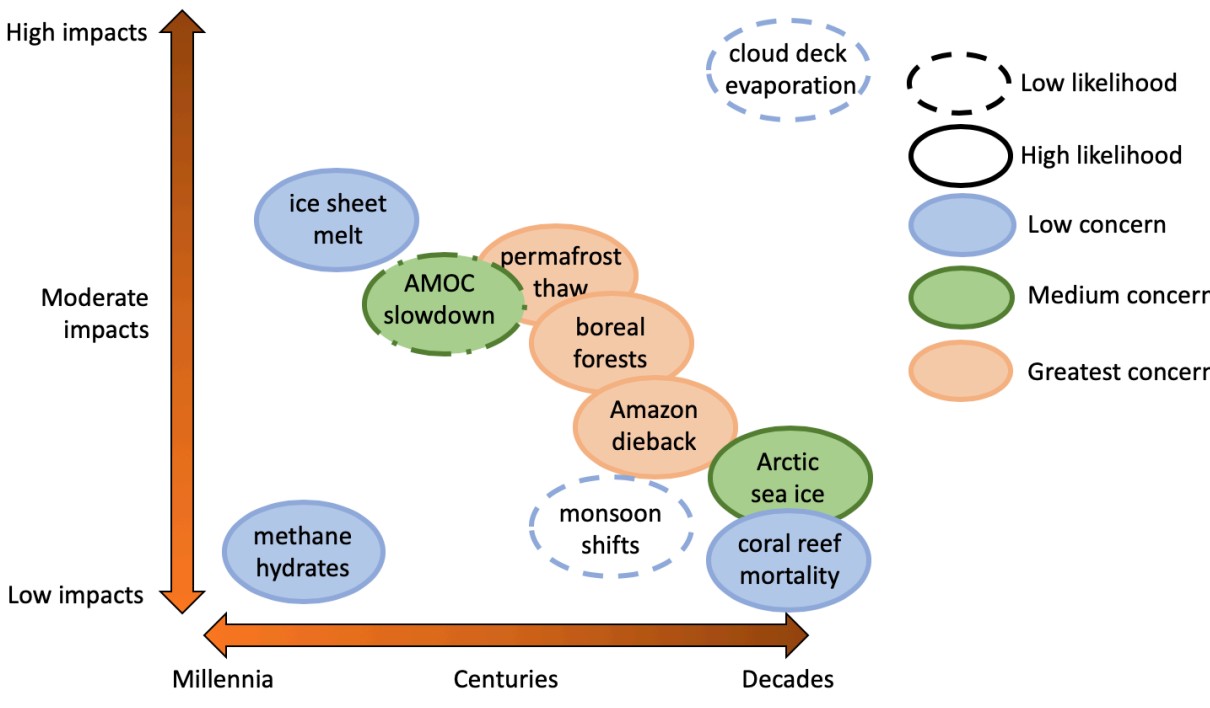

**Figure 9: Qualitative two-dimensional organization of individual tipping elements covered in this report. Tipping element**
**mechanisms are organized vertically according to expected global impacts (y-axis) and horizontally based on the timeframe over which those effects will likely manifest. Tipping elements are qualitatively color-coded based on a combination of their likelihood of occurrence, their impacts on global society and ecosystems, their ability to drive abrupt additional global warming, and the current proximity of the climate/ecosystem thresholds at which these feedbacks are triggered.**
