# Peer review of "ESD Reviews: mechanisms, evidence, and impacts of climate tipping elements"

_Earth System Dynamics, 2020_

## Short Comment (SC1) · 21 Apr 2020

Line 610 to 615 There is a relationship of methane oxidation to soil moisture content. Rate of production of methane from organic sources is unlikely to be temperature dependant as most kinetic equations including those used by IPCC are kinetically temperature independent (though the Winden et al, 2012 temperature observation may be related to the latent heat of fusion -melting of the ice/frost post winter in boreal bogs where there would be rapid boost in methane generation from the saturation of soil). The key factor is moisture content. Organic degradation produces equal molar quantities of methane and CO2 from about 40% moisture (The water saturation point of each

molecule of the simplest sugar glucose). Below 40% soil moisture the methane arising is increasingly oxidised to CO2, probably in logarithmic equation of the form [CH4] = aln[H20] +c (This is one of my modelling equations used to estimate landfill emissions and represents the non-constant moisture variable in a Fick Laws(1855) adaptation – the precursor of modern kinetics models used by IPCC – the moisture becomes constant over the lifetime of the landfill). Below 10% moisture the almost all of the methane is converted to CO2 ([CH4}=0%] when [H2O]<=10%] as shown in landfilled waste decomposition by Hartz and Ham 1978 (supported by other papers in text). Increased atmospheric temperature in post melting scenario may represent an evapotranspiration process, increasing moisture evaporation from the soil, introducing air (oxygen) to surface soils and thus increasing but variable methane oxidation as noted in Popp et al 2000).

Hartz KE and Ham RK (1983) Moisture level and movement effects on methane production rates in landfill samples. Waste Management & Research 1: 139-145, https://journals.sagepub.com/doi/pdf/10.1177/0734242X8300100116

Fick A (1855) On liquid diffusion. Annalen der Physik und Chemie. 94: 59. Reprinted in: Fick A (1995) On liquid diffusion. Journal of Membrane Science 100: 33–38. doi:10.1016/0376-7388(94)00230-v
* * *

---

## Short Comment (SC2) · 21 Apr 2020

Daniel Gilford, PhD (daniel.gilford@rutgers.edu) 4/21/20

Summary

Thank you for this paper, it is a helpful addition to the literature, and it will be well-suited for publication following review. I have made several comments specifically in the introduction and ice-sheet sections, which I hope the authors will consider and find useful for improving the manuscript. My primary suggestion is the inclusion of a more thorough discussion on how uncertainty affects the physics, projections, and

understanding of timescales of ice-sheet tipping points.

General Comments

Line 7: “Shifting towards” seems a bit odd in the definition of climate tipping elements. It might be better to frame tipping elements in this context as a clearing a threshold, rather than the sign of a derivative? This wording appears more like it is referring to a climate anomaly, rather than moving to/arriving at a new climate paradigm.

Line 35: This paragraph would be strengthened by including a mention of adaptation. Certainly once a tipping point has been passed, significant changes in planning, decision-making, and climate adaptation will need to be adopted.

Line 94: As written, it was unclear to me what “0.54 in the high-end of the RCP8.5 warming scenario” means, or refers to. It probably needs to be either explained more directly and with appropriate context, or should be mentioned generally and more specifically covered in section 4.

Section 2.3.1: For context and contrast with the ice-sheet changes, it should be noted that sea-level rise from other components (thermal expansion, mountain glaciers) is also irreversible… but these do not necessarily have tipping points and exhibit more immediate gains upon mitigation (e.g. Solomon et al. 2009, Lenaerts et al. 2013, Zickfeld et al. 2016, Ehlert et al. 2018).

Line 388: This paragraph would benefit from being very specific about what is meant by “less insulated”. Why are they less insulated than the EAIS? The reasons and time frames are quite different between GIS and WAIS because of the differences in their physical responses to climate change (see the Hamlington et al. 2020 review), so grouping them like this confuses the message. Specificity and expanding this section would probably help clarify the point of this paragraph as I understand it: that there are heterogeneous tipping points across the polar ice sheets.

Lines 400-405: Marine ice-sheet instability is confusingly explained in this section, and

will likely lose some readers. It might be helpful to refer to other papers which explain this process (especially less technical and more general papers about MISI, such as Robel et al. 2019), and rewrite for clarity.

Section 2.3.2: In the context of ice-sheet instabilities it would be helpful to more carefully introduce the flux rate across the grounding line as being proportional to the height at the grounding line, and then discuss where reverse sloping beds are found, and how that has the potential to affect the (in)stability of WAIS and EAIS (e.g. see topography in Le Brocq et al. 2010).

Line 409: 90m is not an exact threshold for MICI, it was a suggested model parameter value proposed in DeConto and Pollard (2016). Rather than speculate on these parameter values which are most relevant in a modeling context, it would be much more useful to instead rewrite this sentence describing the physical phenomenon being discussed, e.g., "MICI postulates that ice cliffs become unstable and collapse under their own weight if they exceed a critical height threshold (Pollard et al. 2015)"

Line 416: In introducing MICI, it would be useful to mention Clerc et al. (2019), which argues that MICI is unlikely given viscous relaxation dominating the response to ice-shelf removal. In contrast to Edwards et al. (2019) which makes a modeling/statistical argument, Clerc et al. makes a physical one.

Line 425: This paragraph is missing a key discussion on the deep uncertainty surrounding the Antarctic ice-sheet response. Deep uncertainty is characterized by the lack of agreement between experts (e.g. Bakker et al. 2017, Bamber et al. 2019), which has critical implications for decision-maker response to possible AIS tipping points (Rasmussen et al. 2020).

Line 425: Likewise missing is a discussion that our understanding of ice-sheet tipping points comes primarily from the paleoclimate record, at points in time when we know the AIS was at least partially deglaciated (e.g. Dutton et al. 2015, Capron et al. 2019). Current paleoclimate understanding has limited power in constraining our understanding of ice-sheet instabilities (Edwards et al. 2019), but improved geological estimates have potential to reduce future projection uncertainties (Gilford et al. 2020, in revision). Modern mass loss rates (e.g. from satellite measurements of SMB model estimates) have very little efficacy for reducing uncertainty on decadal time scales (Kopp et al. 2017).

Lines 445 and 482: It is unnecessary to continue mentioning the uncertainty highlighted by Edwards et al. (2019). Instead it would be better to flush out the original discussion in line 425, noting how it brings in deep uncertainty reflected in Bamber et al. (2019), and highlighting that anytime someone refers to a MICI scenario is a high-end or worst-case scenario being considered (such as the excellent line 477).

Section 2.3.3 and e.g. Line 490: It would be very helpful to reference the Bamber et al. (2019) expert judgement paper for projections; it is probably our best prior for projected ice-sheet mass losses.

Line 505: This should read "Consequently, under our current best understanding, ice-sheet collapse…" It's important to note that given the deep uncertainty (e.g. if MICI is possible), then changes could actually be abrupt. We just don't have a good enough grasp on the physics yet, and our observations (paleo and modern) are inadequate to constrain this property. Furthermore, because of these deep uncertainties, without progression of the science it may be impossible to know whether we are on a trajectory towards a tipping point (for instance, initated MICI) until we have already crossed it (e.g. Kopp et al. 2017, their Figure 5).

Suggestions for Minor grammatical/structural/citation changes:

-Line 7: add comma after "some"

-Line 28 and elsewhere: "sea level rise" should be "sea-level rise". Please adjust accordingly throughout (e.g. lines 356, 358, 377, 386), and likewise for places where "ice sheet" should be "ice-sheet" (e.g. line 368, 377, 386)

-Line 40: Should this be Turetsky et al. (2020)?

-Line 44: "larger uncertainty" is a strange word choice here. I think this is referring to ambiguity (Ellsberg 1961)

-Line 74: It will probably be addressed in type-setting, but this line is not formatted correctly

-Line 87-88: This sentences reads oddly as written, with subject confusion, etc. "the climate impact of the" should be "the climate impact of a", and "uncertainties surrounding factors required" is very wordy. It should be rewritten for clarity

-Throughout section 2.3: the acronyms for AIS, GIS, EAIS, and WAIS need to be introduced only once (in lines 360 and 371-372)

-Line 360: add "By contrast" before "The Greenland Ice Sheet"

-Line 361: can this be more specific than "recently"? Maybe "accelerated over the past four decades (IMBIE 2018)"

-Line 365: A full review of the current understanding of modern ice-sheet contributions to sea-level rise is available in the in press Hamilington et al. (2020) review

-Lines 370-374: This sentence is written very oddly and is difficult to follow. I suggest rewriting for clarity

-Line 377: Suggest rewriting this sentence as "Major ice-sheet processes have had a dominating influence on sea-level patterns..."

-Line 380: Add "global mean" before "increase of 3.4m"

-Line 392: Add "(no net sea-level rise)" after "mass balance", for clarity

-Line 392-393: Remove "and precipitation", it is redundant, and add that it is related to increased atmospheric moisture from increasing temperatures.

-Line 393: Add "this balance is" before "subject to considerable..." for clarity

-Line 395: Rewrite the beginning of this sentence as "Ice on the AIS margins is typically...", and remove "at their edges" from the end of the sentence

-Line 401: Suggest citing Weertman (1974) at the beginning of the MISI discussion

-Line 406: Add "can" after "heating"

-Line 409: Replace "is" with "would be"

-Line 412: Please add a citation for the 800m threshold

-Line 453: Should "proving" be "providing"?

-Line 456: "net" before "feedbacks" would help clarify this sentence

-Line 457: Is this referring to geological field observations?

-Line 457: One key warming threshold paper for GIS that should be mentioned is Robinson et al. (2012)

-Line 478: Add "(but very uncertain)" before "MICI feedback,"

-Line 489: Would this discussion be helped by adding a sentence or two about the regional variability of ice-sheet loss projections (e.g. Kopp et al. 2017)?

-Line 495: Do albedo feedbacks also make it more difficult to stabilize the GIS?

-Line 496: Add "ice-sheet" between "individual basins,"

-Line 517: "instability" might read better as "instabilities"

References:

Bakker, A.M.R., Louchard, D. & Keller, K. Sources and implications of deep uncertainties surrounding sea-level projections. Climatic Change 140, 339–347 (2017). https://doi.org/10.1007/s10584-016-1864-1

Clerc, F., Minchew, B. M., & Behn, M. D. ( 2019). Marine Ice Cliff Instability Mitigated

by Slow Removal of Ice Shelves. Geophysical Research Letters, 46, 12108– 12116. https://doi.org/10.1029/2019GL084183

Ehlert, Dana and Zickfeld, Kirsten (2018) Irreversible ocean thermal expansion under carbon dioxide removal. Earth System Dynamics, 9 (1). pp. 197-210. DOI 10.5194/esd-9-197-2018.

Ellsberg, D. (1961). Risk, Ambiguity, and the Savage Axioms. The Quarterly Journal of Economics, 75(4), 643-669. Retrieved April 21, 2020, from www.jstor.org/stable/1884324

Daniel M. Gilford, Erica L. Ashe, Robert E. Kopp, Robert M. DeConto, David Pollard, Alessio Rovere, 2019: Can the Last Interglacial Constrain Projections of Future Antarctic Ice Mass Loss and Sea-level Rise? JGR-Earth Surface (in revision Feb. 2020).

Hamlington, B. D., Gardner, A. S., Ivins, E., Lenaerts, J. T. M., Reager, J. T., Trossman, D. S., et al ( 2020). Understanding of Contemporary Regional Sea‐level Change and the Implications for the Future. Reviews of Geophysics, 58, e2019RG000672. https://doi.org/10.1029/2019RG000672

Kopp, R. E., DeConto, R. M., Bader, D. A., Hay, C. C., Horton, R. M., Kulp, S., Oppenheimer, M., Pollard, D., & Strauss, B. H. (2017). Evolving Understanding of Antarctic Ice-Sheet Physics and Ambiguity in Probabilistic Sea-Level Projections, Earth's Future, 5, 1217–1233, https://doi.org/10.1002/ 2017EF000663

Le Brocq, A. M., Payne, A. J., and Vieli, A.: An improved Antarctic dataset for high resolution numerical ice sheet mod- els (ALBMAP v1), Earth Syst. Sci. Data, 2, 247–260, 2010, http://www.earth-syst-sci-data.net/2/247/2010/.

Lenaerts, J. T. M., van Angelen, J. H., van den Broeke, M. R., Gardner, A. S., Wouters, B. and van Meijgaard, E. ( 2013), Irreversible mass loss of Canadian Arctic Archipelago glaciers, Geophys. Res. Lett., 40, 870– 874, doi:10.1002/grl.50214.

Pollard, D., DeConto, R. M. & Alley, R. B. Potential Antarctic Ice Sheet retreat driven by

hydrofracturing and ice cliff failure. Earth Planet. Sci. Lett. 412, 112–121 (2015).

Rasmussen, D. J., Buchanan, M. K., Kopp, R. E., & Oppenheimer, M. ( 2020). A flood damage allowance framework for coastal protection with deep uncertainty in sea level rise. Earth's Future, 8, e2019EF001340. https://doi.org/10.1029/2019EF001340

Alexander A. Robel, Hélène Seroussi, Gerard H. Roe: Marine ice sheet instability amplifies and skews uncertainty in projections of future sea-level rise. Proceedings of the National Academy of Sciences Jul 2019, 116 (30) 14887-14892; DOI: 10.1073/pnas.1904822116

Robinson, A., Calov, R. & Ganopolski, A. Multistability and critical thresholds of the Greenland ice sheet. Nature Clim Change 2, 429–432 (2012). https://doi.org/10.1038/nclimate1449

Susan Solomon, Gian-Kasper Plattner, Reto Knutti, Pierre Friedlingstein: Irreversible climate change due to carbon dioxide emissions. Proceedings of the National Academy of Sciences Feb 2009, 106 (6) 1704-1709; DOI: 10.1073/pnas.0812721106

Kirsten Zickfeld, Susan Solomon, Daniel M. Gilford: Centuries of thermal sea-level rise due to anthropogenic emissions of short-lived greenhouse gases. Proceedings of the National Academy of Sciences Jan 2017, 114 (4) 657-662; DOI: 10.1073/pnas.1612066114

---

## Short Comment (SC3) · 22 Apr 2020

In the text of my comment, the reference to Hartz and Ham 1978 should be Hartz and Ham 1983 - Apologies (shows I am a real person).
* * *

---

## Short Comment (SC4) · 4 May 2020

Nisbet et al (2019) and others provide evidence that rising atmospheric methane concentrations over the last 13 years may be driven in part by tropical wetlands responding to global warming. This effect is not explicitly parameterized in coupled atmosphere-ocean general circulation models. While the uncertainty about the effect is large, the available evidence indicating that the effect may already have significant impact on global methane levels further indicates that this effect should be included in any broad forward looking overview of potential "tipping elements" - such as this discussion.

[Figure]

2020.

---

## Short Comment (SC5) · 4 May 2020

Discussion should be widened to consider the abrupt loss of ocean sinks. Discussion of such appears limited to a single off-topic reference in the discussion of AMOC slow down. See, for example, Stevens et al 2020: https://www.nature.com/articles/s41558-020-0722-3

---

## Short Comment (SC6) · Here is the reference to Nisbet et al. · 4 May 2020

Nisbet et al, 2019: Very Strong Atmospheric Methane Growth in the 4 Years 2014–2017: Implications for the Paris Agreement. Biogeochemical Cycles. 5 Feb 2019. https://doi.org/10.1029/2018GB006009

Your policy to restrict comments to holders of a PhD and to ignore other comments is explicitly elitist and not in the interest of knowledge creation.

---

## Editor Comment (EC1) · Anders Levermann (Editor) · 4 May 2020

Dear Hunter Cutting,

this is a scientific journal and has to keep this standard. It is not for the general discussion. Your comment does not include the full reference of the paper that you refer to and can thus not be accounted for.

Thank you for your understanding and please do not comment on a scientific paper in a scientific journal if you are not a scientist.

The qualification of being a scientist is proven by a PhD thesis which has either been

completed or us being prepared under supervision by a professor.

Best wishes, Anders Levermann

---

## Short Comment (SC7) · 5 May 2020

Dear Anders Levermann:

Thank you for your response.

You state that your process is "elitist in a good way," but you fail to establish that.

You characterize my comments as "perfectly legitimate" yet you also find that my comments must be ignored lest they constitute "public dispute on climate change in an EGU peer-reviewed journal." Your logic here is self-contradicting (in the first instance) and circular (in the 2nd).

[Figure]

Simply stating that something must be so is not the same thing as establishing it.

Further, ignoring "perfectly legitimate" comments is a disservice to the science.

I appreciate the time you have taken to share your views.

I understand my comments are not welcome here. So, I will not comment further in this thread (understanding you may comment again) or elsewhere on this platform.

Thank you for your attention here.

Best regards,

Hunter Cutting
* * *

---

## Editor Comment (EC2) · Anders Levermann (Editor) · 5 May 2020

Dear Hunter Cutting,

I am sorry to disappoint you and am grateful for your contribution. We have to draw the line somewhere, especially in an open review process which has to remain a peer-review. I understand that this can be considered elitist, but in this sense it is a elitist in a good way. You are free to comment and address the author's directly. All emails are openly available. I hope you understand that even though your comments were perfectly legitimate, we can not provide a platform for a public dispute on climate change in an EGU journal peer-review. This is not a science chat room. We have to protect

the authors and other reviewers. The scientific debate is open and free, especially with the many available channels of communication these days, but the journals have to be a peer-review world.

I hope you understand. Best wishes, Anders

————————————————

---

## Author Comment (AC1) · 15 Jun 2020

We thank the reviewer for these thoughtful comments, and have added additional text to further emphasize the importance of soil moisture in modulating methane fluxes from thawed permafrost. The suggestion that methane production from peat bogs is unlikely to be dependent on temperature is inconsistent with field and laboratory studies that indicate a strong temperature relationship (e.g. Blanc-betes et al. 2016, Metje and Frenzel 2007, Rivkina et al. 2004, van Winden et al. 2012). The Winden et al. finding, which utilized incubated peat samples, would not be influenced by seasonal transitions in soil moisture content. This said, the reviewer's points regarding the role of

soil moisture in governing the relative production of methane and carbon dioxide from thawed permafrost are important to stress, and our edits have further emphasized this within the passage indicated.

Edited passage now reads:

"Consequently, higher soil water content and inundated conditions for thawed permafrost would favor increased methane generation while drier environments would result in reduced methane production and a relatively higher fraction of $CO_2$ release (Blanc-Betes et al., 2016; Walter et al., 2001; Zhuang et al., 2004). Additionally, methanogenic bacteria also exhibit a strong temperature response, with experiments demonstrating a greater than hundred-fold increase in methane production for a temperature rise of $10°C$ (Metje and Frenzel, 2007; Rivkina et al., 2004; Tveit et al., 2015). However, warming temperatures themselves drive soil moisture effects that could reduce or increase methane fluxes if soils become respectively drier and more oxic or wetter and more anoxic in response to permafrost thaw (Blanc-Betes et al., 2016)."

Blanc-Betes, E., Welker, J. M., Sturchio, N. C., Chanton, J. P. and Gonzalez-Meler, M. A.: Winter precipitation and snow accumulation drive the methane sink or source strength of Arctic tussock tundra, Glob. Chang. Biol., 22(8), 2818–2833, doi:10.1111/gcb.13242, 2016.

Metje, M. and Frenzel, P.: Methanogenesis and methanogenic pathways in a peat from subarctic permafrost, Environ. Microbiol., 9(4), 954–964, doi:10.1111/j.1462-2920.2006.01217.x, 2007.

Rivkina, E., Laurinavichius, K., McGrath, J., Tiedje, J., Shcherbakova, V. and Gilichinsky, D.: Microbial life in permafrost, Adv. Sp. Res., 33(8), 1215–1221, doi:10.1016/j.asr.2003.06.024, 2004.

Tveit, A. T., Urich, T., Frenzel, P. and Svenning, M. M.: Metabolic and trophic interactions modulate methane production by Arctic peat microbiota in response

to warming, Proc. Natl. Acad. Sci. U. S. A., 112(19), E2507–E2516, doi:10.1073/pnas.1420797112, 2015.

van Winden, J. F., Reichart, G. J., McNamara, N. P., Benthien, A. and Damsté, J. S. S.: Temperature-induced increase in methane release from peat bogs: A mesocosm experiment, PLoS One, 7(6), 4–8, doi:10.1371/journal.pone.0039614, 2012.

Walter, B. P., Heimann, M. and Matthews, E.: Modeling modern methane emissions from natural wetlands: 1. Model description and results, J. Geophys. Res. Atmos., 106(D24), 34189–34206, doi:10.1029/2001JD900165, 2001. Wang, J. A., Sulla-Menashe, D., Woodcock, C. E., Sonnentag, O., Keeling, R. F. and Friedl, M. A.: Extensive land cover change across Arctic–Boreal Northwestern North America from disturbance and climate forcing, Glob. Chang. Biol., (July), 1–16, doi:10.1111/gcb.14804, 2019.

Zhuang, Q., Melillo, J. M., Kicklighter, D. W., Prinn, R. G., McGuire, A. D., Steudler, P. A., Felzer, B. S. and Hu, S.: Methane fluxes between terrestrial ecosystems and the atmosphere at northern high latitudes during the past century: A retrospective analysis with a process-based biogeochemistry model, Global Biogeochem. Cycles, 18(3), n/a-n/a, doi:10.1029/2004GB002239, 2004.

---

## Author Comment (AC2) · 15 Jun 2020

Asterisks denote reviewer comments

**Summary** **Thank you for this paper, it is a helpful addition to the literature, and it will be well-suited for publication following review. I have made several comments specifically in the introduction and ice-sheet sections, which I hope the authors will consider and find useful for improving the manuscript. My primary suggestion is the inclusion of a more thorough discussion on how uncertainty affects the physics, projections, and understanding of timescales of ice-sheet tipping points.**

[Figure]

**Line 7: "Shifting towards" seems a bit odd in the definition of climate tipping elements. It might be better to frame tipping elements in this context as a clearing a threshold, rather than the sign of a derivative? This wording appears more like it is referring to a climate anomaly, rather than moving to/arriving at a new climate paradigm.**

Following the reviewer's suggestions, we have revised this wording to better emphasize that the new system state represents a very different paradigm from the original and better reflect the threshold-based definition of tipping elements.

New text:

"Increasing attention is focusing upon "climate tipping elements" – large-scale earth systems anticipated to respond through positive feedbacks to anthropogenic climate change by transitioning towards dramatically different long-term states upon passing key thresholds."

**Line 35: This paragraph would be strengthened by including a mention of adaptation. Certainly once a tipping point has been passed, significant changes in planning, decision-making, and climate adaptation will need to be adopted.**

We have incorporated a mention of climate adaptation in this sentence based on this suggestion: "Many tipping mechanisms may also be difficult to halt, reverse, mitigate, or adapt to once they have begun shifting between states in response to climate perturbations."

**Line 94: As written, it was unclear to me what "0.54 in the high-end of the RCP8.5 warming scenario" means, or refers to. It probably needs to be either explained more directly and with appropriate context, or should be mentioned generally and more specifically covered in section 4.**

We have provided additional context before this sentence and have clarified this language: "and find the additional global mean surface temperature increase over the 21st century with the inclusion of tipping elements to be around 0.54C in the high-emissions
RCP8.5 scenario."

**Section 2.3.1: For context and contrast with the ice-sheet changes, it should be noted that sea-level rise from other components (thermal expansion, mountain glaciers) is also irreversible. . . but these do not necessarily have tipping points and exhibit more immediate gains upon mitigation (e.g. Solomon et al. 2009, Lenaerts et al. 2013, Zickfeld et al. 2016, Ehlert et al. 2018).**

We now note, with explanation, that these components contributing to sea-level rise are irreversible. After some consideration, we have decided it would be best to avoid elaborating further on threshold-like behaviour of glaciers and thermal expansion or exploring their response to different levels of mitigation, as these go beyond the scope of a section on ice sheet feedbacks that is already lengthly.

Added text at the end of the first paragraph of section 2.3.1: "While thermal expansion and loss of glaciers outside the GIS and AIS cause irreversible sea-level rise on human timescales due to the ocean's gradual response to warming (Ehlert and Zickfeld, 2018; Solomon et al., 2009; Zickfeld et al., 2017) and to shifts away from climate conditions that permit mountain glaciers to persist (Lenaerts et al., 2013), their relative contribution to future sea-level change will diminish (Clark et al., 2016). Mass losses from the Greenland and Antarctic ice sheets are similarly irreversible but will contribute the majority of expected future sea-level changes."

**Line 388: This paragraph would benefit from being very specific about what is meant by "less insulated". Why are they less insulated than the EAIS? The reasons and time frames are quite different between GIS and WAIS because of the differences in their physical responses to climate change (see the Hamlington et al. 2020 review), so grouping them like this confuses the message. Specificity and expanding this section would probably help clarify the point of this paragraph as I understand it: that there are heterogeneous tipping points across the polar ice sheets.**

We agree with the need indicated here to expand upon this passage. We have replaced

"less insulated to current and projected climate change" with "are more sensitive to current and projected climate change" and elaborated significantly on this section based on the reviewer's suggestion. In particular, we have moved the discussion of Greenland surface mass balance and related feedbacks up so that it directly follows this paragraph.

Edited passage and new transition into Greenland SMB paragraph are as follows: "Generally, ice basins of the GIS and WAIS are more sensitive to current and projected climate change and will likely reach key thresholds first, while the EAIS region responds more to higher intensities of warming (Golledge et al., 2015; Robinson et al., 2012). Significant ice loss is already occurring for both the GIS and WAIS in the present day, with an ongoing sea-level rise contribution of 1.20 mm/yr (Bamber et al., 2018; Oppenheimer et al., 2019; The IMBIE team, 2018). The EAIS is potentially at a mass balance (no net sea-level contribution) currently thanks to increased snowfall caused by a warming-induced increase in atmospheric moisture, although this balance is subject to considerable temporal variability and uncertainty that do not rule out the possibility of net loss since observations began (Bamber et al., 2018; Boening et al., 2012; Martin-Español et al., 2017; The IMBIE team, 2018; Velicogna et al., 2014).

The feedbacks affecting the major ice-sheets and the patterns and timeframes of their physical responses to climate change differ markedly between regions. In contrast to the Antarctic sheets, Greenland glaciers typically terminate on land prior to reaching the sea."

**Lines 400-405: Marine ice-sheet instability is confusingly explained in this section, and will likely lose some readers. It might be helpful to refer to other papers which explain this process (especially less technical and more general papers about MISI, such as Robel et al. 2019), and rewrite for clarity. Section 2.3.2: In the context of ice-sheet instabilities it would be helpful to more carefully introduce the flux rate across the grounding line as being proportional to the height at the grounding line, and then discuss where reverse sloping beds are found, and how that has the potential to affect

the (in)stability of WAIS and EAIS (e.g. see topography in Le Brocq et al. 2010).**

This passage was indeed unclear as originally written, and we have reorganized and revised it for clarity accordingly. We have also incorporated the suggested references and included additional discussion of the general topography of Antarctic ice sheet beds and the implications for WAIS and EAIS stability:

"However, marine warming represents the primary driver of Antarctic ice loss, as ice on the AIS margins is typically in direct contact with the ocean. Consequently, such marine-terminating glaciers are at risk of mass loss from processes that result from both oceanic warming (Shepherd et al., 2004) as well as atmospheric warming (Figure 3) (DeConto and Pollard, 2016). The observational record has established the key role of ocean-driven melt in the thinning and retreat of Antarctic ice shelves (Khazendar et al., 2016; Liu et al., 2015; Wouters et al., 2015), although ocean temperatures are themselves subject to atmospheric variability and forcing (Jenkins et al., 2016).

For marine-terminating ice-sheets, the rate at which ice flows out to sea is proportional to the height of the glacier above the grounding line beneath it where the submerged ice-sheet first contacts the bedrock below. On reverse slopes, where the bedrock's height decreases with further distance inland, this relationship results in a positive feedback where the height of the glacier above the grounding line increases as the ice-sheet retreats, which then further accelerates the rate at which the glacier flows out to sea (Schoof, 2007). This irreversible positive feedback mechanism for ice loss is called Marine Ice Shelf Instability (MISI) (Thomas and Bentley, 1978; Weertman, 1974). While not all vulnerable Antarctic glaciers terminate on reverse slopes and are currently thought to be undergoing MISI today, several major basins are currently retreating thanks to processes that may indicate MISI dynamics (Favier et al., 2014; Joughin et al., 2014; Rignot et al., 2014). In this process, warm subsurface waters cause melt beneath ice shelves, resulting in inland retreat of the grounding line (Shepherd et al., 2004). Should grounding lines retreat beyond forward slopes and onto reverse-sloping topography, as is the potential case for many major Antarctic basins

(Ross et al., 2012), the process of MISI would begin rapidly accelerating ice loss from many major glaciers. Much of the WAIS lies on reverse-sloping bedrock well below sea-level (Le Brocq et al., 2010), leading to generally higher susceptibility to MISI and greater instability of the WAIS under modest warming scenarios relative to the EAIS (Pattyn et al., 2018)."

**Line 409: 90m is not an exact threshold for MICI, it was a suggested model parameter value proposed in DeConto and Pollard (2016). Rather than speculate on these parameter values which are most relevant in a modeling context, it would be much more useful to instead rewrite this sentence describing the physical phenomenon being discussed, e.g., "MICI postulates that ice cliffs become unstable and collapse under theirown weight if they exceed a critical height threshold (Pollard et al. 2015)"**

We agree that specific parameter values should not be included here and have revised the text along the suggested lines:

"a feedback mechanism known as Marine Ice Cliff Instability (MICI) may trigger at locations where the height of cliffs at an exposed ice-sheet's edge exceeds critical thresholds. Beyond such heights, the shear strength of ice would be insufficient to withstand longitudinal stress at the cliff face"

**Line 416: In introducing MICI, it would be useful to mention Clerc et al. (2019), which argues that MICI is unlikely given viscous relaxation dominating the response to iceshelf removal. In contrast to Edwards et al. (2019) which makes a modeling/statistical argument, Clerc et al. makes a physical one.**

This is indeed an important reference and we have now included it in our discussion of MICI in this section.

Line 425: This paragraph is missing a key discussion on the deep uncertainty surrounding the Antarctic ice-sheet response. Deep uncertainty is characterized by the lack of agreement between experts (e.g. Bakker et al. 2017, Bamber et al. 2019), which has

critical implications for decision-maker response to possible AIS tipping points (Rasmussen et al. 2020).

We have revised the manuscript to elaborate upon the large uncertainties regarding ice sheet dynamics and the resulting wide range in sea-level projections within the literature and as shown by structured expert judgement. However, we have chosen to discuss this at greater length in Section 2.3.3 rather than in this paragraph focused on MICI. We have added a sentence to the MICI paragraph to guide readers to this discussion in the following subsection.

**Line 425: Likewise missing is a discussion that our understanding of ice-sheet tipping points comes primarily from the paleoclimate record, at points in time when we know the AIS was at least partially deglaciated (e.g. Dutton et al. 2015, Capron et al. 2019). Current paleoclimate understanding has limited power in constraining our understanding of ice-sheet instabilities (Edwards et al. 2019), but improved geological estimates have potential to reduce future projection uncertainties (Gilford et al. 2020, in revision). Modern mass loss rates (e.g. from satellite measurements of SMB model estimates) have very little efficacy for reducing uncertainty on decadal time scales (Kopp et al. 2017).**

Following this suggestion, we have added a passage discussing the role of paleoclimate work in assessing the response of future sea-level rise and ice sheet loss to climate change and the associated limitations (lines). We have included these suggested references aside from the paper in review, also referencing a couple of other relevant studies:

"Paleoclimate evidence points strongly towards past instances of significant ice-sheet loss and associated global mean sea-level rise during past warming episodes, Researchers have leveraged such paleorecords to estimate the response of global sea-levels to global mean temperatures (Levermann et al., 2013) and atmospheric CO2 concentrations (Foster and Rohling, 2013). However, uncertainties remain large due

to the limitations of proxy data, including the resolution of time dating and the need to extrapolate regionally or globally from available datasets. Paleoclimate records also possess limited ability to resolve key temperature thresholds for major ice-sheet loss, the ice-sheet mechanisms responsible (Edwards et al., 2019), and historical rates and magnitudes of sea-level rise, as reviewed by (Capron et al., 2019; Dutton et al., 2015). Nevertheless, further paleoclimate research carries the potential to better constrain the sensitivity of ice-sheets and global sea-level rise during periods that could serve as historical parallels to current warming."

**Lines 445 and 482: It is unnecessary to continue mentioning the uncertainty high-lighted by Edwards et al. (2019). Instead it would be better to flush out the original discussion in line 425, noting how it brings in deep uncertainty reflected in Bamber et al. (2019), and highlighting that anytime someone refers to a MICI scenario is a high-end or worst-case scenario being considered (such as the excellent line 477).**

We now omit these redundant statements and have revised the indicated passage to highlight the nature of MICI scenarios as upper-end: "studies including MICI dynamics arguably represent high-end or worst-case scenarios based on ice cliff collapse mechanisms not yet fully validated by field observations"

**Section 2.3.3 and e.g. Line 490: It would be very helpful to reference the Bamber et al. (2019) expert judgement paper for projections; it is probably our best prior for projected ice-sheet mass losses.**

We have followed this suggestion and highlighted this reference within this section to emphasize the wide uncertainties associated with end-of-century projections and beyond: "Nevertheless, projections of future sea-level rise from ice sheet losses remain highly uncertain, primarily due to limited observational records, incomplete understanding of ice sheet dynamics, and model limitations. Estimates of ice sheets' contribution to sea-level by 2090 evident across a selection of published literature span a fourfold range (Bakker et al., 2017), with structured expert judgement yielding a similarly wide

range of estimates (median estimate: 0.51 m, 95th percentile estimate: 1.78 m) (Bamber et al., 2019)."

\*\*Line 505: This should read "Consequently, under our current best understanding, icesheet collapse..." It's important to note that given the deep uncertainty (e.g. if MICI is possible), then changes could actually be abrupt. We just don't have a good enough grasp on the physics yet, and our observations (paleo and modern) are inadequate to constrain this property. Furthermore, because of these deep uncertainties, without progression of the science it may be impossible to know whether we are on a trajectory towards a tipping point (for instance, initated MICI) until we have already crossed it (e.g. Kopp et al. 2017, their Figure 5).\*\*

We have rephrased this sentence accordingly based on the reviewer's suggestion: "Consequently, under our current best understanding, ice sheet collapse cannot generally be considered an abrupt phenomenon"

\*\*Suggestions for Minor grammatical/structural/citation changes:\*\* \*\*-Line 7: add comma after "some"\*\*

To maintain the flow of the introductory element to this sentence, we elect to omit this change.

\*\*-Line 28 and elsewhere: "sea level rise" should be "sea-level rise". Please adjust accordingly throughout (e.g. lines 356, 358, 377, 386), and likewise for places where "ice sheet" should be "ice-sheet" (e.g. line 368, 377, 386)\*\*

These adjustments have been made.

\*\*-Line 40: Should this be Turetsky et al. (2020)?\*\*

This is the case – change made.

\*\*-Line 44: "larger uncertainty" is a strange word choice here. I think this is referring to ambiguity (Ellsberg 1961)\*\*

Rephrased as suggested.

**-Line 74: It will probably be addressed in type-setting, but this line is not formatted correctly**

ESD submission format requests a justified rather than right-aligned layout, and this should indeed be addressed during type-setting.

**-Line 87-88: This sentences reads oddly as written, with subject confusion, etc. "the climate impact of the" should be "the climate impact of a", and "uncertainties surrounding factors required" is very wordy. It should be rewritten for clarity**

We have rewritten this sentence for greater clarity: "Evaluation of the risks posed by climate tipping elements requires considering their timescales of action, climate impacts, and important uncertainties surrounding triggering thresholds and associated factors."

**-Throughout section 2.3: the acronyms for AIS, GIS, EAIS, and WAIS need to be introduced only once (in lines 360 and 371-372)**

We have made this change.

**-Line 360: add "By contrast" before "The Greenland Ice Sheet"**

Addition made as suggested.

**-Line 361: can this be more specific than "recently"? Maybe "accelerated over the past four decades (IMBIE 2018)"**

This is a good suggestion and we now employ this wording.

**-Line 365: A full review of the current understanding of modern ice-sheet contributions to sea-level rise is available in the in press Hamilington et al. (2020) review**

We now reference this review at the start of section 2.3.3.

**-Lines 370-374: This sentence is written very oddly and is difficult to follow. I suggest rewriting for clarity**

We have rewritten this sentence for clarify and brevity: "Climate change is expected to cause collapse of the GIS and large-scale losses from the West Antarctic Ice-sheet (WAIS) at lower levels of climate forcing, followed by further Antarctic ice loss as vulnerable basins of the East Antarctic Ice-sheet (EAIS) retreat under higher levels of warming (Meredith et al., 2019)."

**-Line 377: Suggest rewriting this sentence as "Major ice-sheet processes have had a dominating influence on sea-level patterns: : :"**

Rewritten to "Major ice-sheet processes have exerted a dominating influence on sea-levels over the geologic past"

**-Line 380: Add "global mean" before "increase of 3.4m"**

Addition made.

**-Line 392: Add "(no net sea-level rise)" after "mass balance", for clarity**

Addition made.

**-Line 392-393: Remove "and precipitation", it is redundant, and add that it is related to increased atmospheric moisture from increasing temperatures.**

Edits made.

**-Line 393: Add "this balance is" before "subject to considerable. . ." for clarity**

This has been clarified as suggested.

**-Line 395: Rewrite the beginning of this sentence as "Ice on the AIS margins is Typically. . .", and remove "at their edges" from the end of the sentence**

Changes made.

**-Line 401: Suggest citing Weertman (1974) at the beginning of the MISI discussion**

After consideration, the current placement of this reference seems appropriate to us.

[Figure]

\*\*-Line 406: Add "can" after "heating"\*\*

Addition made.

\*\*-Line 409: Replace "is" with "would be"\*\*

Replacement made.

\*\*-Line 412: Please add a citation for the 800m threshold\*\*

This sentence was removed and is no longer present in the revised manuscript.

\*\*-Line 453: Should "proving" be "providing"?\*\*

This is indeed a typo and has been corrected.

\*\*-Line 456: "net" before "feedbacks" would help clarify this sentence\*\*

Edit made.

\*\*-Line 457: Is this referring to geological field observations?\*\*

Yes. We have clarified this.

\*\*-Line 457: One key warming threshold paper for GIS that should be mentioned is Robinson et al. (2012)\*\*

We have now cited this paper further above towards the start of Section 2.3.2.

\*\*-Line 478: Add "(but very uncertain)" before "MICI feedback,"\*\*

Addition made.

\*\*-Line 489: Would this discussion be helped by adding a sentence or two about the regional variability of ice-sheet loss projections (e.g. Kopp et al. 2017)?\*\*

This is a good suggestion and we have added a sentence to this effect: "Model projections also produce variable results for the magnitude and distribution of future regional sea-level rise if different assumptions are used to model Greenland and Antarctic ice

loss (Kopp et al., 2017)."

**-Line 495: Do albedo feedbacks also make it more difficult to stabilize the GIS?**

We have clarified this.

**-Line 496: Add "ice-sheet" between "individual basins,"**

Addition made

**-Line 517: "instability" might read better as "instabilities"**

Change made.

New references featured in this comment:

Bakker, A. M. R., Louchard, D. and Keller, K.: Sources and implications of deep uncertainties surrounding sea-level projections, Clim. Change, 140(3–4), 339–347, doi:10.1007/s10584-016-1864-1, 2017.

Bamber, J. L., Oppenheimer, M., Kopp, R. E., Aspinall, W. P. and Cooke, R. M.: Ice sheet contributions to future sea-level rise from structured expert judgment, Proc. Natl. Acad. Sci. U. S. A., 166(23), 11195–11200, doi:10.1073/pnas.1817205116, 2019.

Capron, E., Rovere, A., Austermann, J., Axford, Y., Barlow, N. L. M., Carlson, A. E., de Vernal, A., Dutton, A., Kopp, R. E., McManus, J. F., Menviel, L., Otto-Bliesner, B. L., Robinson, A., Shakun, J. D., Tzedakis, P. C. and Wolff, E. W.: Challenges and research priorities to understand interactions between climate, ice sheets and global mean sea level during past interglacials, Quat. Sci. Rev., 219, 308–311, doi:10.1016/j.quascirev.2019.06.030, 2019.

Ehlert, D. and Zickfeld, K.: Irreversible ocean thermal expansion under carbon dioxide removal, Earth Syst. Dyn., 9(1), 197–210, doi:10.5194/esd-9-197-2018, 2018.

Foster, G. L. and Rohling, E. J.: Relationship between sea level and climate forcing by CO2 on geological timescales, Proc. Natl. Acad. Sci. U. S. A., 110(4), 1209–1214,

doi:10.1073/pnas.1216073110, 2013.

Khazendar, A., Rignot, E., Schroeder, D. M., Seroussi, H., Schodlok, M. P., Scheuchl, B., Mouginot, J., Sutterley, T. C. and Velicogna, I.: Rapid submarine ice melting in the grounding zones of ice shelves in West Antarctica, Nat. Commun., 7, 1–8, doi:10.1038/ncomms13243, 2016.

Kopp, R. E., DeConto, R. M., Bader, D. A., Hay, C. C., Horton, R. M., Kulp, S., Oppenheimer, M., Pollard, D. and Strauss, B. H.: Evolving Understanding of Antarctic Ice-Sheet Physics and Ambiguity in Probabilistic Sea-Level Projections, Earth's Futur., 5(12), 1217–1233, doi:10.1002/2017EF000663, 2017.

Jenkins, A., Dutrieux, P., Jacobs, S., Steig, E. J., Gudmundsson, G. H., Smith, J. and Heywood, K. J.: Decadal ocean forcing and Antarctic ice sheet response: Lessons from the Amundsen Sea, Oceanography, 29(4), 106–117, doi:10.5670/oceanog.2016.103, 2016.

Le Brocq, A. M., Payne, A. J. and Vieli, A.: An improved Antarctic dataset for high resolution numerical ice sheet models (ALBMAP v1), Earth Syst. Sci. Data, 2(2), 247–260, doi:10.5194/essd-2-247-2010, 2010.

Lenaerts, J. T. M., Van Angelen, J. H., Van Den Broeke, M. R., Gardner, A. S., Wouters, B. and Van Meijgaard, E.: Irreversible mass loss of Canadian Arctic Archipelago glaciers, Geophys. Res. Lett., 40(5), 870–874, doi:10.1002/grl.50214, 2013.

Levermann, A., Clark, P. U., Marzeion, B., Milne, G. A., Pollard, D., Radic, V. and Robinson, A.: The multimillennial sea-level commitment of global warming, Proc. Natl. Acad. Sci. U. S. A., 110(34), 13745–13750, doi:10.1073/pnas.1219414110, 2013.

Liu, Y., Moore, J. C., Cheng, X., Gladstone, R. M., Bassis, J. N., Liu, H., Wen, J. and Hui, F.: Ocean-driven thinning enhances iceberg calving and retreat of Antarctic ice shelves, Proc. Natl. Acad. Sci. U. S. A., 112(11), 3263–3268, doi:10.1073/pnas.1415137112, 2015.

[Figure]

Robinson, A., Calov, R. and Ganopolski, A.: Multistability and critical thresholds of the Greenland ice sheet, Nat. Clim. Chang., 2(6), 429–432, doi:10.1038/nclimate1449, 2012.

Solomon, S., Plattner, G. K., Knutti, R. and Friedlingstein, P.: Irreversible climate change due to carbon dioxide emissions, Proc. Natl. Acad. Sci. U. S. A., 106(6), 1704–1709, doi:10.1073/pnas.0812721106, 2009.

Zickfeld, K., Solomon, S. and Gilford, D. M.: Centuries of thermal sea-level rise due to anthropogenic emissions of short-lived greenhouse gases, Proc. Natl. Acad. Sci. U. S. A., 114(4), 657–662, doi:10.1073/pnas.1612066114, 2017.
* * *

---

## Short Comment (SC8) · 24 Aug 2020

With this post I would like to clarify that Earth System Dynamics (ESD) as well as EGU publications and Copernicus Publications in general encourage an open, transparent, and inclusive scientific peer-review process, irrespective of the academic qualification of the commenter.

The comments in the forum should be fair, objective, and constructive. They should be comprehensive and be representative of a scientific review, that is, collect multiple points of comments and critiques on the manuscript in order to assess its validity and merits, as well as to identify potential points where the manuscript can be improved and

strengthened. The contributions in the discussion forum should not, however, be used as a chat-like forum. Such quick notes could be exchanged with the corresponding authors directly by e-mail, with the contact information provided in the manuscript.

Axel Kleidon ESD Co-Chief Editor

This note was coordinated with Martin Rasmussen (Copernicus) and Katja Fennel (chair, EGU Publications Committee)
* * *

---

## Referee Comment (RC1) · Anonymous Referee #1 · 14 Sep 2020

Review of ESD Reviews: mechanisms, evidence, and impacts of climate tipping elements, by Wang and Hausfather

This is an unusual paper to review. During a first read, it reminded me of a student assignment more than a review paper in a scientific journal. In a number of places it even made me cringe. Throughout, this paper gives the impression that the authors are not very familiar with the subject, have no real ability to assess the papers they discuss, present a questionable subset of the relevant literature (missing quite a lot) and appear to be well out of their depth. Made curious by the strange reading experience, I looked up the authors and was surprised to find that they do not have a

track record of relevant research, and they do not work at a climate science institution but at an agenda-driven "think tank" conducting "research focused on clean energy innovation, energy efficiency, and energy for human development", according to its self-description. Unfortunately this lack of expertise on climate tipping points shows in every paragraph. Going through everything which is wrong about this paper would require a review as long as the actual manuscript, and far more time than I am willing to dedicate to such a low-quality manuscript. Thus I will limit myself to a couple of examples. The authors introduce a thoroughly confusing terminology by mixing up tipping elements (e.g. the Greenland ice sheet or the Amazon forest) and the associated tipping points (which are not physical entities but theoretical concepts). They write: "We also evaluate which tipping elements are more imminent..." A tipping element cannot be "imminent". "Some tipping elements are perhaps more accurately termed climate feedbacks" This is complete nonsense, the Greenland ice sheet or the Amazon forest are not "feedbacks". Feedbacks are mechnisms, not physical entities. "Climate tipping elements" – large-scale mechanisms or systems" A tipping element is a system and not a mechanism. "In this report, we have adopted the convention proposed by (Kopp et al., 2016) to maximize clarity, characterizing rapidly acting (within a decade) systems as "tipping points" and otherwise utilizing the term "tipping elements". We describe systems with a more linear, direct, predictable response to climate forcing simply as "climate feedbacks". Again, a tipping element is a physical, tangible system like an ice sheet, and it can have a tipping point, but it can neither be a tipping point nor a feedback. I was amazed to read that the authors claim this terminology is that of Kopp et al. and checked this reference – and of course it is not. Kopp et al. explain: "In the literal example of the rail wagon of coal, the wagon itself would be the tipping element; the point at which the wagon's physical dynamics commit it to falling on its side and emptying its contents would be the tipping point." And they write: "We propose terminology to clarify the distinction between "tipping points" in the popular sense, the critical thresholds exhibited by climatic and social "tipping elements," and "economic shocks." So the distinction they propose is between the popular "tipping points" and

the "critical thresholds" – not the tipping elements. Another random example is that the authors claim: "The first few years of observations from OSNAP have also produced findings that have dramatically changed the current understanding of the AMOC system. These measurements have revealed that in contrast to the dominant paradigm that most NADW originates from the Labrador Sea [. . .]" Here they uncritically repeat the wildly overblown rhetoric with which the first OSNAP results were promoted by one project scientist, which is an example for how Wang and Hausfather lack any ability to judge such claims. A few years of data in a highly variable system can hardly dramatically change the current understanding of the AMOC, and of course have done no such thing. And so it goes on. I do not see any hope that a major revision could turn this into a publishable paper, since I do not see how the authors have the expertise or understanding to do this.

---

## Referee Comment (RC2) · Anonymous Referee #2 · 17 Nov 2020

At first I was was excited to read this paper as a review on current understanding of climate tipping elements and where the gaps in understanding lie and where the research community should focus their efforts. However, while the authors attempt to clear up some of the terminology around what a tipping element is, they actually end up creating more confusion which suggests the authors do not have a good understanding themselves. A tipping element is a physical component of the Earth system, which is characterized by a threshold over which it can transition into a different state under perturbations of the climate system (e.g. Lenton et al. 2008). Thus, rising air temperatures could push a component of the Earth system, i.e. the complete collapse of the west Antarctic ice sheet that will not recover once air temperatures return back

to lower temperatures. Some of these tipping points could be reinforced by feedbacks, but feedbacks are not climate tipping elements. Thus, the entire way the introduction is written makes it clear that the authors do not fully understand the basic definition of climate tipping elements, climate tipping points or climate feedbacks. Based on the confusion terminology presented in the introduction, I decided not to continue reading this very lengthy paper. While a synthesis paper on our current understanding of which components of the Earth system are characterized by threshold behavior, and how close to a tipping point each of the elements are, the impacts of these tipping points, is worthwhile, yet I feel this paper fails to achieve that in its present form. I'm not sure what we learn from what is already presented in Lenton et al. 2008. If we updated some of the tipping elements discussed in that paper with more recent knowledge, that could be helpful but this paper fails to do that. Also this is an incredibly lengthy paper which feels more like a report rather than a synthesis paper. The format of Lenton et al. (2008) is much more appropriate, which at the time provided a critical view of our understanding of which Earth system components were likely to exhibit threshold behavior and what the realistic risks are and how policy can help.

I feel that the authors will have to completely redesign this paper before it can be considered in any journal.

---

## Author Comment (AC3) · 14 Dec 2020

While we thank Referee 1 for their comments on the manuscript, we strongly believe that criticism of our background and employer are inappropriate in the context of this review. Both of us have relevant academic qualifications to do this work, and have published extensively on related topics. At least in the US, it is not at all unusual for researchers working at academic non-profit organizations to contribute to the peer-reviewed literature.

Furthermore, prior to submission of the document we obtained external evaluation from subject matter experts of each topical chapter to ensure that our review of each can-

didate tipping element was accurate and up-to-date. These evaluators included those named in the acknowledgements: Sijia Zou, Nicholas Foukal, David Archer, Olivier Gagliardini, Ted Schuur, Dirk Notz, William Boos, Laifang Li, Deepti Singh, Daniel Nepstad, Laura Borma, Beth Lenz, Heidi Hirsch, Tapio Schneider, Jacquelyn Shuman, Adrianna Foster, and Richard Betts.

We have worked to resolve the referee concerns about terminology used in describing tipping elements, and extensively updated all sections of the review to reflect the latest literature (and CMIP6 results) that have been published during the past 9 months since this paper was originally submitted. We hope that in light of these revisions you would be willing to reassess the paper; we feel that there is an important gap in the literature at the moment due to the lack of a comprehensive review of climate tipping elements, and hope to provide it here.

We agree entirely with both reviewers that the proper usage of terminology is fundamental to a constructive review of climate tipping elements. Based on the reviewers' feedback, we have thoroughly revised our use of terminology throughout the manuscript. While the terminology adopted previously was internally consistent within the review, both reviewers have accurately pointed out that the usage conflicted with that employed by Kopp et al., 2016. Both reviewers also highlighted that while earth systems drive climate feedbacks, it is not appropriate to label them as feedbacks per se. Agreeing that these revisions to terminology represent essential prerequisites for satisfactorily improving this review, we have now revised the introduction to utilize terminology fully consistent with that proposed by Kopp et al., and revisited the manuscript text, figures, and tables throughout to ensure that the same terminology is utilized throughout the article.

We do however contend that the classification of some tipping elements as "imminent" possesses value in the context of ongoing anthropogenic forcing. Of course a tipping element is not intrinsically imminent or distant, but when considering current rates of greenhouse gas emissions and land use change, it is useful for a review like this to

highlight tipping elements for which current levels of forcing are rapidly approaching critical thresholds.

We agree with the reviewer on the usefulness of better qualifying initial reports on OS-NAP array measurements, and have moderated these statements while also adding the following sentence: "However, the time series of observations from this new monitoring array remains quite short, with continued measurements likely necessary to further substantiate these findings." We do note however that the physical oceanography community is taking this new research finding quite seriously, and while the new measurements are not fully conclusive given high AMOC variability on short timescales, it would not be an exaggeration to note that Lozier et al., 2019 is prompting researchers to reassess the contribution of LSW formation to AMOC.

We have also added additional discussion to further point out that the OSNAP results are hardly the only evidence of a weaker role of Labrador Sea convection in the AMOC, with prior findings from field deployment of RAFOS floats and Lagrangian modeling both raising questions regarding the strength of this linkage (Zou and Lozier, 2016).

In addition to addressing the reviewers' feedback and comments, we have furthermore considerably updated the body of the revised manuscript to incorporate recent new literature related to each sub-topic that has been published since this review was originally submitted, as well as other relevant literature brought to our attention.

The AMOC: In addition to the above-mentioned changes to the section on the AMOC, we have added several recently-published findings on the lack of a distinct trend in the AMOC's measured and estimated in recent decades, while also incorporating new modeling analyses showing stronger 21st century weakening of the AMOC in many CMIP6 models.

Marine methane hydrates: We have updated Table 2 with results from the latest Global Methane Budget paper (Saunois et al., 2020) and referenced a handful of recent relevant additional publications. Upon reflection we have also opted to include a passage

on the diverse body of work quantifying and discussing methane fluxes from the East Siberian Arctic Shelf, as while such seepage is neither climatically significant nor the result of anthropogenic forcing, the topic has remained sufficiently proximate to the subject of potential climate impacts from methane hydrates to warrant a concise mention.

Ice-sheet collapse: we have updated this section with recently-published relevant literature examining critical thresholds, positive feedbacks, and tipping behavior for both the Antarctic and Greenland Ice Sheet, including a new study reporting preliminary 2100 SLR projections from ISMIP6. We have also added passages on paleoclimate studies assessing historic ice sheet extent and response under past climates, on atmospheric forcing of Antarctic ice shelves, on ocean forcing of the GIS, and on the wide uncertainty range of future sea-level rise projections associated with ice-sheet loss.

Permafrost thaw and carbon release: Numerous additions. Deeper discussion of resiliency and vulnerability of different permafrost landscapes. Inclusion of additional discussion of changes to northern Arctic hydrology and associated implications for mineralization of permafrost organic carbon. Addition of a passage detailing limitations of current climate models' representation of permafrost dynamics. More detailed discussion of feedbacks between temperature-induced promotion of plant growth and permafrost thaw. Specific updates to literature reflecting the latest Global Methane Budget, initial assessments of CMIP6 performance, and other relevant new papers.

Boreal forest ecosystem shifts: Further discussion of competing feedbacks due to shifts in plant productivity/respiration and composition of boreal vegetation in response to changing climate and fire regime.

Stratocumulus cloud deck evaporation: Addition of a single new reference by the same research group exploring the mitigation potential of solar geoengineering upon this climate tipping element.

Coral reef biodiversity collapse: Addition of some relevant recent papers of interest.

Amazon rainforest dieback: Presentation of updated findings on current rates of defor-estation, carbon loss, and forest degradation. Inclusion of new regional studies incor-porating results from CMIP6 models, as well as addition of a passage detailing ongoing developments in hydraulic modeling of the response of plant physiology to water stress.

Monsoons: Added discussion of projected changes in monsoon rainfall under CMIP6 models.

Loss of summer Arctic sea ice: Addition of a relevant paper describing CMIP6 Arctic sea ice modeling results, along with some other relevant recent literature.

Zou, S. and Lozier, M. S.: Breaking the linkage between labrador sea water production and its advective export to the subtropical gyre, J. Phys. Oceanogr., 46(7), 2169–2182, doi:10.1175/JPO-D-15-0210.1, 2016.

Saunois, M., Stavert, A. R., Poulter, B., Bousquet, P., Canadell, J. G., Jackson, R. B., Raymond, P. A., Dlugokencky, E. J., Houweling, S., Patra, P. K., Ciais, P., Arora, V. K., Bastviken, D., Bergamaschi, P., Blake, D. R., Brailsford, G., Bruhwiler, L., Carlson, K. M., Carrol, M., Castaldi, S., Chandra, N., Crevoisier, C., Crill, P. M., Covey, K., Curry, C. L., Etiope, G., Frankenberg, C., Gedney, N., Hegglin, M. I., Höglund-Isaksson, L., Hugelius, G., Ishizawa, M., Ito, A., Janssens-Maenhout, G., Jensen, K. M., Joos, F., Kleinen, T., Krummel, P. B., Langenfelds, R. L., Laruelle, G. G., Liu, L., Machida, T., Maksyutov, S., McDonald, K. C., McNorton, J., Miller, P. A., Melton, J. R., Morino, I., Müller, J., Murguia-Flores, F., Naik, V., Niwa, Y., Noce, S., O'Doherty, S., Parker, R. J., Peng, C., Peng, S., Peters, G. P., Prigent, C., Prinn, R., Ramonet, M., Regnier, P., Riley, W. J., Rosentreter, J. A., Segers, A., Simpson, I. J., Shi, H., Smith, S. J., Steele, L. P., Thornton, B. F., Tian, H., Tohjima, Y., Tubiello, F. N., Tsuruta, A., Viovy, N., Voulgarakis, A., Weber, T. S., van Weele, M., van der Werf, G. R., Weiss, R. F., Worthy, D., Wunch, D., Yin, Y., Yoshida, Y., Zhang, W., Zhang, Z., Zhao, Y., Zheng, B., Zhu, Q., Zhu, Q. and Zhuang, Q.: The Global Methane Budget 2000–2017, Earth Syst. Sci. Data, 12(3), 1561–1623, doi:10.5194/essd-12-1561-2020, 2020.

---

## Author Comment (AC4) · 14 Dec 2020

We agree entirely with both reviewers that the proper usage of terminology is fundamental to a constructive review of climate tipping elements. Based on the reviewers' feedback, we have thoroughly revised our use of terminology throughout the manuscript. While the terminology adopted previously was internally consistent within the review, both reviewers have accurately pointed out that the usage conflicted with that employed by Kopp et al., 2016. Both reviewers also highlighted that while earth systems drive climate feedbacks, it is not appropriate to label them as feedbacks per se. Agreeing that these revisions to terminology represent essential prerequisites for

satisfactorily improving this review, we have now revised the introduction to utilize terminology fully consistent with that proposed by Kopp et al., and revisited the manuscript text, figures, and tables throughout to ensure that the same terminology is utilized throughout the article.

In focusing primarily on terminology, however, we note that the reviewers have largely omitted to discuss the considerable value of the literature reviews we have presented for each earth system component covered. We point out that we solicited the input and feedback of leading experts in the process of preparing each subsection, in order to ensure that our synthesis accurately reflected the current state of knowledge on each sub-topic.

While our chosen format for this review has produced a long manuscript overall, we point out that each of ten candidate tipping elements as well as a section discussing their potential to interact synergistically is presented in just a few pages - close to the minimum length necessary to provide background upon each sub-topic and associated concepts for non-specialists while still highlighting the key relevant findings and current research directions. Our chosen format thus covers a wider breadth of topics than an individual specialized review paper focusing on a single candidate tipping element, enabling comparison and discussion of potential interactions, while also providing needed detail and depth that the short format of Lenton et al., 2008 or Steffen et al., 2018 would not allow. The lack of any comprehensive review of tipping elements in the literature is a notable gap; numerous members of the research community have reached out to us following the online publication of the pre-print, expressing that they found this review to be helpful and see it as a valuable contribution to the literature.

We strongly disagree with the referee's assertion that this review fails to update the scientific understanding of the tipping elements discussed in Lenton et al., 2008. The review presents significant updates in the state of scientific knowledge driven by new research, referencing a sizable body of recent literature, incorporating feedback from experts in each sub-field, and in some cases directly highlighting shifts in scientific

opinion on specific candidate tipping elements that have taken place since the publication of Lenton et al., 2008, as is the case for assessing the potential climate impact of marine methane hydrate dissociation and for projecting future changes in monsoon circulation, for instance.

In addition to addressing the reviewers' feedback and comments, we have furthermore considerably updated the body of the revised manuscript to incorporate recent new literature related to each sub-topic that has been published since this review was originally submitted, as well as other relevant literature brought to our attention.

The AMOC: In addition to the changes to the section on the AMOC as detailed in our response to Referee #1, we have added several recently-published findings on the lack of a distinct trend in the AMOC's measured and estimated in recent decades, while also incorporating new modeling analyses showing stronger 21st century weakening of the AMOC in many CMIP6 models.

Marine methane hydrates: We have updated Table 2 with results from the latest Global Methane Budget paper (Saunois et al., 2020) and referenced a handful of recent relevant additional publications. Upon reflection we have also opted to include a passage on the diverse body of work quantifying and discussing methane fluxes from the East Siberian Arctic Shelf, as while such seepage is neither climatically significant nor the result of anthropogenic forcing, the topic has remained sufficiently proximate to the subject of potential climate impacts from methane hydrates to warrant a concise mention.

Ice-sheet collapse: we have updated this section with recently-published relevant literature examining critical thresholds, positive feedbacks, and tipping behavior for both the Antarctic and Greenland Ice Sheet, including a new study reporting preliminary 2100 SLR projections from ISMIP6. We have also added passages on paleoclimate studies assessing historic ice sheet extent and response under past climates, on atmospheric forcing of Antarctic ice shelves, on ocean forcing of the GIS, and on the wide

uncertainty range of future sea-level rise projections associated with ice-sheet loss.

Permafrost thaw and carbon release: Numerous additions. Deeper discussion of resiliency and vulnerability of different permafrost landscapes. Inclusion of additional discussion of changes to northern Arctic hydrology and associated implications for mineralization of permafrost organic carbon. Addition of a passage detailing limitations of current climate models' representation of permafrost dynamics. More detailed discussion of feedbacks between temperature-induced promotion of plant growth and permafrost thaw. Specific updates to literature reflecting the latest Global Methane Budget, initial assessments of CMIP6 performance, and other relevant new papers.

Boreal forest ecosystem shifts: Further discussion of competing feedbacks due to shifts in plant productivity/respiration and composition of boreal vegetation in response to changing climate and fire regime.

Stratocumulus cloud deck evaporation: Addition of a single new reference by the same research group exploring the mitigation potential of solar geoengineering upon this climate tipping element.

Coral reef biodiversity collapse: Addition of some relevant recent papers of interest.

Amazon rainforest dieback: Presentation of updated findings on current rates of deforestation, carbon loss, and forest degradation. Inclusion of new regional studies incorporating results from CMIP6 models, as well as addition of a passage detailing ongoing developments in hydraulic modeling of the response of plant physiology to water stress.

Monsoons: Added discussion of projected changes in monsoon rainfall under CMIP6 models.

Loss of summer Arctic sea ice: Addition of a relevant paper describing CMIP6 Arctic sea ice modeling results, along with some other relevant recent literature.

[Figure]

2020.